# Widespread and intron-rich mirusviruses are predicted to reproduce in nuclei of unicellular eukaryotes

**Sofia Medvedeva** [1,8], **Ulysse Guyet** [2,3,8], **Eric Pelletier** [2,3], **Hans-Joachim Ruscheweyh** [4], **Shinichi Sunagawa** [4], **Hiroyuki Ogata** [5], **Frank O. Aylward** [6], **Morgan Gaïa** [2,3], **Natalya Yutin** [7], **Eugene V. Koonin** [7,9] ✉, **Mart Krupovic** [1,9] ✉ & **Tom O. Delmont** [2,3,9] ✉

Mirusviruses infect unicellular eukaryotes and are related to tailed bacteriophages and herpesviruses. Here we expand the known diversity of mirusviruses by screening diverse metagenomic assemblies and characterizing 1,202 non-redundant environmental genomes. *Mirusviricota* comprises a highly diversified phylum of large and giant eukaryotic viruses that rivals the evolutionary scope and functional complexity of nucleocytoviruses. Critically, major *Mirusviricota* lineages lack essential genes encoding components of the replication and transcription machineries and, concomitantly, encompass numerous spliceosomal introns that are enriched in virion morphogenesis genes. These features point to multiple transitions from cytoplasmic to nuclear reproduction during mirusvirus evolution. Many mirusvirus introns encode diverse homing endonucleases, suggestive of a previously undescribed mechanism promoting the horizontal mobility of spliceosomal introns. Available metatranscriptomes reveal long-range trans-splicing in a virion morphogenesis gene. Collectively, our data strongly suggest that nuclei of unicellular eukaryotes across marine and freshwater ecosystems worldwide are a major niche for replication of intron-rich mirusviruses.

The phylum *Mirusviricota* consists of double-stranded DNA viruses with large and complex genomes[1]. Mirusviruses are abundant in marine and freshwater ecosystems where they are predicted to infect a broad range of unicellular eukaryotes (protists)[1–7]. Integrated mirusvirus genes have been identified in genomes from a wide range of eukaryotic lineages, suggestive of past infections across the eukaryotic tree of life[8,9]. Furthermore, some mirusviruses have been maintained over long periods in protist cultures, either as circular episomes capable of producing infectious particles (persistent infections) or chromosomal integrants[10,11]. The virion morphogenesis module of mirusviruses that consists of the HK97-type major capsid protein (MCP) and several proteins involved in viral genome packaging is characteristic of the realm *Duplodnaviria*, with an apparent direct evolutionary relationship with animal-infecting herpesviruses and a more distant relationship with tailed viruses of bacteria and archaea[1]. In addition, the mirusvirus informational module genes encoding proteins involved in DNA replication

[1]Institut Pasteur, Université Paris Cité, CNRS UMR6047, Cell Biology and Virology of Archaea Unit, Paris, France. [2]Génomique Métabolique, Genoscope, Institut François Jacob, CEA, CNRS, Univ Evry, Université Paris-Saclay, Evry, France. [3]Research Federation for the Study of Global Ocean Systems Ecology and Evolution, FR2022/Tara GOsee, Paris, France. [4]Department of Biology, Institute of Microbiology and Swiss Institute of Bioinformatics, ETH Zürich, Zurich, Switzerland. [5]Bioinformatics Center, Institute for Chemical Research, Kyoto University, Uji, Japan. [6]Department of Biological Sciences, Center for Emerging, Zoonotic and Arthropod-Borne Pathogens, Virginia Tech, Blacksburg, VA, USA. [7]Computational Biology Branch, Division of Intramural Research, National Library of Medicine, National Institutes of Health, Bethesda, MD, USA. [8]These authors contributed equally: Sofia Medvedeva, Ulysse Guyet. [9]These authors jointly supervised this work: Eugene V. Koonin, Mart Krupovic, Tom O. Delmont. ✉e-mail: koonin@ncbi.nlm.nih.gov; mart.krupovic@pasteur.fr; tomodelmont@gmail.com

and transcription are homologous to informational genes of large and giant eukaryotic DNA viruses of the phylum *Nucleocytoviricota* (realm *Varidnaviria*)[1].

Protists play major roles in biogeochemical cycles and plankton ecology[12–14]. *Mirusviricota* and *Nucleocytoviricota* are the two phyla of double-stranded DNA viruses that actively infect protists by encoding a broad array of functions. Nucleocytoviruses have been extensively studied for decades using cultivation-based approaches[15,16], and thousands of metagenome-assembled genomes (MAGs) have been characterized worldwide[1,17–19]. Most nucleocytoviruses replicate in the host cytoplasm relying on the virus-encoded DNA replication and transcription machineries, with some lineages also going through an early nuclear replication phase[20–24]. Most mirusviruses characterized thus far (just over 100 genomes) encode the full sets of proteins required for DNA replication, deoxyribonucleotide synthesis and transcription, echoing the core *Nucleocytoviricota* functions required for cytoplasmic replication[1]. Thus, with mirusviruses and nucleocytoviruses predicted to mainly replicate in the cytoplasm, the nucleus of unicellular eukaryotes currently appears to be a largely vacant ecological niche for large DNA virus infections.

Here, we demonstrate the high prevalence of mirusviruses in aquatic ecosystems worldwide and characterize more than a thousand high-quality and non-redundant MAGs, some of which surpass 500 kb. We find that *Mirusviricota* is a highly diversified phylum of large and giant eukaryotic viruses containing three major putative orders and many additional deep-branching lineages. Two major putative orders include genomes rich in spliceosomal introns (a hallmark of eukaryotic genomes that are processed exclusively inside the nucleus[25]) and, mostly, lack genes required for cytoplasmic replication. The abundance of introns and the lack of replication and transcription machineries strongly suggest that these mirusviruses replicate in the host nucleus, completely relying on the host enzymatic apparatus.

## Results

### Mirusviruses are prevalent in aquatic ecosystems and beyond

Mirusviruses encode a single HK97-type MCP, which, owing to its relatively large size and structural conservation combined with sufficient divergence from the closest homologues in the realm *Duplodnaviria*[1], is an effective marker for detection of *Mirusviricota*. Here, we built a far-reaching hidden Markov model (HMM) (Methods) and detected 21,560 mirusvirus MCP sequences among 4,152 metagenomic assemblies of the mOTUs database[26,27] and 11 large *Tara* Oceans metagenomic co-assemblies[13] (Supplementary Table 1). Most MCPs were identified in marine (79.6%) and freshwater (14.4%) ecosystems, including surface and deeper layers of all oceans and seas, as well as lakes, thaw ponds and rivers across continents (for example, refs. 28–30) (Supplementary Table 1). Most of the remaining MCPs were identified in biofilms and sediments[31,32], with signal also present at the ocean bottom (hydrothermal plumes[33] and oceanic crust[34]), in continental groundwater[35], ice and streams of glaciers[36], as well as soil[37] and thawing permafrost[38]. Finally, although we detected some mirusvirus MCPs in shipworm, sponge and coral specimens (for example, skeleton samples of *Porites lutea*[39] and *Isopora palifera*[40]), this signal might come from co-occurring unicellular eukaryotes. Overall, this global survey of *Mirusviricota* MCPs dramatically increased the known diversity of mirusviruses, revealing their global prevalence that echoes the well-documented ecological prominence of nucleocytoviruses.

### *Mirusviricota* is a highly diverse phylum of large and giant viruses

We characterized mirusvirus genomes from a broad range of distantly related lineages using an iterative MCP-centric genome-resolved metagenomic approach (Methods; Supplementary Table 2 and Supplementary Information). We created a database of 1,257 high-quality, non-redundant mirusvirus genomes (average nucleotide identity

<98%, in line with previous reports[1,13,41]), with a mean size of 265 kb, mean completion of 88% and GC content ranging from 25% to 72% (Supplementary Table 3). Notably, 21 genomes were larger than 500 kb, the standard genome size threshold defining giant viruses, making *Mirusviricota* the second phylum of large and giant eukaryotic viruses. The genomic database includes four episomes[8,10], the only known chromosomal integrant[10], 50 previously characterized MAGs[1,2] and 1,202 newly characterized MAGs. We used metatranscriptomic data (Supplementary Table 4) to train the detection of introns and built a gene model for mirusviruses (Methods). We identified 295,521 genes including ~21,000 clusters of homologous genes and ~51,000 singletons. Only ~60,000 genes were associated with a Pfam[42] functional annotation. We used dedicated HMM models to annotate the virion morphogenesis module (MCP, terminase, portal, conserved jelly-roll protein, Triplex 1 and Triplex 2) (Supplementary Table 4). Altogether, the gene pool of mirusviruses is far less functionally annotated and is enriched in singletons compared with comprehensive gene sets for nucleocytoviruses (1,644 genomes) and herpesviruses (121 genomes) (Fig. 1 and Supplementary Table 5).

We explored the evolutionary relationships among mirusviruses by analysing individual phylogenies of the MCP, terminase and portal, along with the concatenation of these three core proteins (Fig. 1 and Extended Data Fig. 1). The phylogenies of these three markers were highly congruent (for example, Pearson's correlation coefficient of 0.83 between the MCP and terminase trees; Extended Data Fig. 1), indicative of the predominantly vertical transmission of virion morphogenesis genes throughout the evolution of mirusviruses. We delineated major taxonomic ranks by applying the relative evolutionary distance (RED) approach to the phylogenetic tree of the concatenated marker proteins (Methods), following a recent analysis of *Nucleocytoviricota*[43]. We identified 17 putative orders that encompass 62 putative families (at least two members each), most of these with strong phylogenetic support (Supplementary Table 3). The genome database also includes 102 putative families, each represented by a single genome, that substantially expand the evolutionary scope of the phylum *Mirusviricota*.

*Mirusviricota* includes three well-delineated major putative orders, all represented mainly in aquatic ecosystems (Fig. 1), which we provisionally named *Demutovirales* (12 families, 413 genomes), *Okeanovirales* (4 families, 281 genomes) and *Styxvirales* (5 families, 224 genomes). *Demutovirales* was already partially characterized in our previous study[1] (Supplementary Table 3). By contrast, only four MAGs were previously characterized for *Styxvirales*[2], whereas *Okeanovirales* was entirely overlooked. *Demutovirales* and *Okeanovirales* are only found in marine ecosystems, whereas *Styxvirales* is also prevalent in freshwater ecosystems. Most marine and freshwater metagenomic MCPs are affiliated with these three putative orders (Fig. 1 and Supplementary Table 1). In addition, all episomes and the chromosomal integrant together with several MAGs form another well-delineated, albeit much less prevalent, putative order, *Soporavirales* (3 families, 16 genomes), which might consist of viruses causing persistent infections given the distinct signal from protist cultures and microscopy data[11].

The remaining 323 MAGs in our database cover 13 putative orders that are poorly defined owing to genomic undersampling (hereafter referred to as cryptic). These cryptic putative orders occur in a wide range of ecosystems (Supplementary Table 1), are substantially enriched in singletons (43% on average; Fig. 1) and encompass most of the genomic diversity of mirusviruses. Furthermore, 22% of the detected metagenomic MCPs remained unassigned at the order level owing to their divergence from classified homologues in our genomic database (identity <50% at the amino acid level), exposing an additional level of undersampled *Mirusviricota* diversity, in this case, without closely related representatives with sequenced genomes (Fig. 1 and Supplementary Table 1). Overall, the emerging genomic diversity of mirusviruses contrasts that of nucleocytoviruses, which for the most part could be affiliated with six well-supported orders with very few

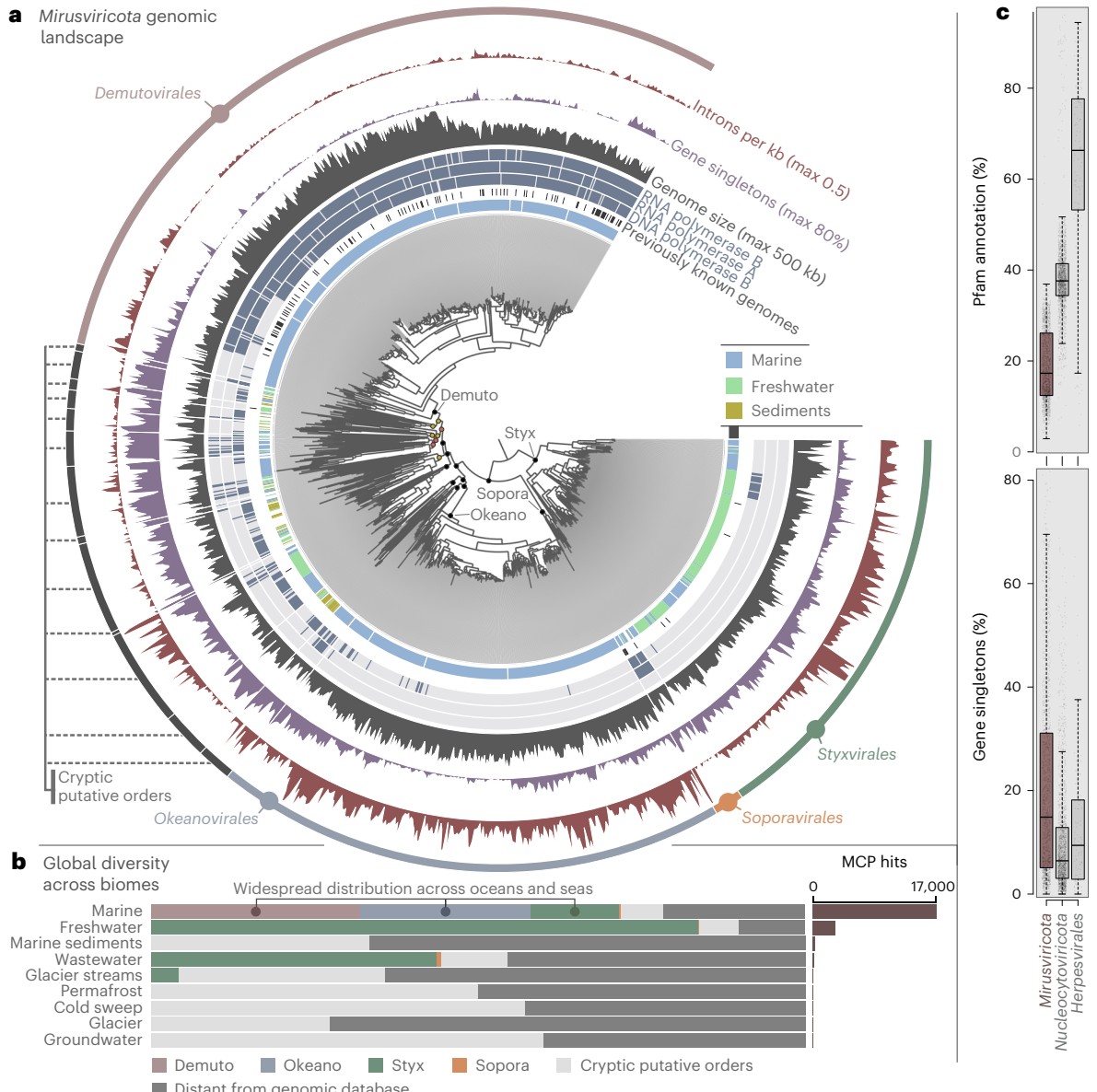

**Fig. 1 | Genomic landscape of the phylum *Mirusviricota*. a**, A maximum-likelihood phylogenomic tree of 1,204 *Mirusviricota* genomes based on the concatenation of manually curated alignments of MCP, terminase and portal proteins (1,871 amino acid positions). The tree was built using IQTree with the LG + F + R10 model and rooted between *Styxvirales* and the rest. Nodes were considered as strongly supported when SH-like aLRT was ≥80% and UFBoot was ≥95% (black dots), moderately supported when only one of the two cut-offs was met (yellow dots) and poorly supported when none of the two cut-offs were met (red dots). The tree was decorated with layers of complementary information and visualized with anvi'o. **b**, The number and order-level taxonomy of metagenomic MCPs in nine ecosystems. Metagenomic MCPs too distant from those in the genomic database (percentage identity <50% or bitscore <100 at the amino acid level) were not linked to any putative order ('distant from genomic database' category). **c**, Box plots summarizing the percentage of genes with a Pfam annotation and percentage of singleton genes for genomes affiliated with *Mirusviricota* (*n* = 1,257 genomes), *Nucleocytoviricota* (*n* = 1,644 genomes) and *Herpesvirales* (*n* = 121). Box plots correspond to the lower whisker, first quartile, median, third quartile and upper whisker.

single-genome putative families[43] and, as computed here, including substantially fewer singletons (Fig. 1, Extended Data Fig. 2 and Supplementary Table 5).

**Non-uniform spread of spliceosomal introns in mirusviruses**

We found that numerous mirusvirus genes contain spliceosomal introns (37,703 introns across 17,119 genes covering almost all genomes). An overwhelming majority of identified introns contained well-conserved canonical GT–AG splice sites predicted to be processed by the major spliceosome and displayed a length distribution peaking at ~80 bp, typical of spliceosomal introns in protists[13,25] (Extended Data Fig. 3 and Supplementary Table 4). Although uncommon, we also identified

a small fraction of introns with the AT–AC splice sites (enriched in one *Demutovirales* putative family), characteristic of the minor spliceosome (non-canonical splicing)[44]. We predicted different branching point consensus sequences for the introns of *Demutovirales* (WCTAAC, found in 10.9% of introns), *Okeanovirales* (CYSAC, 43.5%) and *Styxvirales* (CTGAC, 18.1%). In the most extreme cases, we identified up to 50 introns per gene and up to 271 introns per genome. Notably, the virion morphogenesis genes (2.7% of all genes) harbour 29% of the characterized mirusvirus introns, which corresponds to a 14-fold higher intron density (defined as the number of introns per kb) compared with the rest of the genes (Supplementary Table 4). This substantial enrichment of introns cannot be explained solely by metatranscriptomic

signal (Extended Data Fig. 3). The prevalence of introns varied considerably between mirusvirus families (Extended Data Fig. 3 and Supplementary Table 3), pointing to complex dynamics of intron proliferation. Intron density was on average low in *Demutovirales* (0.042 introns per kb), and much higher in *Okeanovirales* (0.2 introns per kb) and *Styxvirales* (0.18 introns per kb) (Extended Data Fig. 3 and Supplementary Table 4). Notably, *Soporavirales* displayed an intriguing trend, with high intron density in the episome (0.65 introns per kb) and chromosomal integrant (0.62 introns per kb) of one eukaryotic isolate[10] and very few introns in the other 14 genomes (0.025 introns per kb). The episomal mirusvirus genome has been shown to produce virus particles in the nucleus[11], confirming that the mirusvirus introns can be correctly processed. The density and spread of spliceosomal introns observed in mirusvirus genomes (up to 36% of genes with introns) contrasts the case of nucleocytoviruses that generally lack spliceosomal introns, with the notable exceptions of decaying endogenous viral elements, pandoraviruses and a small number of chloroviruses, both of which have an early nuclear replication phase[45–47]. The extent and uneven distribution of spliceosomal introns provide a strong indication that a substantial proportion of transcripts in *Okeanovirales* and *Styxvirales* (especially those for late gene transcription) are processed by the spliceosome in the host nucleus. In sharp contrast, in most genomes of *Demutovirales*, we did not detect any introns in the virion morphogenesis genes (Extended Data Fig. 4).

The variable localization of introns even in closely related mirusvirus genomes, as illustrated by the MCP genes (Extended Data Fig. 5), suggests dynamic gain and/or loss of introns. One potential mechanism for rapid intron gain involves intron-generating transposable elements, known as introners[48,49]. Regardless of their exact transposition mechanism, candidate introners can be identified as groups of introns with high sequence similarity. An all-against-all comparison of mirusvirus introns identified 69 of them as candidate introners (Methods; Supplementary Table 4). These introners were primarily detected in the genes encoding the MCP (*n* = 21) and terminase (*n* = 15) across *Demutovirales*, *Okeanovirales* and *Styxvirales* genomes, particularly those with extremely high intron densities (Supplementary Table 3). Without available host genomes, which are probably the primary source of introns, the introner identification was limited to the relatively small and fast-evolving viral genomes. Thus, although only a handful of mirusvirus introns could be recognized as introners, it appears likely that other introns in the core genes of *Okeanovirales* and *Styxvirales* also proliferated by transposition.

### Introns encoding homing endonucleases and trans-splicing in mirusviruses

Strikingly, whereas cellular spliceosomal introns do not typically carry genes, we identified >2,000 protein-coding genes within mirusvirus spliceosomal introns. Although most of the encoded proteins lack functional annotation (Supplementary Table 4), 334 genes located inside introns (hereafter intron-harboured genes) encode divergent endonucleases of the HNH (*n* = 171), GIY–YIG (*n* = 132), VSR-like (*n* = 16) and PD-(D/E)XK (*n* = 15) superfamilies (Extended Data Fig. 6). These four endonuclease types are often associated with self-splicing group I and group II introns (group II introns occur widely in phages[50]), as well as inteins, and promote the spread of these elements by cleaving homologous intron and intein-free sites, a process known as homing[51,52]. However, association of any of these endonucleases with spliceosomal introns has not been observed until now. The mirusvirus introns encoding homing endonucleases lack recognizable signatures of group I or II introns, namely, complementary interactions between 5′-terminal and 3′-terminal regions that result in distinct, stable secondary structures. Instead, mirusvirus introns are flanked by typical spliceosomal donor and acceptor sites. Thus, mirusvirus genomes contain a previously unrecognized type of intron, which we propose to name spliceosomal homing introns (shintrons; Extended Data Fig. 6 and

Supplementary Table 4). Most of the nested endonucleases occur in *Okeanovirales* genomes. Remarkably, shintrons are for the most part inside the virion morphogenesis genes (64.1% of all shintrons) and appear to have target-gene specificity, such that the MCP genes are targeted by shintrons encoding HNH endonucleases (*P* value <1 × 10⁻¹⁶, chi-squared test), whereas the jelly-roll protein gene is invaded by shintrons encoding GIY–YIG endonucleases (*P* value <1 × 10⁻¹⁶, chi-squared test) (Fig. 2, Extended Data Fig. 7 and Supplementary Table 4). This observation suggests that the corresponding homing endonucleases evolved to specifically recognize and cleave some of the key mirusvirus genes. In the case of co-infection with intron-containing and intronless mirusviruses, such cleavage is likely to promote the shintron spread to intron-free copies of the corresponding genes through homologous recombination.

We detected 271 intron-harboured genes that encode a family of proteins of unknown function, with nearly identical predicted three-dimensional (3D) structures (Extended Data Fig. 7). Although these proteins lack conserved residues that could constitute an active site of a nuclease, the corresponding introns appear to have a strong target gene preference, similar to shintrons (*P* value <1 × 10⁻¹⁶, chi-squared test). Specifically, these introns were detected exclusively within the terminase genes of *Okeanovirales*, *Styxvirales* and in one intron-rich family from a cryptic putative order (Fig. 2, Extended Data Fig. 7 and Supplementary Table 4). With no detectable homologues outside of *Mirusviricota*, we refer to this gene family as the 'Mirusvirus Intron Gene 1' (MING-1) family. An outstanding feature of MING-1 introns is their positioning in the exact same insertion site within the nuclease-encoding region of the terminase gene (Fig. 2) despite the considerable evolutionary divergence of the three mirusvirus orders in which MING-1 was found. MING-1 protein could function as an RNA chaperone facilitating folding of the intron, as described for proteins encoded by self-splicing introns[53–55], and targeting it to the unique site in the terminase gene, although the mechanism of such targeting remains obscure. The apparent long-lasting inheritance of a conserved gene-carrying intron indicates that some mirusvirus introns are highly stable through long evolutionary spans.

We found that most mirusvirus terminase genes outside of *Demutovirales* and *Soporavirales* are trans-spliced (a hallmark shared with alloherpesviruses[56–58]), with the ATPase and nuclease domains encoded in different parts of the genome and, occasionally, on opposite DNA strands (Fig. 2 and Extended Data Fig. 7). Trans-splicing, which has only been hypothesized to occur in alloherpesviruses on the basis of genome analysis, was confirmed in mirusviruses by metatranscriptome analysis, which showed that the two messenger RNA fragments are joined at the RNA level, before translation, using canonical splicing sites (Fig. 2 and Extended Data Fig. 8). The trans-splicing site location varied among the mirusvirus clades, with some perfectly matching those in alloherpesvirus clades (Extended Data Fig. 8). It seems intriguing that a highly conserved mirusvirus gene, responsible for one of the key functions occurring very late in the viral reproduction cycle (packaging of DNA into the preformed capsids), displays such staggering complexity of transcript processing including canonical introns, long-term persistence of MING-1 introns and trans-splicing.

### Potential nuclear reproduction mode of intron-rich mirusvirus families

Functional annotation was available for a much lower fraction of genes of *Okeanovirales* (average of 13% of genes per genome with a Pfam annotation) and *Styxvirales* (14%) compared with *Demutovirales* (27%), *Soporavirales* (23%) and, particularly, *Nucleocytoviricota* (38%) (Supplementary Figs. 2 and 4 and Supplementary Table 3). Apart from the virion morphogenetic module, we identified several additional core functions represented in most mirusvirus genomes (Supplementary Tables 3 and 4). These include TATA-binding proteins (transcription factors), heliorhodopsins (light-sensitive receptor

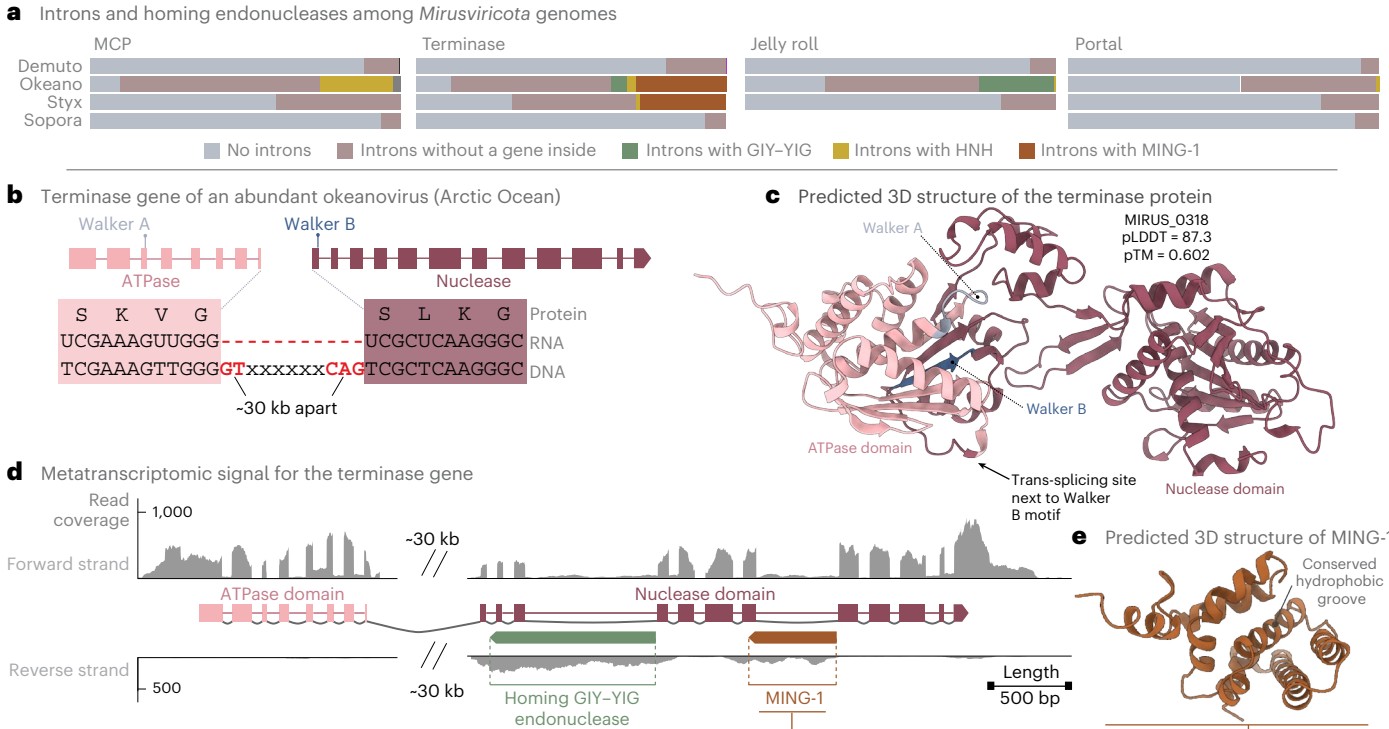

**Fig. 2 | Spliceosomal introns and intron-encoded homing endonucleases in mirusviruses. a**, The prevalence of spliceosomal introns and the intron-encoded homing endonucleases in four hallmark morphogenesis genes, across genomes from the major *Mirusviricota* putative orders. **b**, Structural features of the terminase gene from the genome 'Mirus_G_0318' of the putative order *Okeanovirales* that is highly abundant in parts of the Arctic Ocean. The terminase gene of this mirusvirus contains 20 spliceosomal introns and apparently undergoes trans-splicing of ATPase and nuclease domains that are encoded ~30 kb apart in the genome but form a contiguous transcript in the metatranscriptome. **c**, The 3D structure prediction of the terminase with the two terminase domains and the trans-splicing site. **d**, The metatranscriptomic signal (read recruitment for the forward and reverse strands) for the genome regions encoding the two trans-spliced domains of the terminase gene. The two genes (homing GIY–YIG endonuclease and MING-1) nested inside introns of the nuclease domain of the terminase are also shown. **e**, The predicted structure of MING-1 from **d**.

proteins), C3HC4 RING-type zinc-finger proteins (E3 component of ubiquitin ligases), histones, Snf2-family ATPases (chromatin remodelling) and Snf7 proteins (protein trafficking). The Snf7 proteins have not been identified in nucleocytoviruses, whereas heliorhodopsins and histones occur in only a few nucleocytovirus lineages[59,60], indicating that these genes represent distinct core functionalities of *Mirusviricota*.

The small number of *Mirusviricota* core functions is partly explained by the notable lack of genes involved in replication and transcription in *Okeanovirales* and *Styxvirales* (Fig. 3, Extended Data Fig. 9 and Supplementary Tables 3 and 4). Specifically, these intron-rich major putative orders lack most if not all genes responsible for the synthesis of deoxyribonucleotides from ribonucleotides (ribonucleoside diphosphate reductase subunits, glutaredoxin, dUTP diphosphatase, thymidylate synthase, thymidine kinase and dihydrofolate reductase) (Fig. 3). Ribonucleoside diphosphate reductase is a key enzyme for the cytoplasmic synthesis of deoxyribonucleotides in eukaryotic cells[61], and most nucleocytoviruses encode their own ribonucleoside diphosphate reductases (Supplementary Table 5). Also lacking in the intron-rich mirusviruses are genes responsible for DNA replication (family B DNA polymerase, DNA topoisomerase II, proliferating cell nuclear antigen and Holliday junction resolvase) and transcription (DNA-dependent RNA polymerase subunits A and B and transcription elongation factor TFIIS) (Fig. 3). The absence of these functions, including DNA polymerase of any known family, was validated by alternative functional annotations (Methods; Supplementary Table 4) and confirmed by 3D structure predictions for all proteins identified in a complete *Styxvirales* genome and a near-complete *Okeanovirales* genome (Extended Data Fig. 10 and Supplementary Table 6). This reliance on the eukaryotic host enzymes

for DNA replication is unique to *Mirusviricota* among known viruses with genomes exceeding 140 kb (ref. 62). By sharp contrast, functions for the synthesis of deoxyribonucleotides, DNA replication and transcription are conserved in most demutoviruses and nucleocytoviruses (Fig. 3 and Supplementary Tables 3 and 4). Our results strongly suggest that okeanoviruses and styxviruses, similar to herpesviruses, complete all stages of their reproduction cycle in the nucleus. These mirusviruses appear to be even more radically adapted to nuclear reproduction than herpesviruses in that they lack the DNA polymerase and other replication enzymes and are dramatically more enriched in spliceosomal introns. Conversely, the presence of these functions in most demutoviruses, together with the depletion of introns in their virion morphogenesis module genes (Extended Data Fig. 4), suggests that some of the most prevalent mirusviruses replicate partly or entirely in the cytoplasm, despite the occurrence of a few spliceosomal introns.

### 'Steal and escape' versus 'evolutionary trap' models

Insights from our genomic survey suggest cytoplasmic replication for *Demutovirales* and nuclear replication for *Okeanovirales* and *Styxvirales* (Fig. 3). Notably, *Demutovirales* families present a gradient of intron densities, with several intron-rich lineages nested among intron-poor ones, representing potential evolutionary intermediates in the transition between cytoplasmic and nuclear replication (Fig. 4). Several relatively intron-rich families of demutoviruses encompass the genes for the transcription machinery but lack most of those for DNA precursor synthesis and DNA replication. Two opposite models can be envisioned to explain the overall trends in *Mirusviricota* evolution: the cytoplasm-to-nucleus transition ('evolutionary trap' model) and

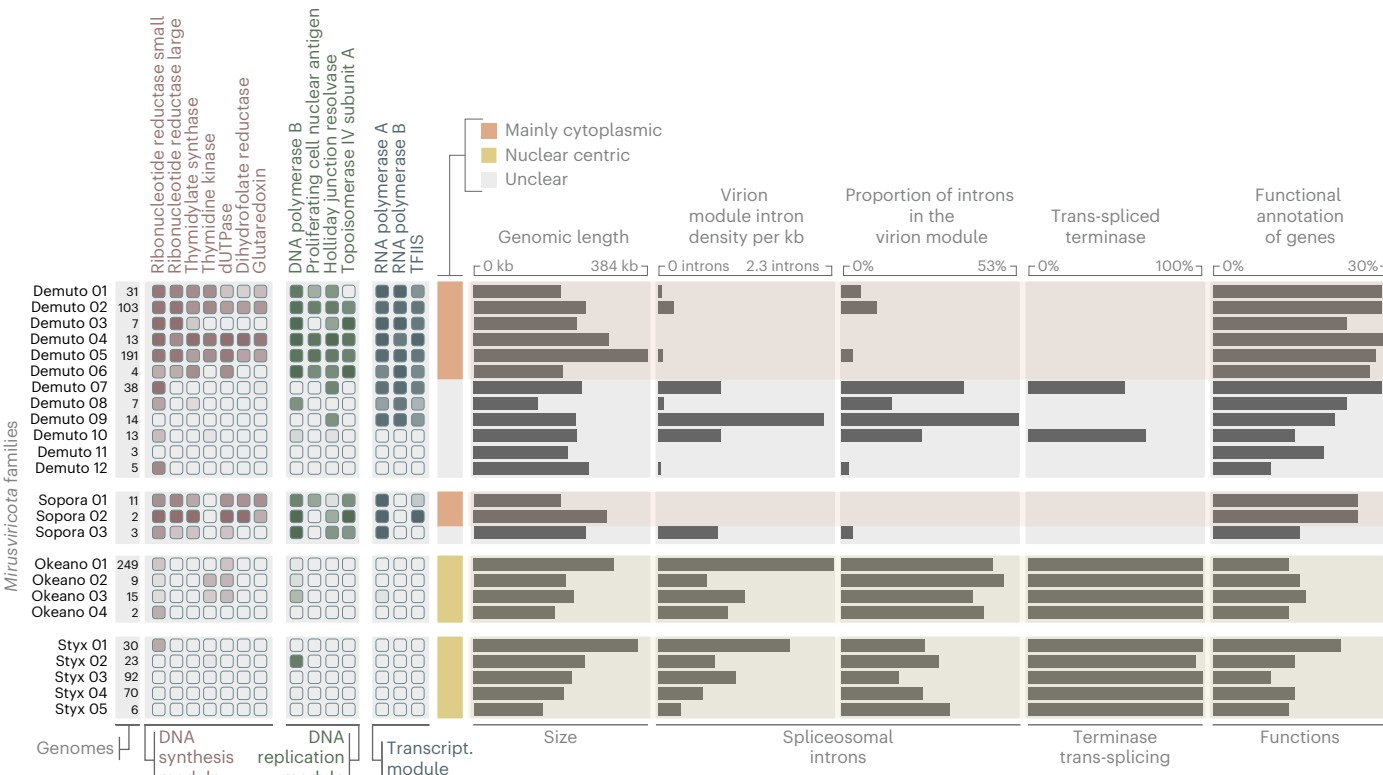

**Fig. 3 | Niche partitioning of *Mirusviricota* between the cytoplasm and the nucleus.** The figure summarizes average genomic trends (functions, spliceosomal introns and terminase trans-splicing) across putative families of *Demutovirales*, *Okeanovirales*, *Styxvirales* and *Soporavirales*. For each family, the fraction of genomes containing each gene of the different functional modules (DNA synthesis, DNA replication and transcription) is presented as a heatmap, with the colour intensity reflecting the corresponding fraction.

the nucleus-to-cytoplasm transition ('steal and escape' model) (Fig. 4). Under the 'evolutionary trap' model, the *Mirusviricota* ancestor replicated at least in part in the cytoplasm, which was followed by a single or multiple transitions to a nucleus-centric lifestyle, accompanied by the loss of genes required for autonomous cytoplasmic replication and transcription, and massive gain of spliceosomal introns. Conversely, in the 'steal and escape' model, the ancestral *Mirusviricota* lineage started out in the nucleus but upon acquisition of the DNA replication and transcription machineries, and purge of introns, transitioned to the cytoplasm. The 'evolutionary trap' model has the critical advantage of not requiring the intron purge, a rare and arguably unlikely event enabled by reverse transcription[63]. Generally, acquisition of spliceosomal introns appears to be a ratchet-type phenomenon, trapping a virus in the host nucleus for an early phase of replication if introns are only inserted into early viral genes—as it appears to be the case in most intron-poor demutoviruses and pandoraviruses—or a complete nuclear cycle phenomenon, if late genes encoding morphogenetic proteins contain introns as in okeanoviruses, styxviruses as well as some lineages of demutoviruses.

## Discussion

With genomes reaching more than 500 kb in size, we hereby introduce *Mirusviricota* as the second phylum of large and giant eukaryotic viruses. Mirusviruses are highly diversified and their genomes substantially expand the evolutionary scope and functional complexity of giant viruses. Apart from various deep-branching cryptic putative orders, *Mirusviricota* includes three major putative orders, which comprise most of the currently available genomes: *Demutovirales*, *Okeanovirales* and *Styxvirales*. Demutoviruses are prevalent in marine ecosystems, contain few spliceosomal introns and possess key informational module genes shared with most nucleocytoviruses and

required for replication in the cytoplasm. Despite forming entirely different virions[1], demutoviruses and nucleocytoviruses appear to occupy at least partially the same replicative niche, the host cytoplasm. By contrast, okeanoviruses (marine viruses) and styxviruses (marine and freshwater viruses) encompass many spliceosomal introns; lack the genes required for deoxyribonucleotide biosynthesis, DNA replication and transcription; and, in all likelihood, cannot reproduce in the cytoplasm. Thus, although at least partial virion uncoating in the cytoplasm is plausible, our results suggest that early gene transcription takes place in the nucleus. The most notable function lacking in most okeanoviruses and styxviruses is the viral DNA polymerase, which until now had been considered indispensable for the replication of large viral genomes, in particular, nucleocytoviruses and herpesviruses[62]. Although much uncertainty remains regarding the host range of mirusviruses, our survey and other studies[1–7] all indicate that they predominantly infect unicellular eukaryotes, which are abundant in aquatic ecosystems.

With their apparent reliance on the host nuclear machinery and the unprecedented enrichment of spliceosomal introns in virion morphogenesis genes, okeanoviruses and styxviruses are pivotal to our understanding of the ecology and evolution of large and giant eukaryotic viruses. Their hallmark features, such as predicted nucleus-centric replication and terminase gene trans-splicing, reinforce the direct evolutionary connection between mirusviruses and animal-infecting alloherpesviruses[1]. The diversity and distribution of spliceosomal introns in mirusvirus genomes reveal the complex, previously underappreciated dynamics of introns in eukaryotic viruses. The spliceosomal homing introns are particularly notable. Their prevalence in distantly related mirusvirus genomes, specific targeting of essential virion morphogenesis genes and apparent lack of equivalents in the genomes of eukaryotes and other viruses suggest

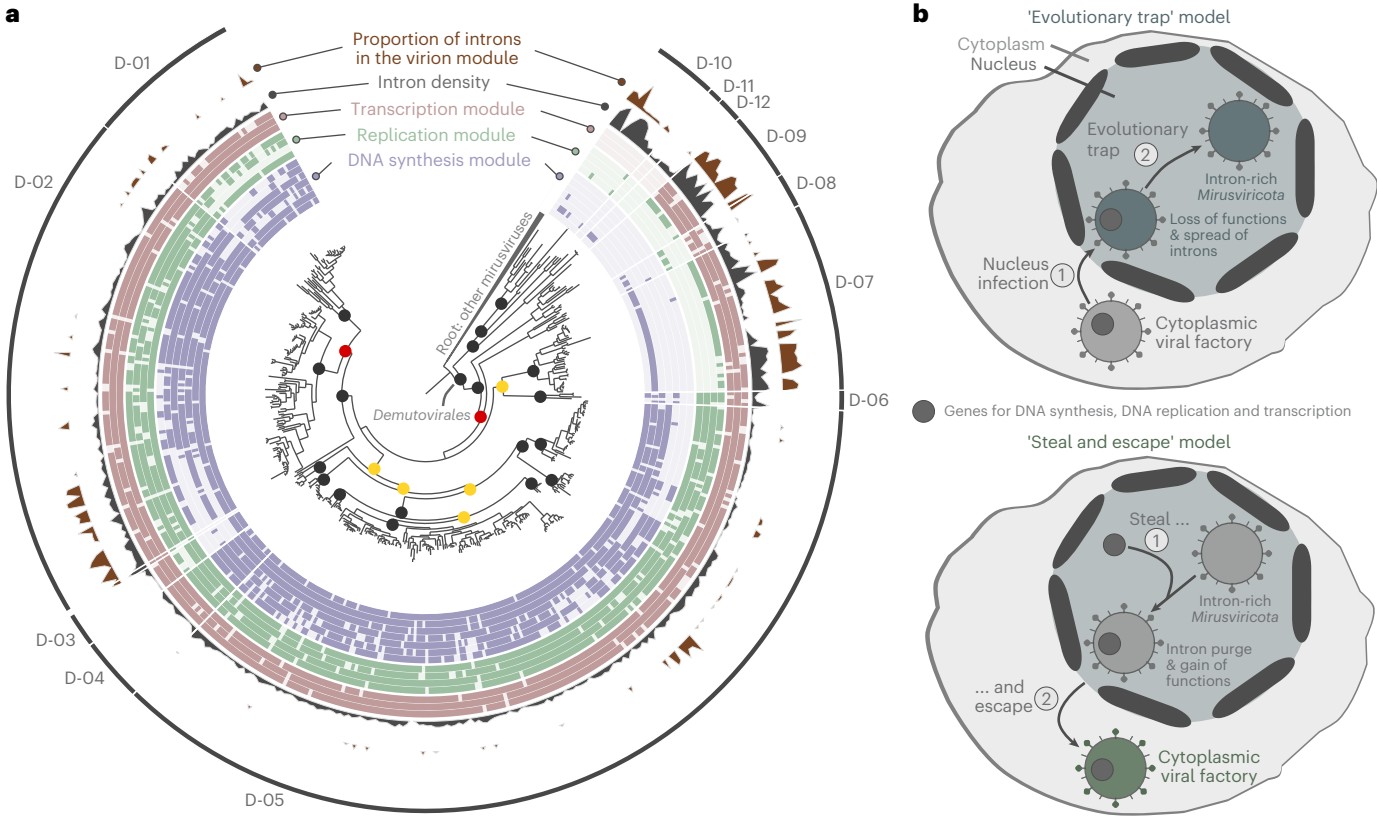

**Fig. 4 | 'Evolutionary trap' and 'steal and escape' models of evolution for *Mirusviricota*. a**, Apparent transitory states between a cytoplasmic replication and a nucleus replication lifestyle among *Demutovirales* families. The phylogenomic tree is the same as in Fig. 1 but rooted between *Demutovirales* and the rest of *Mirusviricota*. Nodes were considered as strongly supported when SH-like aLRT was ≥80% and UFBoot was ≥95% (black dots), moderately supported when only one of the two cut-offs was met (yellow dots) and poorly supported when none of the two cut-offs were met (red dots). The tree was decorated with layers of complementary information and visualized with anvi'o. **b**, The two main models of evolution for *Mirusviricota* ('evolutionary trap' versus 'steal and escape') in the context of the *Demutovirales* transitory states.

that such introns originated in an ancient nucleus-centric mirusvirus. In the case of endogenous nucleocytoviruses, introns are considered a sign of viral genome decay, endogenization and assimilation into the eukaryotic genome, consistent with the lack of transcription of the viral genes[47]. This is in stark contrast with mirusviruses, where genomes are transcriptionally active, with high levels of expression and processing of the intron-rich genes. It has been suggested that viruses can serve as vectors for the spread of introners in microbial populations[64,65], and intron-rich mirusviruses might play a major role in this process.

The *Demutovirales* families provide a unique snapshot of an evolutionary transition from the host cytoplasm to the nucleus as the major replicative niche. This transition, under our evolutionary trap model, is accompanied by the spread of spliceosomal introns and gradual loss of key functions, ultimately forming an effective lock system preventing the formation of cytoplasmic viral factories and reversion to cytoplasmic replication. The fact that intron-rich mirusviruses are among the most abundant and active DNA viruses in the sunlit oceans suggests that transition from the cytoplasm to the nucleus was associated with considerable reproductive success. Overall, our results strongly suggest that mirusviruses fill a major subcellular virus reproduction niche among protists, the host cell nucleus, previously thought to be only sparsely occupied by giant viruses. In the future, establishing relevant laboratory cultures will be key for testing predictions made in our study and for performing mechanistic studies of the homing endonucleases and enigmatic MING-1 proteins encoded by spliceosomal introns of mirusviruses.

## Methods

### Metagenomic survey of the mirusvirus MCP
We used a previously designed HMM[1] to identify mirusvirus MCPs among 11 large *Tara* Oceans metagenomic co-assemblies[13]. This step allowed the characterization of mirusvirus clades distant from those initially characterized[1]. We iteratively improved the HMM by incorporating newly identified protein sequences. The final HMM iteration was used to screen for mirusvirus MCPs in those co-assemblies as well as in a wide range of metagenomic assemblies from the mOTUs database[26,27] (e-value cut-off of $10 \times 10^{-5}$).

### An initial mirusvirus genomic database using manual binning
We performed a genome-resolved metagenomic survey on the basis of the MCP signal across *Tara* Oceans metagenomic co-assemblies, using anvi'o v8 (refs. 66,67) and manual binning principles already broadly applied to planktonic lineages[1,13,41,68]. We characterized and manually curated 115 MAGs corresponding to *Mirusviricota* clades entirely overlooked in the initial environmental genomic survey of the same dataset (mainly from *Okeanovirales* and *Styxvirales*). In addition, we also recovered 114 mirusvirus MAGs corresponding to single mOTUs contigs >100 kb containing the mirusvirus MCP. We subsequently combined all the available mirusvirus genomes (including those characterized in the first study[1]), defined a preliminary set of major clades using a phylogeny of their MCPs from within anvi'o and built a clade-aware reference database of mirusvirus proteins (Supplementary Table 2). *Demutovirales*, *Okeanovirales* and *Styxvirales* orders were first identified using this database, and preliminary sets of single copy core genes

(sccg; a gene was considered an sccg if it appeared in one copy for at least 50% of MAGs within the order) were characterized as described previously[1] to estimate genomic completion and redundancy.

## A global genomic resource for high-ranking taxonomic assignment of contigs

We combined the reference database of mirusvirus proteins with that of MAGs manually characterized and curated from the sunlit ocean and corresponding to bacterial, archaeal, eukaryotic and plastid populations[13,41,69] as well as *Nucleocytoviricota*[1,70]. We turned this taxonomy-aware protein database into a DIAMOND protein alignment database (diamond v2.1.8 (ref. 71)), which we used for high-ranking taxonomic annotation of contigs (see next section).

## An iterative automated binning workflow for *Mirusviricota*

Contigs in the metagenomic assemblies processed in our MCP survey were binned using sequence composition and differential coverage information as part of the mOTUs database[26,27] (Supplementary Table 2). We processed all 2,659 metagenomic bins containing at least one mirusvirus MCP with anvi'o v8. Contigs smaller than 2,500 nt were excluded, and proteins in the remaining contigs (2,109,138 contigs for a total of 5.9 Gb in length) were predicted using prodigal v2.6 (ref. 72). We performed two iterations of protein alignment using diamond v2.18 ('--ultrasensitive' option and percentage identity of at least 30%) against the global genomic resource (see previous section) to expand the scope of detection for *Mirusviricota* contigs. In the first iteration, contigs were assigned to *Mirusviricota* if at least 25% of the corresponding proteins had the best hit for a reference mirusvirus protein and if this percentage was above that of any other high-rank taxonomic category (Bacteria, Archaea, Eukarya, plastids or *Nucleocytoviricota*). In addition, mirusvirus contigs were assigned to a clade when at least 50% of their proteins had a best match for the same one. For each bin, contigs assigned to the same mirusvirus clade were assigned to a MAG ID. We excluded MAGs <50 kb or >650 kb in length, as well as *Demutovirales*, *Okeanovirales* and *Styxvirales* MAGs with a quality score (completion minus redundancy) below 50%. Proteins from the newly identified mirusvirus MAGs (n = 1,993) were integrated into the global DIAMOND protein alignment database, which we used for a second iteration of the 2,109,138 contigs using the exact same strategy, allowing improvement of previously characterized MAGs as well as the characterization of 149 additional MAGs. Note that metagenomic assemblies from the SPIRE database[73] were also screened using the same two iterations (see Supplementary Information for additional details).

## Recovery of highly distant mirusvirus genomes

The two iterations of automatic binning (see previous section) effectively characterized MAGs sharing enough protein sequence similarities with that of the first set of mirusvirus genomes integrated into the global genomic resource. Yet, several MCP-containing contigs could still not be linked to *Mirusviricota*, probably owing to their considerable evolutionary divergence compared with those in our database. A third automatic binning iteration was performed specifically to fill this critical gap, this time in two steps. In the first step, still within the scope of MCP-containing bins, we collected contigs with a high likelihood of being part of *Mirusviricota* using the following strategy: we (1) excluded bins >1 Mbp, (2) excluded bins displaying a good level of DIAMOND hits (>25% of proteins), (3) excluded contigs <10 kb in remaining bins (4) and finally excluded contigs with >10% of proteins having a best hit for any of the high-rank taxonomic categories of Bacteria, Archaea and plastids. The main rational was that long contigs in bins with a mirusvirus MCP and displaying very low levels of DIAMOND hits would most probably correspond to *Mirusviricota*. Remaining contigs in each of the considered bins were labelled as part of a putative mirusvirus MAG, if their cumulative length reached at least 50 kb. In the second step, all the mirusvirus MAGs characterized after iteration 2 (including the putative ones) were integrated into the global DIAMOND protein alignment database, which we used for a third iteration solely focused on expanding the scope of the putative MAGs. We used the same cut-offs as for the previous iterations and retained a total of 285 mirusvirus putative MAGs >50 kb in length. Various hallmark genes of *Mirusviricota* (for example, terminase, portal and heliorhodopsins in addition to the MCP) supported the biological relevance of these distant mirusvirus genomes, which fall within the scope of cryptic putative orders and for many corresponded to single-genome putative families.

## Quality score of *Mirusviricota* genomes

We merged mirusvirus genomes from the literature with those we characterized manually and automatically here. We used Prodigal v2.6 and Orthofinder v2.5.5 (ref. 74) to generate protein clusters. For each mirusvirus clade with at least ten genomes (n = 17; using a phylogeny of the MCP as guidance), we generated a collection of sccg corresponding to genes occurring as a single copy in at least 50% of the genomes. A similar approach was already applied to previously characterized mirusvirus lineages[3]. Note that for a given protein cluster, multiple occurrences within the same contig were counted as a single occurrence to account for intron-driven gene fragmentations. We used HMMer v3.4 (ref. 75) to generate an HMM for each sccg and excluded HMMs with an average number of hits across the corresponding genomes outside a range of 0.7–1.5 or those that did not provide a single hit per genome across at least 70% of the genomes. As for the protein clusters, for a given HMM, multiple hits within the same contig were counted as a single hit. The remaining 942 HMMs were used to assess the completion and redundancy of genomes affiliated with the most represented mirusvirus families, providing a much-needed quality metric (especially in the context of automatic binning) to refine the overall quality of our database (Supplementary Table 2).

## A final non-redundant genomic database for *Mirusviricota*

We merged mirusvirus genomes from the literature with those we characterized manually and automatically here, after excluding MAGs with a quality score <50% or with redundancy >25%. We subsequently built a non-redundant database using fastANI v1.34 (ref. 76) (average nucleotide identity <98%, minimum 25% genomic alignment) of 1,257 mirusvirus genomes by retaining the genome with the highest quality score and, if not available, simply the largest genome. Out of the final 1,257 genomes, 933 have a quality score. This database was used for all the following analyses (intron-aware gene calling, phylogenomics, taxonomic framework and functional annotations).

## Gene model for *Mirusviricota*

Gene predictions were made and refined with intron-aware gene model. Metatranscriptomic reads from the entire *Tara* Oceans project were mapped onto the final database of mirusvirus MAGs using hisat2 (ref. 77) (v2.2.1; parameters: --pen-noncansplice 1 --max-intronlen 5000). Low-complexity genomic regions were soft-masked using dustmasker[78] with default parameters (v1.0.0; package: blast 2.16.0). The first gene model was built separately for high-GC genomes (>0.55 GC content) and low-GC genomes with BRAKER2 (ref. 79) using mapped metatranscriptomic data (v2.1.6; --eptmmode --min_contig=900 --gc_probability=0.1 --downsampling_lambda=0 --max_intron=5000 --UTR=off). For the metatranscriptomic data, we used a default threshold for the minimal read coverage to support the presence of an intron (minimum of ten reads). Despite capturing the overall intron structure, the first model struggled with long introns (>300 bp), confirmed by metatranscriptomic data. Long introns were enriched in conserved genes (terminase, portal, MCP and jelly roll) and sometimes contained homing endonucleases. We manually identified problematic introns in the top 4 highly transcribed genes (list available in Supplementary Table 4) and soft-masked similar sequences of potential homing endonucleases in all genomes using tblastn. The second gene model was built

by BRAKER2 with addition of refined protein sequences from the four most highly transcribed genomes as a proteome (--prot_seq) and using soft-masked low-complexity and homing endonuclease regions. In parallel, genes were predicted without introns using Prodigal v2.6.3. Two gene sets (independently predicted by Prodigal and BRAKER2) were combined, favouring the BRAKER2 prediction when available. The resulting gene set contains 198,879 genes predicted by BRAKER2 and 96,642 genes predicted by Prodigal. During merging of the BRAKER2 and Prodigal results, two special cases were addressed. First, the BRAKER2 algorithm sometimes incorrectly fused two or more independent genes by connecting them with introns. In this case, we checked that all exon sequences correspond to independent open reading frames (ORFs) (all exons are >300 bp in length and each exon overlaps by >95% with a distinct ORF predicted by Prodigal) and favoured Prodigal prediction with multiple genes (9,851 genes were corrected when addressing this issue). Second, we observed that BRAKER2 could sometimes incorrectly predict two exons in an intronless gene, with one exon being very short. If the length of the short exon was less than 5% of the length of second exon in the gene, and if Prodigal predicted an ORF in the same region overlapping by >95% with the BRAKER2 prediction (8,913 occurrences), we selected the Prodigal gene prediction. Finally, the trans-splicing of terminase genes was detected by manual curation of genomic sequences, manual analysis of metatranscritomic data and tblastn searches for homologues of the C-terminal region of the terminase encompassing the Walker B motif of the ATPase domain and the nuclease domain in *Okeanovirales* and *Styxvirales* genomes.

### Introners

We extracted all intron sequences with the 10 bp-long exon flanking regions, following a pipeline for the identification of introners in eukaryotic genomes[48]. An introner was defined as a group of introns (at least two introns) from the same mirusvirus genome, with a high sequence similarity within the group (blastn e-value $<1 \times 10^{-5}$). We excluded cases where the region of similarity extended beyond the intron into exon flanking sequences to filter out similar introns from paralogous genes. In addition, we compared the mirusvirus intron sequences with those in an available database of introners from eukaryotic genomes[48]. No hits were found (blastn e-value $<1 \times 10^{-5}$).

### Genes inside introns

For identification of genes inside introns ('intron-harboured' genes), the sequences of long introns (>300) were processed by Prodigal (v2.6.3; default parameters). Only genes longer than 100 amino acids were selected for subsequent analysis. Predicted genes were clustered by phammseqs[80] (v1.0.3; default parameters). Clusters were annotated using HHpred[81]. Predictions of protein 3D structure for representative sequences of the clusters were made using Alphafold 3 web server[82].

### Identification of the mirusvirus virion morphogenesis module

We used HMMs dedicated to the identification of the virion morphogenesis module of major *Demutovirales* families[1] and iteratively improved them using the Orthofinder protein clusters and HMMer v3.4. The improved HMMs successfully identified the virion morphogenesis module of a majority of mirusvirus genomes in our database.

### Phylogenies of the mirusvirus virion morphogenesis module

For each hallmark gene of the mirusvirus virion morphogenesis module, we excluded the unusually small genes and subsequently performed alignments at the amino acid level using MAFFT[83] v7.490 and the FFT-NS-i algorithm with default parameters. In each of the protein alignments, sites with >70% gaps were trimmed using trimAl[84] v1.4.1. Phylogenetic reconstructions (both for individual hallmark genes and concatenations) were performed using IQ-TREE[85] v1.6.12 with '-m MFP -safe -alrt 1000 -bb 1000' parameters. ModelFinder[86] was used to determine and select the best-fitting model, which in all cases was

the LG + F + R10 model. As a result, this model was used for all genes included in the concatenated phylogeny. Supports were computed from 1,000 replicates for the Shimodaira–Hasegawa (SH)-like approximate likelihood ratio test (aLRT)[87] and UFBoot[88]. Nodes were considered strongly supported when SH-like aLRT was ≥80% and UFBoot was ≥95%, moderately supported when only one of the two cut-offs was met and poorly supported when none of the two cut-offs were met. Anvi'o v.8 was used to visualize and root the phylogenetic trees.

### A taxonomic framework for *Mirusviricota*

RED values were calculated for each node of our concatenated phylogenomic tree (MCP, terminase and portal) by applying the 'get_reds' function of the castor R package[89] on 11 distinct tree rooting positions corresponding to all the major deep-branching positions of the tree. This multiple-rooting strategy was used because the root position in the phylogenetic tree of *Mirusviricota* is currently unknown owing to the lack of an appropriate outgroup. The average RED value from the 11 rooting positions was used to define nodes corresponding to putative orders (average RED score below 0.22) and putative families (average RED score below 0.65). We propose formal names for four well-delineated putative orders of *Mirusviricota*:

(1) The order *Demutovirales* because of its families depicting different timeframes of the transitional states between a cytoplasmic and nucleus-centric replication lifestyle (*Demuto*: Latin verb for 'change, alter, become different'). The first representatives were characterized using metagenomics by Gaia et al.[1].

(2) The order *Okeanovirales* because of its prevalence in marine ecosystems ('Okeanos' can refer to the oceans and seas in Greek mythology). The first representative was characterized here.

(3) The order *Styxvirales* owing to its prevalence in freshwater ecosystems (Styx: river of the underworld). The first freshwater and marine representatives were characterized using metagenomics by Zhang et al.[2] and here, respectively.

(4) The order *Soporavirales* because of the apparent dormant nature or else persistent infection capability of episomes characterized from distantly related unicellular eukaryotic lineages (*Soporatus*: Latin adjective for 'asleep, sleeping'). The first representatives were characterized from a eukaryotic culture by Collier et al.[10].

### Functional annotations and their differential occurrence across viral clades

We combined the final non-redundant mirusvirus genomic database with two databases corresponding to *Herpesvirales* and *Nucleocytoviricota* and extracted from the Virus-Host database[70]. We computed the merged viral database using anvi'o v8 and ran Pfam[90] annotations. This information was used to calculate the percentage of functional annotations across viral genomes. In addition, we used anvi'o functional enrichment programs[91] to help identify the most differentially occurring functions across viral clades. Finally, functional annotations were retrieved from the eggNOG database v5 (ref. 92) using the eggNOG-mapper v2 tool[93].

### Singleton genes across viral clades

We individually applied Orthofinder v2.5.5 on each high-ranking clade of the merged viral database (*Mirusviricota*, *Nucleocytoviricota* and *Herpesvirales*) with the '-M'msa'' option. Proteins not assigned to a cluster of at least two sequences were labelled as singletons. This information was used to calculate the percentage of singleton genes across viral genomes.

### Statistical analyses

Chi-squared tests were performed to test whether some virion morphogenesis genes (MCP, terminase, portal, jelly-roll, Triplex 1 and

Triplex 2) were targeted by specific homing endonucleases (HNH, GIY–YIG, VSR-like or PD-(D/E)XK). Those analyses include 214 pairs of virion morphogenesis genes–homing endonucleases that cover a total of 156 mirusvirus genomes. Contingency tables were constructed (details in Supplementary Table 4), and $P$ values were calculated using the chi-squared test in R (stats package).

## Reporting summary

Further information on research design is available in the Nature Portfolio Reporting Summary linked to this article.

## Data availability

All databases our study generated are available via figshare at https://doi.org/10.6084/m9.figshare.28955240 (ref. 94). Those include (1) MCP proteins identified in the global metagenomic survey, (2) the non-redundant database of 1,257 *Mirusviricota* genomes (both contigs and predicted genes in the form of nucleotide and protein coding sequences), (3) the intron-informed gene model for *Mirusviricota*, (4) HMMs for the *Mirusviricota* virion morphogenesis module (including the updated HMM for the MCP of mirusviruses), (5) the database of single-copy genomic MCPs used for the taxonomic annotations of metagenomic MCPs, (6) protein alignments and phylogenetic trees corresponding to the MCP, portal and terminase hallmark genes, (7) intron sequences found in the genomic database, (8) the global protein database (Bacteria, Archaea, Eukarya, plastids, *Nucleocytoviricota* and *Mirusviricota*) used for high-ranking taxonomic annotation of contigs and (9) our protein database for *Nucleocytoviricota* and *Herpesvirales*. Source data are provided with this paper.

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

## Acknowledgements

Our survey was made possible thanks to the large amount of metagenomic data that have been produced and made publicly available in recent years. As a result, we thank all the scientists involved in the metagenomic exploration of a wide range of environmental ecosystems. We also thank A. Fernandez-Guerra and C. Vani for their early analytical contributions using AGNOSTOS, L. Meng for early discussions regarding available viral genomic resources to contextualize the *Mirusviricota* genomic database and T. Antoine for their involvement in the making of the global genomic resource used here for the identification of mirusvirus contigs. This study was supported by two grants from the French National Research Agency (grant nos. ANR-23-CE02-0022 to T.O.D. and M.K. and ANR-23-CE02-0025 to M.G.) as well as the National Institutes of Health (NHI; grant no. 1R35GM147290 to F.O.A.). This article is contribution number 163 of Tara Oceans.

## Author contributions

T.O.D., M.K. and E.V.K. conducted the study. S.M., U.G. and T.O.D. performed most of the primary data analysis, with critical contributions by E.P. (metatranscriptomics), H.-J.R. (automatically generated bins containing mirusvirus MCPs), M.G. (phylogenetic comparisons) and M.K. (virion morphogenesis genes and introns). S.M., U.G., E.P., H.-J.R., S.S., H.O., F.O.A., M.G., N.Y., E.V.K., M.K. and T.O.D. all contributed to interpreting the data and writing the paper.

## Competing interests

The authors declare no competing interests.

## Additional information

**Extended data** is available for this paper at https://doi.org/10.1038/s41564-025-02190-6.

**Correspondence and requests for materials** should be addressed to Eugene V. Koonin, Mart Krupovic or Tom O. Delmont.

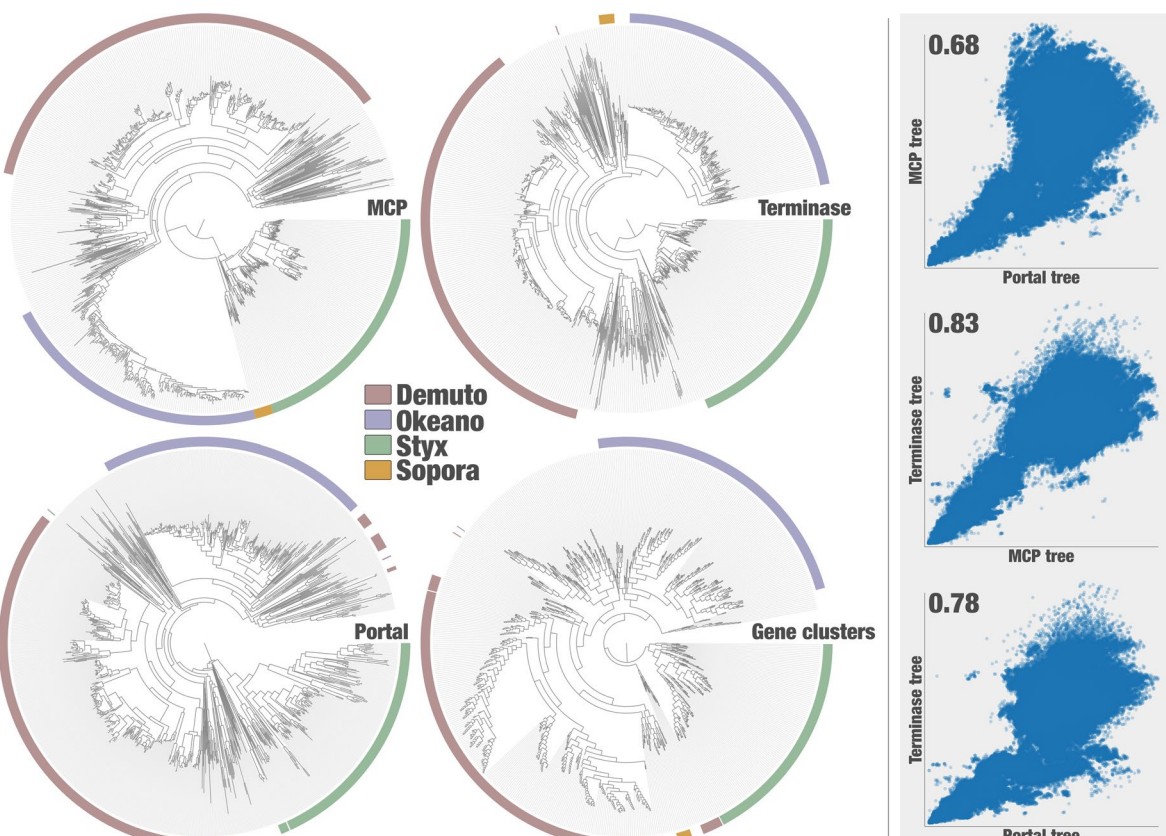

**Extended Data Fig. 1 | Clustering of Mirusviricota genomes based on individual gene phylogenies (MCP, portal, terminase) and the occurrence of protein clusters (Euclidean distance).** Panels display maximum-likelihood phylogenetic trees built using IQTree and based on curated alignments of MCP (1,084 sequences, alignment of 777 amino acids, LG + F + R10), terminase (1,040 sequences, alignment of 609 amino acids, LG + F + R10) and portal proteins (988 sequences, alignment of 543 amino acids, LG + F + R10). In addition, one panel displays the clustering of genomes based on the occurrence of protein clusters (Bray–Curtis dissimilarity and average linkage clustering methods).

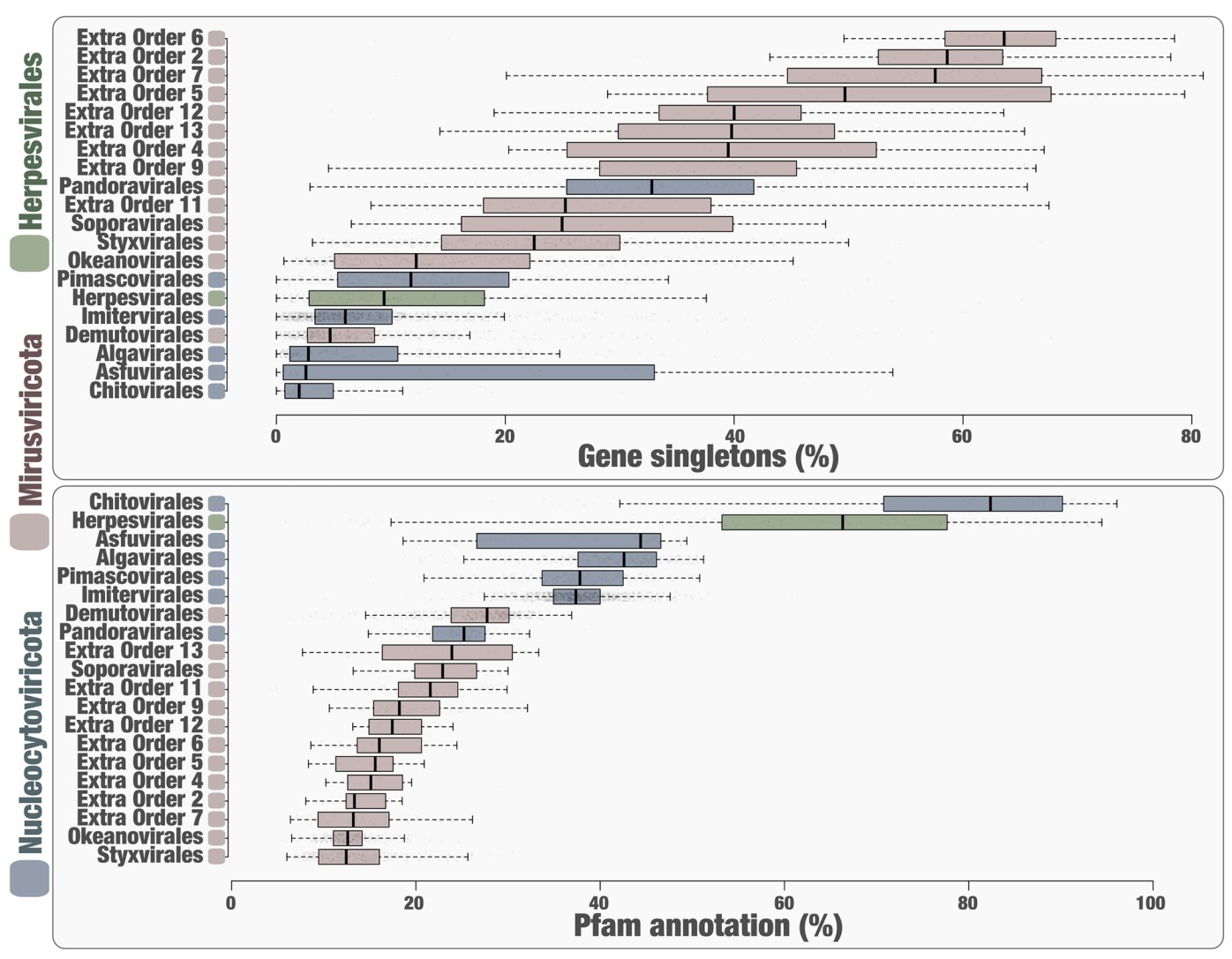

**Extended Data Fig. 2 | Singleton genes and functional annotations across eukaryotic virus orders.** Singletons genes were quantified using Orthofinder, and the functional annotations were quantified using Pfam.

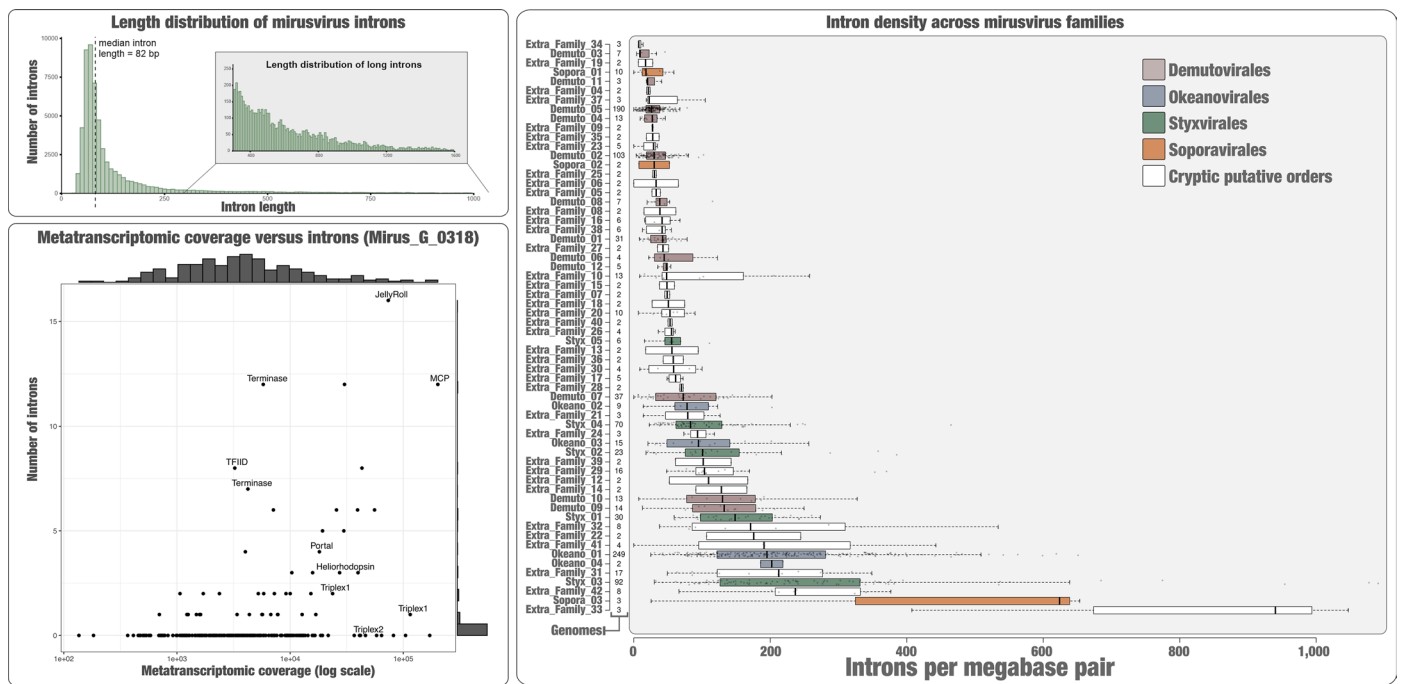

**Extended Data Fig. 3 | *Mirusviricota* introns.** Top left panel displays the length distribution of introns (n = 37703) detected in the 1,257 *Mirusviricota* genomes. Bottom left panel displays the metatranscriptomic coverage (*Tara* Oceans) and occurrence of introns among genes of a highly abundant marine mirusvirus.

Boxplots in the right panel display the intron density (number of introns per megabase pair) of genomes organized by putative family of *Mirusviricota*. All individual values are displayed.

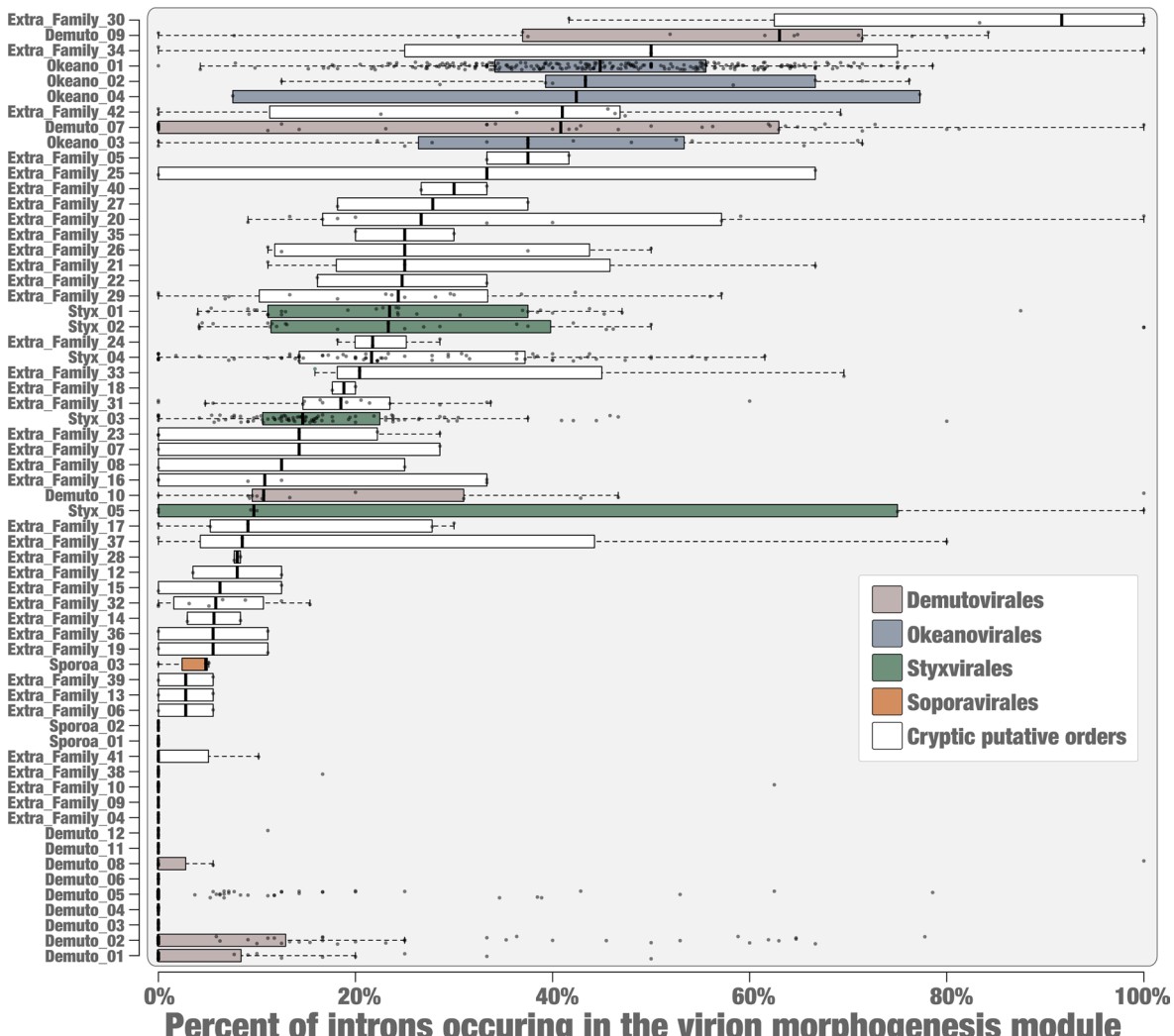

**Extended Data Fig. 4 | Enrichment of introns in the virion morphogenesis module.** The figure displays boxplots summarizing, across genomes of the same mirusvirus putative family, the percent of all introns occurring within genes of the virion morphogenesis module.

# MCP CDS of 9 *Okeanovirales*_01 representatives

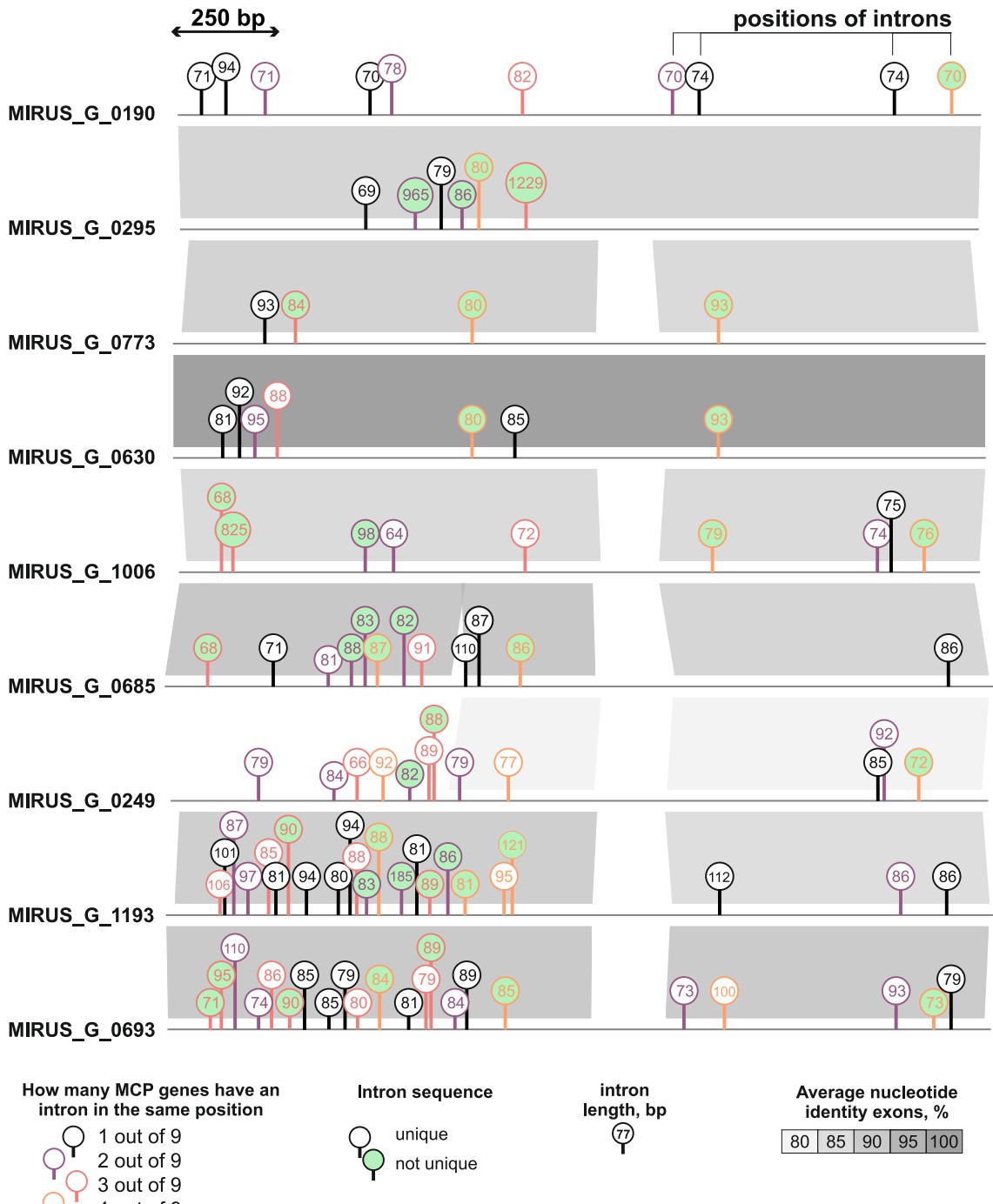

**Extended Data Fig. 5 | Intron dynamics of closely related MCPs.** The diversity of introns is shown for 9 representative mirusvirus genomes of the Okeanovirales_01 family. Coding DNA sequence (CDS) of the MCP genes were compared by blastn nucleotide identity between pairs of closely related genomes (similarity is shown by grey connections). The MCP CDS sequences are ordered based on the protein tree shown in Fig. 1. Intron positions on CDS are shown with 'pins'. Number inside the pin head corresponds to the length of the intron. The color of pin border shows how many MCP genes have an intron in the same position (black - unique position of the intron, orange - 4 out of 9

MCP CDS have an intron in this position). The fill of the pin head shows if the sequence of the intron is unique (white fill) or not (green fill). Several trends can be observed: 1) viruses from the same family (Okeanovirales_1) contain different number of introns in the MCP gene (ranging from 4 to 22 introns), despite high exon sequence similarity (up to 98%); 2) only 40% of introns are conserved by sequence (pins with a green fill); 3) 30% of introns have both unique sequence and unique position (pins with black border and white fill); 4) unique introns can occupy the same splicing site position in different MCP CDS (pins with colored border and white fill).

# Homing endonucleases in spliceosomal introns

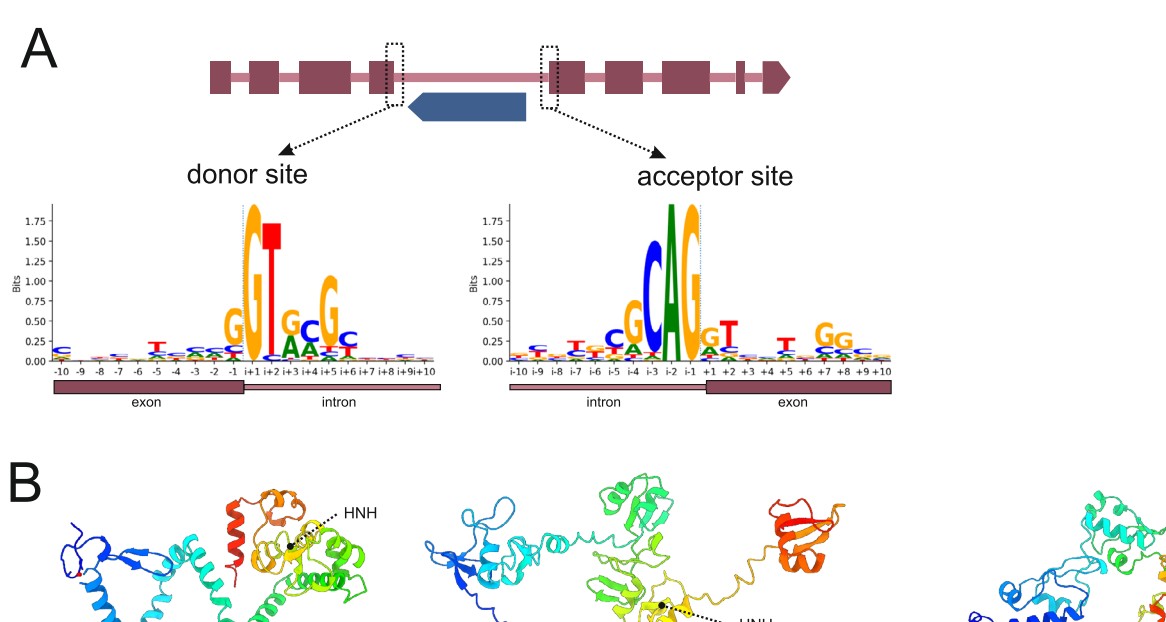

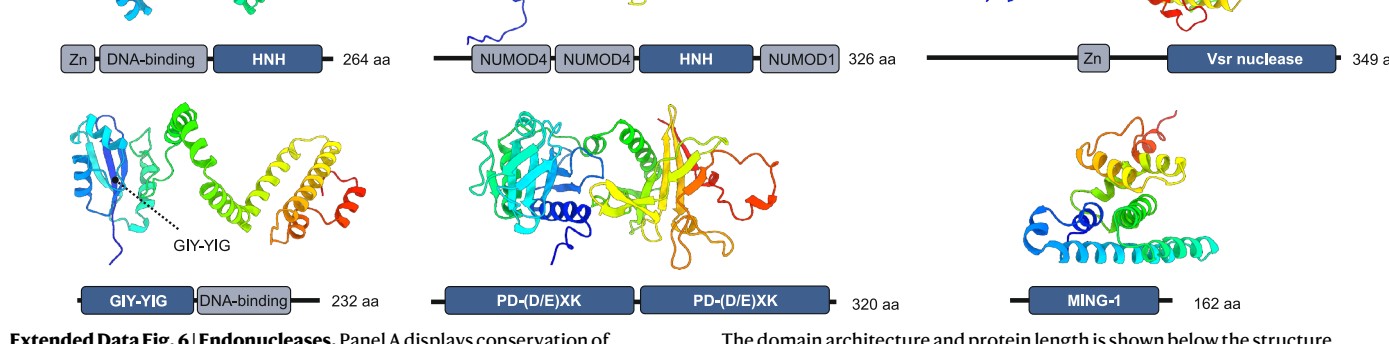

**Extended Data Fig. 6 | Endonucleases.** Panel A displays conservation of canonical GT-AG splice sites in mirusvirus introns. Panel B displays representative homing endonucleases found in spliceosomal introns in mirusvirus genomes. The domain architecture and protein length is shown below the structure prediction made by Alphafold2. Structures are colored by an aminoacid position (blue - N-terminus, red - C-terminus).

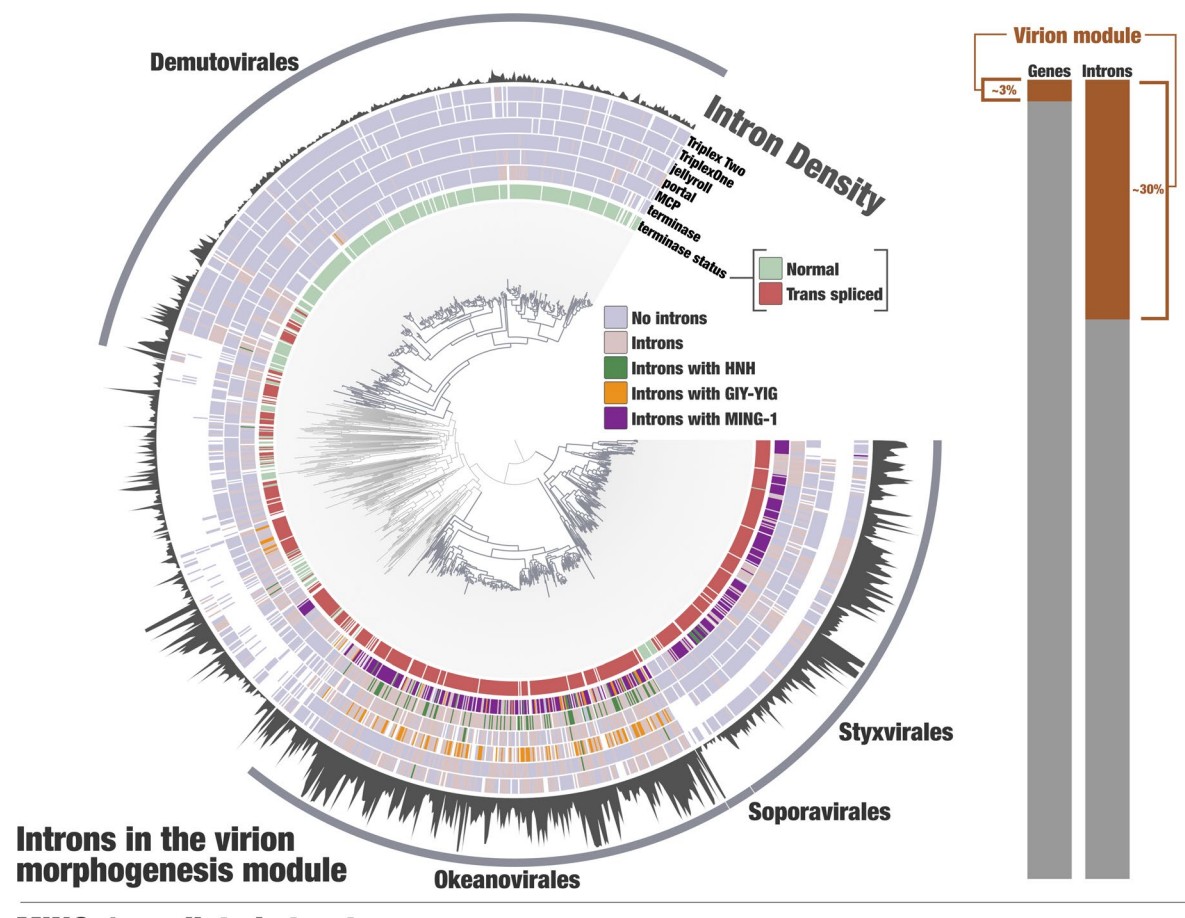

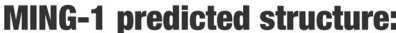

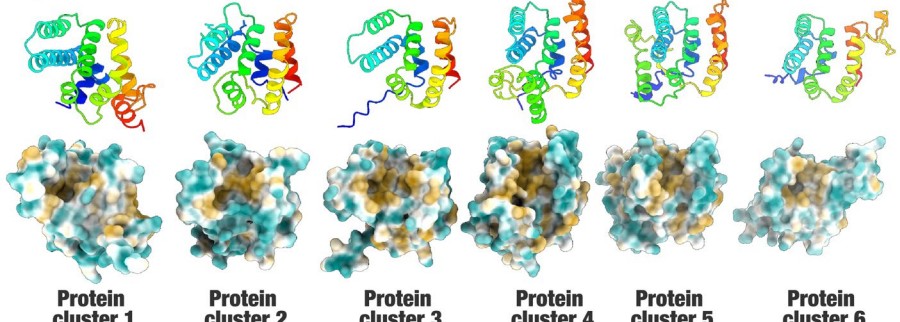

**Extended Data Fig. 7 | Introns in the virion morphogenesis module of**
***Mirusviricota.*** Inner tree in the top panel corresponds to the phylogenomic tree
of *Mirusviricota* displayed in Fig. 1. The occurrence of introns (and their nested
gene when present) is displayed for each gene of the virion morphogenesis
module. The proportion of genes corresponding to the virion morphogenesis
module, and the proportion of all introns occurring in this module are also
summarized. The bottom panel shows predicted structures of six MING-1
variants. In the first lane, the backbone structures are coloured by
aminoacid position (blue N-terminus – red C-terminus). In the second lane,
the same structures are shown as surfaces and coloured by hydrophobicity
(yellow – hydrophobic, blue – hydrophilic).

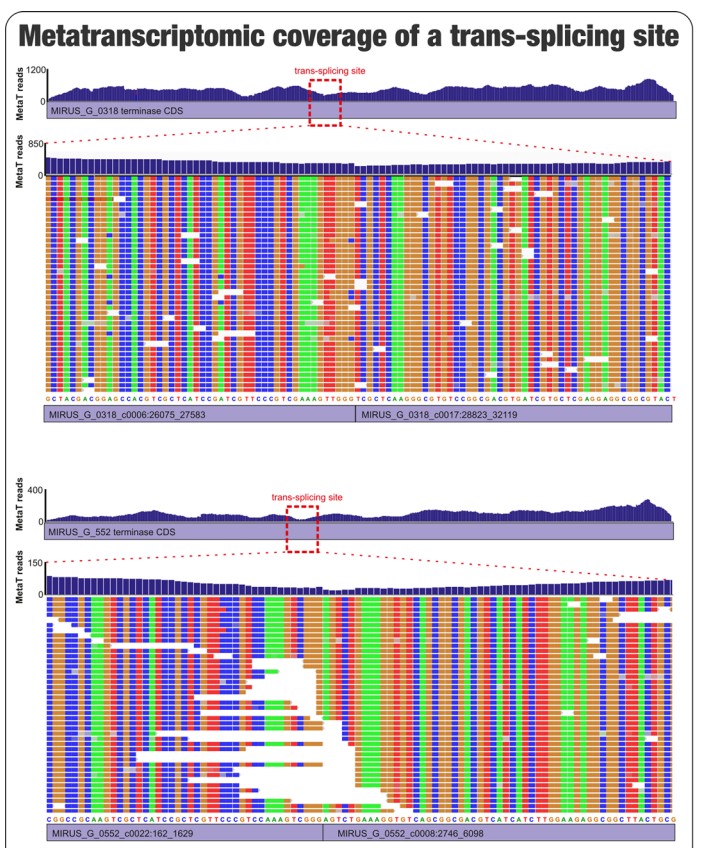

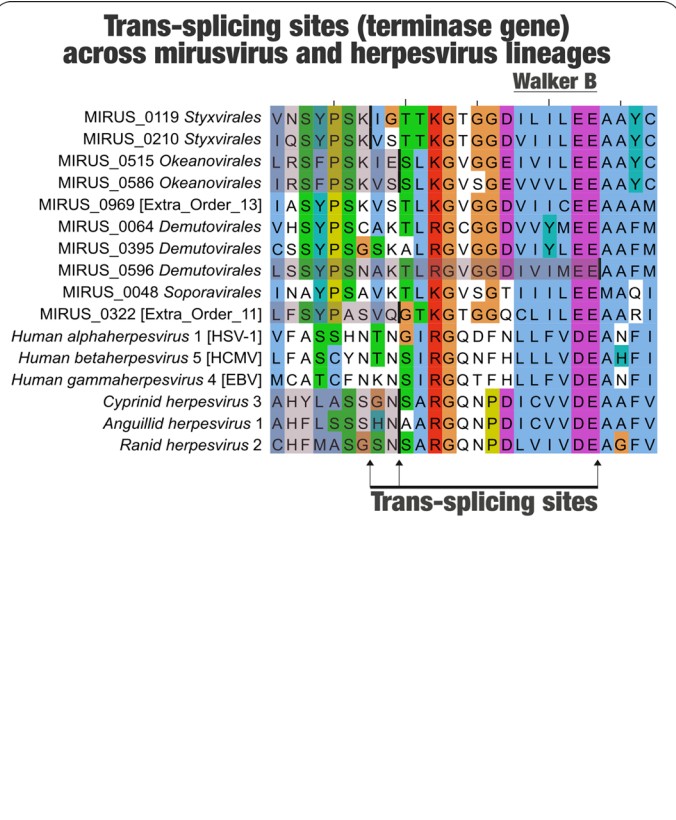

**Extended Data Fig. 8 | Metatranscriptomic signal and position of the trans-splicing site of the terminase.** In the left panel, two examples of highly-transcribed terminase genes (Mirus_G_0318 and Mirus_G_0552) are shown. Metatranscriptomic reads from *Tara* Oceans were mapped on the reconstructed CDS sequences of the terminase, where two parts (ATPase and nuclease domains) are connected together. The coverage does not show any abnormality in the trans-splicing site (red box). The zoom on the region of trans-splicing shows RNAseq reads, with different nucleotides represented by different colors

(A = green, T = red, G = orange, C = blue). The colors correspond to the nucleotide sequence below. In the right panel, the alignment of representative protein sequences of terminase subunit from mirusviruses, Orthoherpesviruses and Alloherpesviruses are shown. The alignment region includes ATPase domain of the terminase, near walker B motif (the position of Walker B is indicated above). The predicted trans-splicing sites are marked with thick black line and arrows. In case of trans-splicing, the shadowed part of the sequence corresponds to the the N-terminal half of the terminase (ATPase domain).

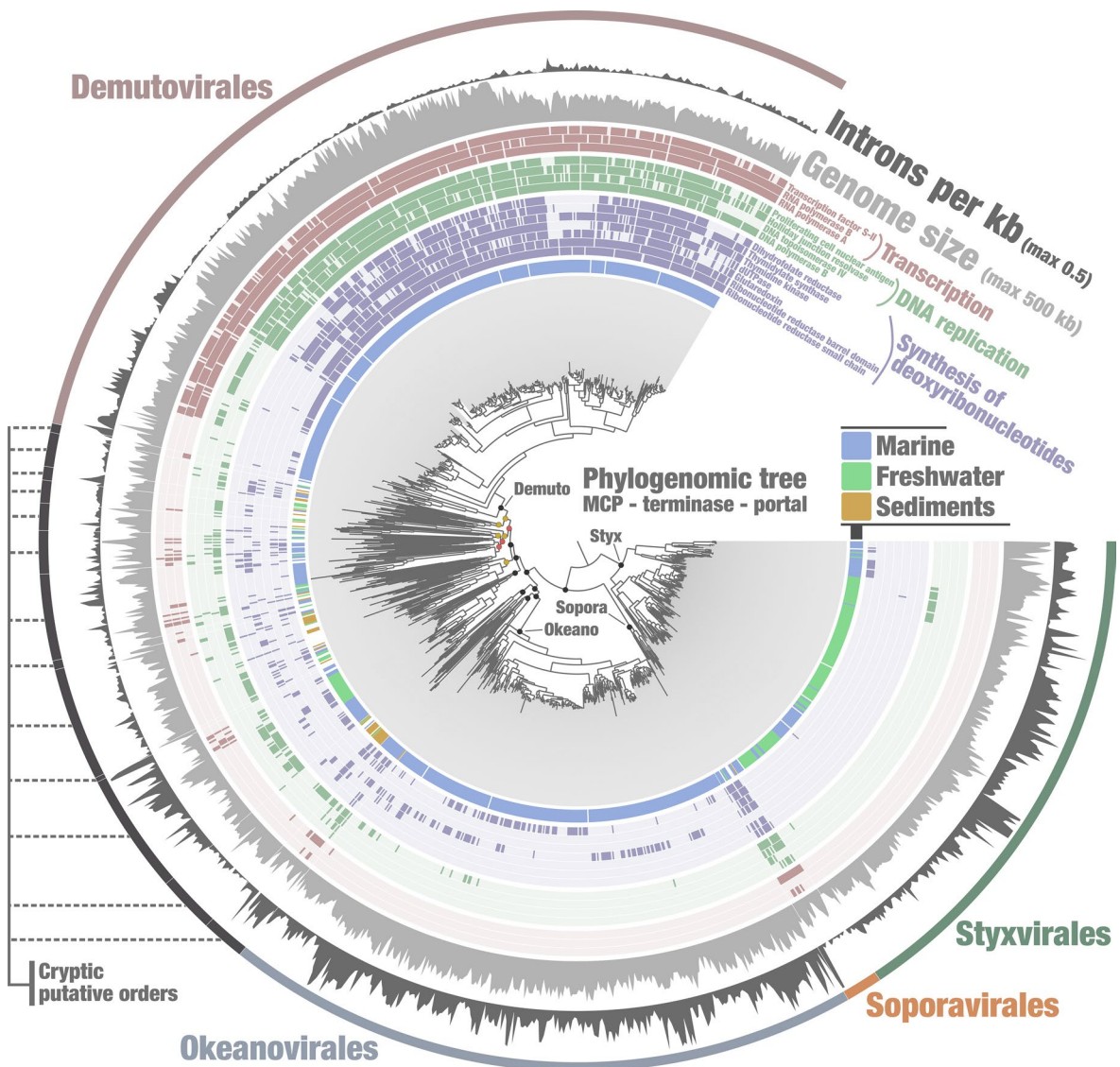

**Extended Data Fig. 9 | Distribution of key functions across mirusvirus genomes.** The figure displays a maximum-likelihood phylogenomic tree of 1,204 *Mirusviricota* genomes based on the concatenation of manually curated alignments of MCP, terminase and portal proteins (1,871 amino acid positions). The tree was built using IQTree2 with the LG + F + R10 model and rooted between *Styxvirales* and the rest. The tree was decorated with layers of complementary information and visualized with anvi'o.

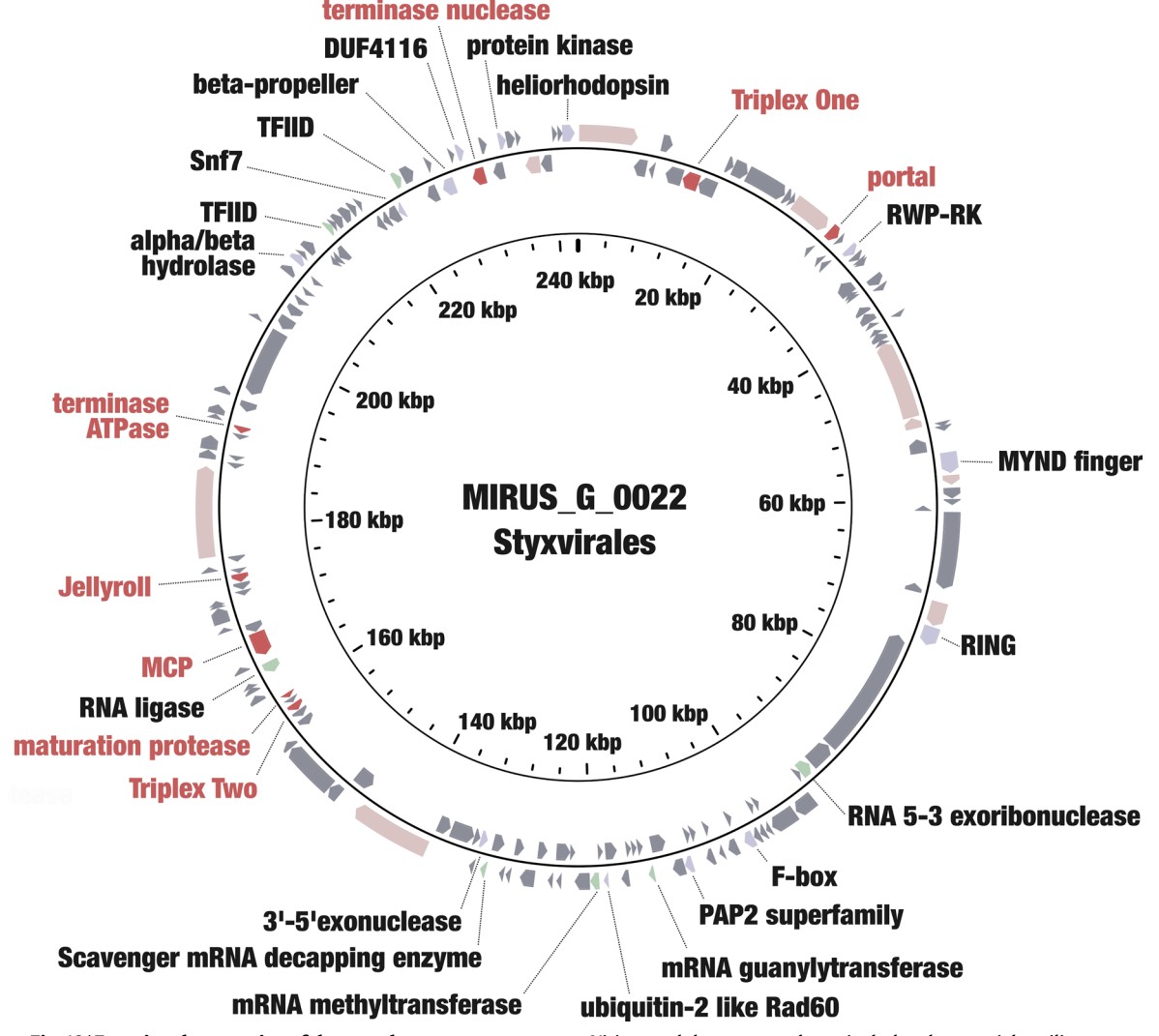

**Extended Data Fig. 10 | Functional annotation of the complete genome 'Mirus_G_0022' (Styxvirales putative order).** Key functions displayed in the figure were identified using Pfam annotations, HMMs for the virion morphogenesis module, and Alphafold2 predictions applied to all the proteins. Virion module genes are shown in dark red, potential auxiliary structural proteins in light red, transcription-related genes in light green, genes with only general functional annotation in light grey and genes without any functional annotation in dark grey.

# Reporting Summary

## Statistics

For all statistical analyses, confirm that the following items are present in the figure legend, table legend, main text, or Methods section.

| n/a | Confirmed | |
|---|---|---|
| ☐ | ☒ | The exact sample size (*n*) for each experimental group/condition, given as a discrete number and unit of measurement |
| ☐ | ☒ | A statement on whether measurements were taken from distinct samples or whether the same sample was measured repeatedly |
| ☐ | ☒ | The statistical test(s) used AND whether they are one- or two-sided<br>*Only common tests should be described solely by name; describe more complex techniques in the Methods section.* |
| ☒ | ☐ | A description of all covariates tested |
| ☒ | ☐ | A description of any assumptions or corrections, such as tests of normality and adjustment for multiple comparisons |
| ☒ | ☐ | A full description of the statistical parameters including central tendency (e.g. means) or other basic estimates (e.g. regression coefficient) AND variation (e.g. standard deviation) or associated estimates of uncertainty (e.g. confidence intervals) |
| ☐ | ☒ | For null hypothesis testing, the test statistic (e.g. *F*, *t*, *r*) with confidence intervals, effect sizes, degrees of freedom and *P* value noted<br>*Give P values as exact values whenever suitable.* |
| ☒ | ☐ | For Bayesian analysis, information on the choice of priors and Markov chain Monte Carlo settings |
| ☒ | ☐ | For hierarchical and complex designs, identification of the appropriate level for tests and full reporting of outcomes |
| ☐ | ☒ | Estimates of effect sizes (e.g. Cohen's *d*, Pearson's *r*), indicating how they were calculated |

*Our web collection on statistics for biologists contains articles on many of the points above.*

## Software and code

Policy information about availability of computer code

| Data collection | We collected publically available metagenomes. |
|---|---|
| Data analysis | anvi'o V.8 was used for data analysis. |

For manuscripts utilizing custom algorithms or software that are central to the research but not yet described in published literature, software must be made available to editors and reviewers. We strongly encourage code deposition in a community repository (e.g. GitHub). See the Nature Portfolio guidelines for submitting code & software for further information.

## Data

Policy information about availability of data

All manuscripts must include a data availability statement. This statement should provide the following information, where applicable:
- Accession codes, unique identifiers, or web links for publicly available datasets
- A description of any restrictions on data availability
- For clinical datasets or third party data, please ensure that the statement adheres to our policy

Data availability: All databases our study generated have been made publicly available at https://figshare.com/s/33ebaa0ef9edf2df1f83. Those include (1) MCP proteins identified in the global metagenomic survey, (2) the non-redundant database of 1,257 Mirusviricota genomes (both contigs and predicted genes in the form of nucleotide and protein coding sequences), (3) the intron-informed gene model for Mirusviricota, (4) HMMs for the Mirusviricota virion morphogenesis module, (5) the database of single-copy genomic MCPs used for the taxonomic annotations of metagenomic MCPs, (6) protein alignments and phylogenetic trees

corresponding to the MCP, portal and terminase hallmark genes, (7) intron sequences found in the genomic database, (8) the global protein database (Bacteria, Archaea, Eukarya, plastids, Nucleocytoviricota, Mirusviricota) used for high-ranking taxonomic annotation of contigs, (9) and our protein database for Nucleocytoviricota and Herpesvirales.

## Research involving human participants, their data, or biological material

Policy information about studies with <u>human participants or human data</u>. See also policy information about <u>sex, gender (identity/presentation), and sexual orientation</u> and <u>race, ethnicity and racism</u>.

| | |
|---|---|
| Reporting on sex and gender | No human participants |
| Reporting on race, ethnicity, or other socially relevant groupings | No human participants |
| Population characteristics | No human participants |
| Recruitment | No human participants |
| Ethics oversight | No human participants |

Note that full information on the approval of the study protocol must also be provided in the manuscript.

## Field-specific reporting

Please select the one below that is the best fit for your research. If you are not sure, read the appropriate sections before making your selection.

☐ Life sciences          ☐ Behavioural & social sciences          ☒ Ecological, evolutionary & environmental sciences

For a reference copy of the document with all sections, see nature.com/documents/nr-reporting-summary-flat.pdf

## Ecological, evolutionary & environmental sciences study design

All studies must disclose on these points even when the disclosure is negative.

| | |
|---|---|
| Study description | We solely explore metagenomic data. |
| Research sample | Our survey covers more than 100,000 metagenomic assemblies from a wide range of ecosystems. However, most of the signal for mirusviruses is aquatic. |
| Sampling strategy | No sampling was done |
| Data collection | No data collection. Only the use of available metagenomes. |
| Timing and spatial scale | No relevant |
| Data exclusions | No relevant |
| Reproducibility | No relevant |
| Randomization | No relevant |
| Blinding | No relevant |

Did the study involve field work?          ☐ Yes          ☒ No

## Reporting for specific materials, systems and methods

We require information from authors about some types of materials, experimental systems and methods used in many studies. Here, indicate whether each material, system or method listed is relevant to your study. If you are not sure if a list item applies to your research, read the appropriate section before selecting a response.

## Materials & experimental systems

| n/a | Involved in the study |
|-----|----------------------|
| ☒ ☐ | Antibodies |
| ☒ ☐ | Eukaryotic cell lines |
| ☒ ☐ | Palaeontology and archaeology |
| ☒ ☐ | Animals and other organisms |
| ☒ ☐ | Clinical data |
| ☒ ☐ | Dual use research of concern |
| ☒ ☐ | Plants |

## Methods

| n/a | Involved in the study |
|-----|----------------------|
| ☒ ☐ | ChIP-seq |
| ☒ ☐ | Flow cytometry |
| ☒ ☐ | MRI-based neuroimaging |

## Plants

| | |
|---|---|
| Seed stocks | No plants |
| Novel plant genotypes | No plants |
| Authentication | No plants |

