## [Peer Review File · Nature Microbiology]

Widespread, intron-rich mirusviruses are predicted to reproduce in nuclei of unicellular eukaryotes

Corresponding Author: Dr Tom O. Delmont

Version 0:

Reviewer comments:

Reviewer #1

(Remarks to the Author)

Medvedeva et al. performed a large-scale exploration of metagenomic data from diverse biomes with a focus on the newly discovered mirusviruses. By improving and utilizing an HMM, they detected more than 21,000 major capsid protein genes. They also assembled a database of ~1,300 mirusvirus MAGs with a high level of completeness, the vast majority of which are entirely novel. They use core gene phylogenies and RED analysis to propose 17 different mirusvirus orders and 62 (decently represented) families, and >100 additional potential families represented by MAG singletons. The diversity of mirusviruses uncovered in this research is truly remarkable and contrasts that seen in the nucleocytoviruses.

A particularly exciting discovery is the presence of introns in many but not all, mirusviral genes, including a non-random distribution of introns in morphogenesis genes (2.7% of all mirusvirus genes harbour almost 30% of the introns). These introns appear to be 'mobile', and some harbour homing endonuclease-encoding regions. Trans splicing has also been inferred. This all speaks to regulation. This is the first description of HEs associated with 'canonical' spliceosomal introns. The Introner connection is also fascinating, although unclear in terms of evolutionary significance.

Specific comments:

- What about the putative hosts of such viruses? The title explicitly nods to unicellular eukaryotes, but there is not a single mention of the eukaryotic lineages known or predicted to be hosts is mentioned anywhere in the text. I think this could be improved and make the paper even more interesting to non-specialists and useful to virologists, microbial ecologists, protistologists etc. who are still new to mirusviruses.

Intron analysis: did the authors search for, and identify, and 'AT-AC' introns spliced by the 'minor' spliceosome?

Introns and splicing: It's not clear from the main text or materials and methods how the metatranscriptomic data were used. For example, what were the thresholds involved for coverage that could be trusted to infer splicing / presence of an intron? Fig 2 shows histogram data for an obviously compelling case involving trans splicing, but that's the only bit of information I can see. And how was trans-splicing detected in the first place? Researchers will want to know.

Figure 3 – what do the different shades of color in the different columns indicate (e.g., dark 'red' to lighter gray/red in DNA synthesis module)? This should be mentioned in the figure legend.

Gene flux section – starting line 409. Very interesting observations, but leaves the reader unsatisfied. One line 417, the authors refer to 'apparent high level of gene flux'. Fig 3 shows between 0 and 10 genes across the mirusvirus families. Is this 'high'? What does provisional definition of 50% identity mean, really, in terms of LGT for highly divergent viruses with very patchy gene presence. Directionality of transfer? What kinds of functions involved? None of this is mentioned in the text (there is Figure S15 which shows four genes / proteins stated as being evidence of transfers, but the phylogenies are difficult to interpret from such a thousand-foot level).

Line 82 – "Thus, with mirusviruses and nucleocytoviruses mainly replicating in the cytoplasm, the nucleus of unicellular eukaryotes currently appears to be a largely vacant ecological niche for large DNA virus infections."

Perhaps a bit more caution here? "appearing to replicate in the cytoplasm"? I don't know that we know enough about mirusviruses, do we? Almost all of the data are bioinformatics predictions. To be fair, later text is suitably nuanced and compelling, but it struck me as a tad too strong at this point.

Line 90, line 137: large and giant viruses. "Analysis of this genomic database shows that Mirusviricota is a highly diversified phylum of large and giant eukaryotic viruses...". "Notably, 21 genomes were larger than 500 Kb, the standard genome size threshold defining giant viruses, making Mirusviricota the second phylum of large and giant eukaryotic viruses.

What is the value of distinguishing between large and 'giant'? It is arbitrary, as the authors know, but also has the potential to clash with taxonomic IDs. From work of co-author Koonin and colleagues, it's clear the viruses (virions and their genomes) have become 'giant' on multiple occasions. Moving forward, what if some members of the taxonomic entity Mirusviricota were found to have 'small' genomes? What would that mean? So the following: "making Mirusviricota the second phylum of large and giant eukaryotic viruses" (and also in the conclusion) doesn't really mean much (to me at least). All this is to say that I think the field of 'giant' virus research has moved to a point where it doesn't really matter how big their genomes are in terms of novelty and the more that is discovered the more we are likely to see a range of sizes.

This is a matter of taste and don't feel strongly about it, but I don't see the value of adding even more arbitrary designations of size (genomes or virus particles?). This is the source of regular confusion for non-specialist audiences and adding more categories now is just setting the field up for more headaches in the future.

Line 120: "However, we also detected mirusvirus MCPs inside shipworm, sponge and coral specimens (e.g., skeleton samples of *Porites lutea*³⁵ and *Isopora palifera*³⁶), "...". Is the implication here that mirusviruses infect animals? Or are the MCPs associated with mirusviruses in protist symbionts / parasites within animals? Probably should be careful at this stage not to provide substrate for misinterpretation?

Line 139: "The genomic database includes four circular episomes^{7,9}, the only known chromosomal integrant⁹ ,..."

Predicting circularity can be difficult, so maybe "obviously circular episomes"? This section of text leaves the reader wondering what the structure of the $1204 - 4 - 1 = 1199$ mirusvirus genomic elements are (inc. the 51 previously ID'd ones). It doesn't really come up again.

Line 238: "Notably, Soporavirales displayed an intriguing trend, with high intron density in the episome (653 introns/Mb) and chromosomal integrant (624 introns/Mb) of one eukaryotic isolate⁹ and very few introns in the other 14 genomes (25 introns/Mb)." If the authors are referring to only one episome and integrant, and those elements are only ~300 Kb, why refer to them in terms of introns / Mb? I realize this is the useful metric for looking at large MAG datasets, but it seems misleading in this case.

Line 252: "contained no introns". Contained no obvious introns?

Line 284: "The mirusvirus introns encoding homing endonucleases lack recognizable signatures of group I or II introns and instead are flanked by typical spliceosomal donor and acceptor sites". What signatures did the authors look for? I think much of the self-splicing intron signatures studied in the literature are secondary structure based. It would be good for the authors to articulate this briefly (either at this part of the text or elsewhere). Were the predicted introns folded (with the HE-encoding regions removed)? I doubt it, as this would have been a ton of work, but I recommend that the authors clarify the extent to which they dug into the different possibilities here. Are these introns group I/II introns in origin that have somehow adopted GT-AG splice boundaries and are now removed by the spliceosome? Or are they canonical spliceosomal introns that have recently acquired HE genes? This could well be questions for the future, but at least touching on it here would seem to be worth it. I can only assume that they are spliceosomal introns first, since most are too small or (in the case of Fig. 2B) too large to be self-splicing (at least based on what is known about GPI and II introns).

Line 289: "Remarkably, SHI are for the most part inside the virion morphogenesis genes (64.1% of all SHI), and display target gene specificity, ...". They are predicted to have target gene specificity, but this hasn't been demonstrated experimentally. This is made clear when talking about the HE function in the following sentence, but I'd suggest clarifying the language a bit here.

Line 318: similar to above, "the corresponding introns display strong target gene preference...". I don't understand exactly what "display" means here. Do you mean the intron insertion sites 'are consistent with' target site preference on the part of the HEs (or whatever proteins)? The introns themselves don't display target gene preference, that is a function of a trans-acting proteins that mediate homing, correct? For this comment and above, I think the language could be improved. I apologize if I have misunderstood.

Line 693: "'Intron-borne" genes". As written this could mean genes that emerged de novo from introns (which can happen), but this is not what the authors mean. They mean genes inside existing introns (and presumably have landed within the intron)?

Reviewer #2

(Remarks to the Author)

In this manuscript Medvedeva et al., leverage on Tara Ocean data and taxonomically demonstrate Mirusviricota comprises a highly diversified second phylum of large and giant eukaryotic viruses that rivals the evolutionary scope and functional complexity of viruses in the first characterized phylum, Nucleocytoviricota. Functionally they describe gene context difference including presence of hallmark genes required for replication in the cytoplasm as well some mirusviruses that included spliceosome introns linked to genes encoding essential proteins. They also propose two evolutionary scenarios. Overall, the manuscript is well structured and provided invaluable insights on the ever-growing NCLV.

Abstract, Line 22: Please remove references from the abstract, as these are typically discouraged in that section.

Line 134: The statement indicates that non-redundant mirusvirus genomes share an average nucleotide identity (ANI) of <98%.

Should this be >98% instead?

Lines 708–713: The methodology for constructing the concatenated gene tree is unclear. Please provide additional details regarding how the tree was built, particularly whether each gene partition was assigned a unique evolutionary model.

Line 719: RED scores are indeed used in GTDB taxonomy assignments. However, the rationale behind selecting a RED score threshold of 11 is not clear. Were any benchmarking efforts performed using ICTV reference sequences to justify this cutoff? Some reasoning or justification for this threshold is necessary.

Line 742: Functional annotation is based on Pfam, which is a reasonable approach. However, Pfam annotations can be ambiguous in certain cases. Have the authors validated the annotations using an additional database, such as eggNOG or KEGG?

Figure S15: How is lateral gene transfer (LGT) defined as recent in this figure? There appears to be no time calibration in the phylogenetic tree to support this statement.

Data Availability: A link to the updated HMM is not provided. The reference provided is to the initial HMM only. Please include access to the updated version.

Additional Question: The authors present 3D-predicted structures of the terminase protein. How do these predicted structures from the MAGs compare with known reference structures? Does a structure-based evolutionary analysis (structure based tree) supports the inferred functions.

While two evolutionary scenario's are proposed in a future direction paragraph under conclusion would be great to discuss how to mechanistically test.

Reviewer #3

(Remarks to the Author)

Summary

This manuscript presents a tour de force in environmental virology, dramatically expanding our knowledge of mirusvirus diversity through the characterisation of over 1,000 high-quality genomes. The authors make the compelling case that Mirusviricota represents a second phylum of large and giant eukaryotic viruses, rivaling Nucleocytoviricota in evolutionary scope and functional complexity. The discovery of widespread nuclear-replicating viruses that lack DNA polymerases and are enriched in spliceosomal introns challenges conventional understanding of large DNA virus biology. While the work is technically impressive and the major conclusions are well-supported, several aspects of the interpretation and presentation warrant further consideration.

Strengths

The scope and scale of this study are remarkable. By surveying over 100,000 metagenomic assemblies and developing iterative binning approaches, the authors have increased the known diversity of mirusviruses by an order of magnitude. The methodological framework, particularly the iterative refinement of genome bins and the development of clade-specific quality metrics, represents best practices in environmental genomics that will be of interest to the field.

The manuscript's most exciting contributions are its novel biological discoveries. The identification of spliceosomal homing introns represents the first documented case of homing endonucleases within spliceosomal introns, potentially revealing a new mechanism for intron mobility. Similarly, the evidence for trans-splicing in terminase genes, supported by metatranscriptomic data, provides rare documentation of this phenomenon in DNA viruses. Perhaps most striking is the discovery of giant viruses that apparently lack DNA polymerases yet maintain genomes exceeding 500 kb, fundamentally challenging our assumptions about the minimal requirements for large DNA virus replication.

The evolutionary framework presented is thoughtful and well-reasoned. The authors' proposal of "evolutionary trap" and "steal and escape" models for transitions between cytoplasmic and nuclear replication provides testable hypotheses for understanding mirusvirus evolution. The phylogenomic analyses are robust, and the use of relative evolutionary divergence (RED) to establish taxonomic ranks follows current best practices in viral taxonomy.

Major Concerns

My primary concern relates to how definitively the nuclear replication hypothesis is presented throughout the manuscript, beginning with the title itself. The phrase "nuclear predators" in the title presents the nuclear replication hypothesis as established fact, when it is actually an inference based on indirect evidence. While the genomic evidence is compelling—the absence of DNA replication machinery combined with extensive spliceosomal introns strongly suggests nuclear dependence—these remain inferences rather than direct observations. The abstract continues this pattern, stating that these viruses "reveal that the nucleus of unicellular eukaryotes is a major niche for giant virus reproduction," again presenting hypothesis as fact.

This issue permeates the manuscript. For instance, statements like "the nucleus of unicellular eukaryotes is a major niche for giant virus reproduction" make strong claims based on indirect evidence. The manuscript would benefit from more nuanced discussion acknowledging the assumptions underlying these inferences. Could some viruses utilise host DNA polymerases in the cytoplasm? Might intron-rich viruses still have early cytoplasmic stages before nuclear entry? The authors briefly mention that some intron-containing demutoviruses might replicate partly in the cytoplasm, but this complexity deserves fuller treatment throughout the manuscript. A more accurate title might be "Intron-rich mirusviruses lacking DNA polymerases suggest widespread nuclear replication in unicellular eukaryotes" or something similarly qualified.

A critical technical concern involves the potential impacts of metagenomic binning, which is not yet widely established practice in viral ecology. The authors report 21 genomes exceeding 500 kb, but it's unclear whether these represent single contigs or binned assemblies. If these are bins rather than contiguous sequences, there's significant risk of genome size inflation through incorrect binning. This concern is particularly acute for the "cryptic putative orders" that are substantially enriched in singleton genes (43% on average). Could some of these unusual genomes represent chimeric bins combining multiple viral genomes or contamination from cellular sources? The authors should clarify: (1) how many of their large genomes are single contigs versus bins, (2) what additional validation was performed for binned genomes, especially those with unusual characteristics, and (3) whether bin contamination could explain some of the extreme genomic features observed. Given that viral binning relies primarily on sequence composition and coverage patterns—methods developed for cellular organisms with different genomic constraints—the possibility of binning artifacts deserves explicit discussion.

The uncertainty surrounding host range represents another significant limitation that deserves more explicit acknowledgment. While the authors broadly claim that "unicellular eukaryotes," direct evidence for host identity exists for only a tiny fraction of the characterised viruses. The vast majority of the 1,257 genomes derive from environmental metagenomes where virus-host linkages cannot be established. The remarkable genomic diversity observed—with GC content ranging from 25% to 72%—certainly suggests a broad host range, but this remains inference. This uncertainty has important implications for the evolutionary scenarios proposed, as different protist lineages possess vastly different nuclear architectures, splicing machineries, and life histories. The authors might consider attempting computational host prediction using established methods, such as co-occurrence patterns, or shared gene content with potential hosts. Alternatively, they should more clearly articulate why host identity uncertainty doesn't fundamentally affect their main conclusions about nuclear replication.

The functional interpretation of the novel genetic elements discovered, while fascinating, remains highly speculative and would benefit from more mechanistic consideration. For the spliceosomal homing introns, the proposed mechanism by which homing endonucleases would promote intron spread between viruses is unclear. Classical homing mechanisms operate within a genome through DNA repair processes, not between different viral genomes during co-infection. What selective advantage would intron acquisition provide to viruses, given that introns generally appear deleterious for viral fitness? Why are these elements specifically enriched in virion morphogenesis genes? Similarly, the MING-1 proteins are proposed to function as RNA chaperones based primarily on structural predictions, but no functional evidence supports this hypothesis. The extreme conservation of MING-1 insertion position across divergent viral lineages implies strong selective pressure, but for what function? The authors could either provide more detailed mechanistic models, present additional bioinformatic support, or more explicitly frame these as mysteries requiring future investigation.

Additional Considerations

The ecological implications of the nuclear versus cytoplasmic replication strategies deserve deeper exploration. How might these different strategies affect viral population dynamics, host range evolution, or ecosystem functioning? The authors touch on niche partitioning between viral phyla but could develop these ideas further.

The statistical analyses appear sound, though the authors should clarify whether multiple testing corrections were applied for the numerous chi-square tests performed on intron distributions.

Recommendations for Improvement

First, the authors should revise the title and abstract to more accurately reflect the inferential nature of the nuclear replication hypothesis. Throughout the manuscript, adopt more careful language distinguishing direct observations from inferences. Phrases like "strongly suggests," "is consistent with," or "likely indicates" would be more appropriate than definitive statements about nuclear replication in many instances.

Second, a dedicated paragraph in the discussion acknowledging the limitations of metagenomic inference would strengthen the manuscript. This should explicitly address what can and cannot be concluded from the available data, the potential impacts of binning artifacts, and outline key experiments or observations that would test the nuclear replication hypothesis.

Third, the novel genetic elements discovered deserve a more thorough mechanistic treatment. Even if definitive functions cannot be assigned, the authors should present specific, testable hypotheses about how spliceosomal homing introns might spread and what advantages MING-1 proteins might provide.

Finally, consider adding a brief section on host prediction attempts or explain why such analyses weren't pursued. This would address the host range uncertainty more directly.

Minor Points

- The choice of 98% ANI for non-redundancy (line 134) should be justified
- Clarify whether genome completeness (line 136) refers to mean or median
- Figure 4B text is difficult to read and should be enlarged
- Minor typographical error at line 452 (extra bracket)

Conclusion

This manuscript makes fundamental contributions to our understanding of giant virus diversity and evolution. The discovery of a major group of potentially nuclear-replicating DNA viruses with unique genomic features will undoubtedly stimulate new research directions in environmental virology and viral evolution. The quality of the genomic data and analyses is exceptional, and the biological insights are profound. While questions remain about host range, replication mechanisms, and the function of novel genetic elements, these uncertainties do not diminish the importance of the findings. With revisions addressing the presentation and interpretation concerns outlined above, this work would make an excellent addition to *Nature Microbiology* and significantly advance the field.

Decision Letter:

15th July 2025

Dear Dr Tom,

Thank you for your patience while your manuscript "Intron-rich mirusviruses are widespread nuclear predators of unicellular eukaryotes" was under peer-review at Nature Microbiology. It has now been seen by 3 referees, whose expertise and comments you will find at the end of this email. Although they find your work of considerable potential interest (as do we, on the editorial side!), they have nevertheless raised a number of concerns that will need to be addressed before we can consider the work further in Nature Microbiology.

In particular, the bulk of the concerns are more editorial in nature, and many fall into the category of parts that need to be toned down to better align with what the results support. There are also a number of methods quibbles, technical choices that should be better justified, cross validated or explained more fully. The referees also have a good suggestion to delve into the host range, and we agree that such an analysis would strengthen the paper. Overall, though there are many comments, these are straightforward reports, and generally speaking, quite positive.

Should further experimental data allow you to address these criticisms, we would be happy to look at a revised manuscript.

Please include a data availability statement as a separate section after Methods but before references, under the heading "Data Availability". This section should inform readers about the availability of the data used to support the conclusions of your study. This information includes accession codes to public repositories (data banks for protein, DNA or RNA sequences, microarray, proteomics data etc...), references to source data published alongside the paper, unique identifiers such as URLs to data repository entries, or data set DOIs, and any other statement about data availability. At a minimum, you should include the following statement: "The data that support the findings of this study are available from the corresponding author upon request", mentioning any restrictions on availability. If DOIs are provided, we also strongly encourage including these in the Reference list (authors, title, publisher (repository name), identifier, year). For more guidance on how to write this section please see: <http://www.nature.com/authors/policies/data/data-availability-statements-data-citations.pdf>

* If you have not done so already we suggest that you begin to revise your manuscript so that it conforms to our Article format instructions at <http://www.nature.com/nmicrobiol/info/final-submission>. Refer also to any guidelines provided in this letter.

When submitting the revised version of your manuscript, please pay close attention to our [href="https://www.nature.com/nature-portfolio/editorial-policies/image-integrity">Digital Image Integrity Guidelines](https://www.nature.com/nature-portfolio/editorial-policies/image-integrity) and to the following points below:

EXTENDED DATA FIGURES

Link Redacted

Note: This url links to your confidential homepage and associated information about manuscripts you may have submitted or be reviewing for us. If you wish to forward this e-mail to co-authors, please delete this link to your homepage first.

Nature Microbiology is committed to improving transparency in authorship. As part of our efforts in this direction, we are now requesting that all authors identified as 'corresponding author' on published papers create and link their Open Researcher and Contributor Identifier (ORCID) with their account on the Manuscript Tracking System (MTS), prior to acceptance. This applies to primary research papers only. ORCID helps the scientific community achieve unambiguous attribution of all scholarly contributions. You can create and link your ORCID from the home page of the MTS by clicking on 'Modify my Springer Nature account'. For more information please visit www.springernature.com/orcid.

If you wish to submit a suitably revised manuscript we would hope to receive it within 6 months. If you cannot send it within this time, please let us know. We will be happy to consider your revision, even if a similar study has been accepted for publication at Nature Microbiology or published elsewhere (up to a maximum of 6 months).

Yours sincerely,

Reviewer Expertise:

Referee #1: marine microbiology, mirusvirus, evolution, bioinformatics

Referee #2: marine viral ecology and evolution, metagenomics, bioinformatics

Referee #3: microbial ecology, viromics, bioinformatics

Reviewer Comments:

Reviewer #1 (Remarks to the Author):

Medvedeva et al. performed a large-scale exploration of metagenomic data from diverse biomes with a focus on the newly discovered mirusviruses. By improving and utilizing an HMM, they detected more than 21,000 major capsid protein genes. They also assembled a database of ~1,300 mirusvirus MAGs with a high level of completeness, the vast majority of which are entirely novel. They use core gene phylogenies and RED analysis to propose 17 different mirusvirus orders and 62 (decently represented) families, and >100 additional potential families represented by MAG singletons. The diversity of mirusviruses uncovered in this research is truly remarkable and contrasts that seen in the nucleocytoviruses.

A particularly exciting discovery is the presence of introns in many but not all, mirusviral genes, including a non-random distribution of introns in morphogenesis genes (2.7% of all mirusvirus genes harbour almost 30% of the introns). These introns appear to be 'mobile', and some harbour homing endonuclease-encoding regions. Trans splicing has also been inferred. This all speaks to regulation. This is the first description of HEs associated with 'canonical' spliceosomal introns. The Introner connection is also fascinating, although unclear in terms of evolutionary significance.

Specific comments:

- What about the putative hosts of such viruses? The title explicitly nods to unicellular eukaryotes, but there is not a single mention of the eukaryotic lineages known or predicted to be hosts is mentioned anywhere in the text. I think this could be improved and make the paper even more interesting to non-specialists and useful to virologists, microbial ecologists, protistologists etc. who are still new to mirusviruses.

Intron analysis: did the authors search for, and identify, and 'AT-AC' introns spliced by the 'minor' spliceosome?

Introns and splicing: It's not clear from the main text or materials and methods how the metatranscriptomic data were used. For example, what were the thresholds involved for coverage that could be trusted to infer splicing / presence of an intron? Fig 2 shows histogram data for an obviously compelling case involving trans splicing, but that's the only bit of information I can see. And how was trans-splicing detected in the first place? Researchers will want to know.

Figure 3 – what do the different shades of color in the different columns indicate (e.g., dark 'red' to lighter gray/red in DNA synthesis module)? This should be mentioned in the figure legend.

Gene flux section – starting line 409. Very interesting observations, but leaves the reader unsatisfied. One line 417, the authors refer to ‘apparent high level of gene flux’. Fig 3 shows between 0 and 10 genes across the mirusvirus families. Is this ‘high’? What does provisional definition of 50% identity mean, really, in terms of LGT for highly divergent viruses with very patchy gene presence. Directionality of transfer? What kinds of functions involved? None of this is mentioned in the text (there is Figure S15 which shows four genes / proteins stated as being evidence of transfers, but the phylogenies are difficult to interpret from such a thousand-foot level).

Line 82 – “Thus, with mirusviruses and nucleocytoviruses mainly replicating in the cytoplasm, the nucleus of unicellular eukaryotes currently appears to be a largely vacant ecological niche for large DNA virus infections.”

Perhaps a bit more caution here? “appearing to replicate in the cytoplasm”? I don’t know that we know enough about mirusviruses, do we? Almost all of the data are bioinformatics predictions. To be fair, later text is suitably nuanced and compelling, but it struck me as a tad too strong at this point.

Line 90, line 137: large and giant viruses. “Analysis of this genomic database shows that Mirusviricota is a highly diversified phylum of large and giant eukaryotic viruses...”. “Notably, 21 genomes were larger than 500 Kb, the standard genome size threshold defining giant viruses, making Mirusviricota the second phylum of large and giant eukaryotic viruses.

What is the value of distinguishing between large and ‘giant’? It is arbitrary, as the authors know, but also has the potential to clash with taxonomic IDs. From work of co-author Koonin and colleagues, it’s clear the viruses (virions and their genomes) have become ‘giant’ on multiple occasions. Moving forward, what if some members of the taxonomic entity Mirusviricota were found to have ‘small’ genomes? What would that mean? So the following: “making Mirusviricota the second phylum of large and giant eukaryotic viruses” (and also in the conclusion) doesn’t really mean much (to me at least). All this is to say that I think the field of ‘giant’ virus research has moved to a point where it doesn’t really matter how big their genomes are in terms of novelty and the more that is discovered the more we are likely to see a range of sizes.

This is a matter of taste and don’t feel strongly about it, but I don’t see the value of adding even more arbitrary designations of size (genomes or virus particles?). This is the source of regular confusion for non-specialist audiences and adding more categories now is just setting the field up for more headaches in the future.

Line 120: “However, we also detected mirusvirus MCPs inside shipworm, sponge and coral specimens (e.g., skeleton samples of *Porites lutea*35 and *Isopora palifera*36), ‘...’. Is the implication here that mirusviruses infect animals? Or are the MCPs associated with mirusviruses in protist symbionts / parasites within animals? Probably should be careful at this stage not to provide substrate for misinterpretation?

Line 139: “The genomic database includes four circular episomes7,9, the only known chromosomal integrant9 ,...”

Predicting circularity can be difficult, so maybe “obviously circular episomes”? This section of text leaves the reader wondering what the structure of the $1204 - 4 - 1 = 1199$ mirusvirus genomic elements are (inc. the 51 previously ID’d ones). It doesn’t really come up again.

Line 238: “ Notably, Soporavirales displayed an intriguing trend, with high intron density in the episome (653 introns/Mb) and chromosomal integrant (624 introns/Mb) of one eukaryotic isolate9 and very few introns in the other 14 genomes (25 introns/Mb).” If the authors are referring to only one episome and integrant, and those elements are only ~300 Kb, why refer to them in terms of introns / Mb? I realize this is the useful metric for looking at large MAG datasets, but it seems misleading in this case.

Line 252: “contained no introns”. Contained no obvious introns?

Line 284: “The mirusvirus introns encoding homing endonucleases lack recognizable signatures of group I or II introns and instead are flanked by typical spliceosomal donor and acceptor sites”. What signatures did the authors look for? I think much of the self-splicing intron signatures studied in the literature are secondary structure based. It would be good for the authors to articulate this briefly (either at this part of the text or elsewhere). Were the predicted introns folded (with the HE-encoding regions removed)? I doubt it, as this would have been a ton of work, but I recommend that the authors clarify the extent to which they dug into the different possibilities here. Are these introns group I/II introns in origin that have somehow adopted GT-AG splice boundaries and are now removed by the spliceosome? Or are they canonical spliceosomal introns that have recently acquired HE genes? This could well be questions for the future, but at least touching on it here would seem to be worth it. I can only assume that they are spliceosomal introns first, since most are too small or (in the case of Fig. 2B) too large to be self-splicing (at least based on what is known about GPI and II introns).

Line 289: “Remarkably, SHI are for the most part inside the virion morphogenesis genes (64.1% of all SHI), and display target gene specificity, ...”. They are predicted to have target gene specificity, but this hasn’t been demonstrated experimentally. This is made clear when talking about the HE function in the following sentence, but I’d suggest clarifying the language a bit here.

Line 318: similar to above, “the corresponding introns display strong target gene preference...”. I don’t understand exactly what “display” means here. Do you mean the intron insertion sites ‘are consistent with’ target site preference on the part of the HEs (or whatever proteins)? The introns themselves don’t display target gene preference, that is a function of a trans-acting proteins that mediate homing, correct? For this comment and above, I think the language could be improved. I apologize if I have misunderstood.

Line 693: “‘Intron-borne’ genes”. As written this could mean genes that emerged de novo from introns (which can happen), but

this is not what the authors mean. They mean genes inside existing introns (and presumably have landed within the intron)?

Reviewer #2 (Remarks to the Author):

In this manuscript Medvedeva et al., leverage on Tara Ocean data and taxonomically demonstrate Mirusviricota comprises a highly diversified second phylum of large and giant eukaryotic viruses that rivals the evolutionary scope and functional complexity of viruses in the first characterized phylum, Nucleocytoviricota. Functionally they describe gene context difference including presence of hallmark genes required for replication in the cytoplasm as well some mirusviruses that included spliceosome introns linked to genes encoding essential proteins. They also propose two evolutionary scenarios. Overall, the manuscript is well structured and provided invaluable insights on the ever-growing NCLV.

Abstract, Line 22: Please remove references from the abstract, as these are typically discouraged in that section.

Line 134: The statement indicates that non-redundant mirusvirus genomes share an average nucleotide identity (ANI) of <98%. Should this be >98% instead?

Lines 708–713: The methodology for constructing the concatenated gene tree is unclear. Please provide additional details regarding how the tree was built, particularly whether each gene partition was assigned a unique evolutionary model.

Line 719: RED scores are indeed used in GTDB taxonomy assignments. However, the rationale behind selecting a RED score threshold of 11 is not clear. Were any benchmarking efforts performed using ICTV reference sequences to justify this cutoff? Some reasoning or justification for this threshold is necessary.

Line 742: Functional annotation is based on Pfam, which is a reasonable approach. However, Pfam annotations can be ambiguous in certain cases. Have the authors validated the annotations using an additional database, such as eggNOG or KEGG?

Figure S15: How is lateral gene transfer (LGT) defined as recent in this figure? There appears to be no time calibration in the phylogenetic tree to support this statement.

Data Availability: A link to the updated HMM is not provided. The reference provided is to the initial HMM only. Please include access to the updated version.

Additional Question: The authors present 3D-predicted structures of the terminase protein. How do these predicted structures from the MAGs compare with known reference structures? Does a structure-based evolutionary analysis (structure based tree) supports the inferred functions.

While two evolutionary scenario's are proposed in a future direction paragraph under conclusion would be great to discuss how to mechanistically test.

Reviewer #3 (Remarks to the Author):

Summary

This manuscript presents a tour de force in environmental virology, dramatically expanding our knowledge of mirusvirus diversity through the characterisation of over 1,000 high-quality genomes. The authors make the compelling case that Mirusviricota represents a second phylum of large and giant eukaryotic viruses, rivaling Nucleocytoviricota in evolutionary scope and functional complexity. The discovery of widespread nuclear-replicating viruses that lack DNA polymerases and are enriched in spliceosomal introns challenges conventional understanding of large DNA virus biology. While the work is technically impressive and the major conclusions are well-supported, several aspects of the interpretation and presentation warrant further consideration.

Strengths

The scope and scale of this study are remarkable. By surveying over 100,000 metagenomic assemblies and developing iterative binning approaches, the authors have increased the known diversity of mirusviruses by an order of magnitude. The methodological framework, particularly the iterative refinement of genome bins and the development of clade-specific quality metrics, represents best practices in environmental genomics that will be of interest to the field.

The manuscript's most exciting contributions are its novel biological discoveries. The identification of spliceosomal homing introns represents the first documented case of homing endonucleases within spliceosomal introns, potentially revealing a new mechanism for intron mobility. Similarly, the evidence for trans-splicing in terminase genes, supported by metatranscriptomic data, provides rare documentation of this phenomenon in DNA viruses. Perhaps most striking is the discovery of giant viruses that apparently lack DNA polymerases yet maintain genomes exceeding 500 kb, fundamentally challenging our assumptions about the minimal requirements for large DNA virus replication.

The evolutionary framework presented is thoughtful and well-reasoned. The authors' proposal of "evolutionary trap" and "steal and escape" models for transitions between cytoplasmic and nuclear replication provides testable hypotheses for understanding mirusvirus evolution. The phylogenomic analyses are robust, and the use of relative evolutionary divergence (RED) to establish taxonomic ranks follows current best practices in viral taxonomy.

Major Concerns

My primary concern relates to how definitively the nuclear replication hypothesis is presented throughout the manuscript, beginning with the title itself. The phrase "nuclear predators" in the title presents the nuclear replication hypothesis as established fact, when it is actually an inference based on indirect evidence. While the genomic evidence is compelling—the absence of DNA replication machinery combined with extensive spliceosomal introns strongly suggests nuclear dependence—these remain inferences rather than direct observations. The abstract continues this pattern, stating that these viruses "reveal that the nucleus of unicellular eukaryotes is a major niche for giant virus reproduction," again presenting hypothesis as fact.

This issue permeates the manuscript. For instance, statements like "the nucleus of unicellular eukaryotes is a major niche for giant virus reproduction" make strong claims based on indirect evidence. The manuscript would benefit from more nuanced discussion acknowledging the assumptions underlying these inferences. Could some viruses utilise host DNA polymerases in the cytoplasm? Might intron-rich viruses still have early cytoplasmic stages before nuclear entry? The authors briefly mention that some intron-containing demotoviruses might replicate partly in the cytoplasm, but this complexity deserves fuller treatment throughout the manuscript. A more accurate title might be "Intron-rich mirusviruses lacking DNA polymerases suggest widespread nuclear replication in unicellular eukaryotes" or something similarly qualified.

A critical technical concern involves the potential impacts of metagenomic binning, which is not yet widely established practice in viral ecology. The authors report 21 genomes exceeding 500 kb, but it's unclear whether these represent single contigs or binned assemblies. If these are bins rather than contiguous sequences, there's significant risk of genome size inflation through incorrect binning. This concern is particularly acute for the "cryptic putative orders" that are substantially enriched in singleton genes (43% on average). Could some of these unusual genomes represent chimeric bins combining multiple viral genomes or contamination from cellular sources? The authors should clarify: (1) how many of their large genomes are single contigs versus bins, (2) what additional validation was performed for binned genomes, especially those with unusual characteristics, and (3) whether bin contamination could explain some of the extreme genomic features observed. Given that viral binning relies primarily on sequence composition and coverage patterns—methods developed for cellular organisms with different genomic constraints—the possibility of binning artifacts deserves explicit discussion.

The uncertainty surrounding host range represents another significant limitation that deserves more explicit acknowledgment. While the authors broadly claim that mirusviruses infect "unicellular eukaryotes," direct evidence for host identity exists for only a tiny fraction of the characterised viruses. The vast majority of the 1,257 genomes derive from environmental metagenomes where virus-host linkages cannot be established. The remarkable genomic diversity observed—with GC content ranging from 25% to 72%—certainly suggests a broad host range, but this remains inference. This uncertainty has important implications for the evolutionary scenarios proposed, as different protist lineages possess vastly different nuclear architectures, splicing machineries, and life histories. The authors might consider attempting computational host prediction using established methods, such as co-occurrence patterns, or shared gene content with potential hosts. Alternatively, they should more clearly articulate why host identity uncertainty doesn't fundamentally affect their main conclusions about nuclear replication.

The functional interpretation of the novel genetic elements discovered, while fascinating, remains highly speculative and would benefit from more mechanistic consideration. For the spliceosomal homing introns, the proposed mechanism by which homing endonucleases would promote intron spread between viruses is unclear. Classical homing mechanisms operate within a genome through DNA repair processes, not between different viral genomes during co-infection. What selective advantage would intron acquisition provide to viruses, given that introns generally appear deleterious for viral fitness? Why are these elements specifically enriched in virion morphogenesis genes? Similarly, the MING-1 proteins are proposed to function as RNA chaperones based primarily on structural predictions, but no functional evidence supports this hypothesis. The extreme conservation of MING-1 insertion position across divergent viral lineages implies strong selective pressure, but for what function? The authors could either provide more detailed mechanistic models, present additional bioinformatic support, or more explicitly frame these as mysteries requiring future investigation.

Additional Considerations

The ecological implications of the nuclear versus cytoplasmic replication strategies deserve deeper exploration. How might these different strategies affect viral population dynamics, host range evolution, or ecosystem functioning? The authors touch on niche partitioning between viral phyla but could develop these ideas further.

The statistical analyses appear sound, though the authors should clarify whether multiple testing corrections were applied for the numerous chi-square tests performed on intron distributions.

Recommendations for Improvement

First, the authors should revise the title and abstract to more accurately reflect the inferential nature of the nuclear replication hypothesis. Throughout the manuscript, adopt more careful language distinguishing direct observations from inferences. Phrases like "strongly suggests," "is consistent with," or "likely indicates" would be more appropriate than definitive statements about nuclear replication in many instances.

Second, a dedicated paragraph in the discussion acknowledging the limitations of metagenomic inference would strengthen the manuscript. This should explicitly address what can and cannot be concluded from the available data, the potential impacts of binning artifacts, and outline key experiments or observations that would test the nuclear replication hypothesis.

Third, the novel genetic elements discovered deserve a more thorough mechanistic treatment. Even if definitive functions cannot be assigned, the authors should present specific, testable hypotheses about how spliceosomal homing introns might spread and what advantages MING-1 proteins might provide.

Finally, consider adding a brief section on host prediction attempts or explain why such analyses weren't pursued. This would

address the host range uncertainty more directly.

Minor Points

- The choice of 98% ANI for non-redundancy (line 134) should be justified
- Clarify whether genome completeness (line 136) refers to mean or median
- Figure 4B text is difficult to read and should be enlarged
- Minor typographical error at line 452 (extra bracket)

Conclusion

This manuscript makes fundamental contributions to our understanding of giant virus diversity and evolution. The discovery of a major group of potentially nuclear-replicating DNA viruses with unique genomic features will undoubtedly stimulate new research directions in environmental virology and viral evolution. The quality of the genomic data and analyses is exceptional, and the biological insights are profound. While questions remain about host range, replication mechanisms, and the function of novel genetic elements, these uncertainties do not diminish the importance of the findings. With revisions addressing the presentation and interpretation concerns outlined above, this work would make an excellent addition to Nature Microbiology and significantly advance the field.

Version 1:

Reviewer comments:

Reviewer #1

(Remarks to the Author)

I acknowledge the efforts made by the authors to revise their manuscript. I particularly appreciated the extraordinarily detailed rebuttal letter, which addressed, among many other things, the issue of metagenomic binning (directed at Reviewer #3 but very helpful for me as well).

All of my concerns with the original manuscript have been satisfactorily addressed. I congratulate the authors on this fine study.

Reviewer #2

(Remarks to the Author)

We thank the authors for taking the time to address the concerns raised and for making the requested changes. I have reviewed both the revised manuscript and the point-by-point response to the both my question as well as the other two reviewers. I have no further comments or suggestions on the revised draft of the manuscript.

Reviewer #3

(Remarks to the Author)

I reviewed this manuscript previously as reviewer #3 from a background of viral ecology, taxonomy and viromics. This was already a very nice manuscript, and now it is even better. I'm happy to see that the authors have fully and thoughtfully responded to the comments from all reviewers. This is a fine example of the review process working exactly as it should. All credit to the authors.

My only remaining comment is that the authors should now probably reference this emerging work from Merk et al. in Nucleic Acids Research (<https://doi.org/10.1093/nar/gkaf761>). I appreciate this late request has slightly unfortunate timing.

Decision Letter:

Our ref: NMICROBIOL-25062017A

18th September 2025

Dear Tom,

Thank you for submitting your revised manuscript "Intron-rich mirusviruses widespread in aquatic habitats are predicted to reproduce in nuclei of unicellular eukaryotes" (NMICROBIOL-25062017A). It has now been seen by the original referees and their comments are below. The reviewers find that the paper has improved in revision, and therefore we'll be happy in principle to publish it in Nature Microbiology, pending minor revisions to satisfy the referees' final requests and to comply with our editorial and formatting guidelines.

Thank you again for your interest in Nature Microbiology Please do not hesitate to contact me if you have any questions.

Sincerely,

Reviewer #1 (Remarks to the Author):

I acknowledge the efforts made by the authors to revise their manuscript. I particularly appreciated the extraordinarily detailed rebuttal letter, which addressed, among many other things, the issue of metagenomic binning (directed at Reviewer #3 but very helpful for me as well).

All of my concerns with the original manuscript have been satisfactorily addressed. I congratulate the authors on this fine study.

Reviewer #2 (Remarks to the Author):

We thank the authors for taking the time to address the concerns raised and for making the requested changes. I have reviewed both the revised manuscript and the point-by-point response to the both my question as well as the other two reviewers. I have no further comments or suggestions on the revised draft of the manuscript.

Reviewer #3 (Remarks to the Author):

I reviewed this manuscript previously as reviewer #3 from a background of viral ecology, taxonomy and viromics. This was already a very nice manuscript, and now it is even better. I'm happy to see that the authors have fully and thoughtfully responded to the comments from all reviewers. This is a fine example of the review process working exactly as it should. All credit to the authors.

My only remaining comment is that the authors should now probably reference this emerging work from Merk et al. in Nucleic Acids Research (<https://doi.org/10.1093/nar/gkaf761>). I appreciate this late request has slightly unfortunate timing.

Version 2:

Decision Letter:

14th October 2025

Dear Tom,

I am pleased to accept your Article "Widespread, intron-rich mirusviruses are predicted to reproduce in nuclei of unicellular eukaryotes" for publication in Nature Microbiology. Thank you for having chosen to submit your work to us and many congratulations.

Authors may need to take specific actions to achieve compliance with funder and institutional open access mandates. If your research is supported by a funder that requires immediate open access (e.g. according to [Plan S principles](https://www.springernature.com/gp/open-science/plan-s-compliance) or the [NIH public access policy](https://www.springernature.com/gp/open-science/us-federal-agency-compliance)) then you should select the gold OA route, and we will direct you to the compliant route where possible. Because authors warrant under our subscription licensing terms that they haven't committed to licensing any version of their article under a licence inconsistent with the terms of our agreement – including the applicable embargo period – publication under the subscription model isn't suitable for authors whose funders require no embargo.

With kind regards,

P.S. Click on the following link if you would like to recommend Nature Microbiology to your librarian <http://www.nature.com/subscriptions/recommend.html#forms>

** Visit the Springer Nature Editorial and Publishing website at http://editorial-jobs.springernature.com?utm_source=ejP_NMicro_email&utm_medium=ejP_NMicro_email&utm_campaign=ejp_NMicro for more information about our career opportunities. If you have any questions please click [here](mailto:editorial.publishing.jobs@springernature.com).**

August 2025

Dear editors,

We are grateful for the time you have invested into the consideration of our manuscript, and for the valuable comments we received from the reviewers. We are gratified that all reviewers agree that the discovery and characterization of new clades of mirusviruses with intron-rich genes is a notable contribution to our understanding of the evolution, diversity and functioning of large and giant eukaryotic viruses. We believe our revision addresses all points raised by the reviewers, which undoubtedly yielded a much stronger manuscript. One exception is the extensive host range prediction analysis, for which we did not include new results but instead cited the current literature more thoroughly. We note that such analyses are underway in our group and we aim to release the results within the next 6 months as a preprint. Here is a summary of some of the key changes we made to address the relevant suggestions and main points of concern of the reviewers:

- (1) We toned down our conclusions regarding nuclear replication of intron-rich mirusviruses lacking DNA polymerase. This is reflected in the title, abstract, Results and Discussion.
- (2) We created a “Supplemental Information” document that extensively describes our two genome-resolved metagenomic approaches (manual binning and curation within *Tara* Oceans followed by the iterative automatic binning of mOTU bins). The document contains detailed method descriptions, including an example of manual binning for a MAG corresponding to one of the cryptic putative orders to gain confidence in the biological relevance of those highly diversified lineages.
- (3) We validated all functional predictions using alternative annotation methods, namely, eggNOG and KEGG Kofams. These results are now part of Table S04.
- (4) We incorporated a full description of the gene flux analysis in the Methods section and substantially expanded the scope of results presented in Table S07.
- (5) We now report that a minority of AT-AC introns are predicted to be spliced by the ‘minor’ spliceosome. These results are now part of the Table S04.

Bellow, you will find our point-by-point responses (**in brown**) to editorial and reviewer comments (**in blue**).

Tom Delmont and Mart Krupovic,
On behalf of all the authors

Brief response to editorial comments:

We have toned down our main conclusions regarding the cytoplasmic versus nuclear replication site of viruses belonging to the putative *Mirusviricota* orders, to better reflect the lack of available cultured representatives or imaging data related to the two putative orders of Okeanovirales and Styxvirales that are central to our investigations. Accordingly, changes were made in the title, abstract and discussion sections. However, we have maintained a confident tone on this issue because it is hard to imagine (a view also shared by the reviewers) how viruses with genes full of spliceosomal introns (especially in the virion morphogenesis gene module) and lacking any polymerase (among other key functions for cytoplasmic replication) can build a viral factory and replicate in the cytoplasm. Regarding the hosts of the newly described mirusviruses, we are working on

a parallel study dedicated to gene flux, which strongly indicates that mirusviruses mainly infect unicellular eukaryotes abundant in aquatic ecosystems worldwide (to be published separately). For the sake of transparency, we share here in confidence with the editors and the reviewers that okeanoviruses most likely mainly infect *Haptista* whereas styxviruses most likely infect *Cryptista*. In the revision of the present manuscript, we have now cited all the available literature on putative hosts of mirusviruses, to address the respective comments of the reviewers. Although these assignments were made in the absence of genomes for most clades of *Mirusviricota*, they already point to unicellular eukaryotes as the main hosts.

Reviewer #1

Medvedeva et al. performed a large-scale exploration of metagenomic data from diverse biomes with a focus on the newly discovered mirusviruses. By improving and utilizing an HMM, they detected more than 21,000 major capsid protein genes. They also assembled a database of ~1,300 mirusvirus MAGs with a high level of completeness, the vast majority of which are entirely novel. They use core gene phylogenies and RED analysis to propose 17 different mirusvirus orders and 62 (decently represented) families, and >100 additional potential families represented by MAG singletons. The diversity of mirusviruses uncovered in this research is truly remarkable and contrasts that seen in the nucleocytoviruses.

A particularly exciting discovery is the presence of introns in many but not all, mirusviral genes, including a non-random distribution of introns in morphogenesis genes (2.7% of all mirusvirus genes harbour almost 30% of the introns). These introns appear to be 'mobile', and some harbour homing endonuclease-encoding regions. Trans splicing has also been inferred. This all speaks to regulation. This is the first description of HEs associated with 'canonical' spliceosomal introns. The Introner connection is also fascinating, although unclear in terms of evolutionary significance.

We thank the reviewer for this positive summary and for their contributions to the overall strength of our manuscript. Their expertise on introns was particularly well appreciated and persuaded us to consider more carefully the potential origins and spreading mechanisms of the enigmatic spliceosomal introns encoding homing endonucleases and MING-1 proteins that appear to specifically target the key virion morphogenesis genes in numerous mirusvirus genomes.

Specific comments:

- What about the putative hosts of such viruses? The title explicitly nods to unicellular eukaryotes, but there is not a single mention of the eukaryotic lineages known or predicted to be hosts is mentioned anywhere in the text. I think this could be improved and make the paper even more interesting to non-specialists and useful to virologists, microbial ecologists, protistologists etc. who are still new to mirusviruses.

We agree that the host range of mirusviruses was not sufficiently described in the submitted manuscript. In the revised manuscript, we amended the main text to more thoroughly cite the published literature on the subject, including a recent preprint by Hiroyuki Ogata's lab (also an author of the present study). In the current literature, it is

unclear what are the main hosts for previously explored mirusviruses (mainly *Demutovirales*), and so we elected not to randomly cite a few high-ranking clades. Although detailed exploration of the host range of mirusviruses is beyond the scope of the present manuscript, we are actively working on a parallel study dedicated to gene flux between mirusviruses and eukaryotes. For the sake of transparency, we share here in confidence that okeanoviruses most likely infect *Haptista*, whereas styxviruses most likely infect *Cryptista*. The results for *Demutovirales* are less clear but the strongest eukaryotic signal is for the MAST clade. Overall, this ongoing study clearly links the 3 major putative orders of *Mirusviricota* with 3 major lineages of unicellular eukaryotes: Haptista, Cryptista and Stramenopiles. Given the preliminary state of the results and the volume of the analyses, the results of the host prediction cannot be integrated into the present manuscript without compromising the focus of the study.

Intron analysis: did the authors search for, and identify, and ‘AT-AC’ introns spliced by the ‘minor’ spliceosome?

The overwhelming majority of introns predicted using BRAKER2 in this study contain ‘GT-AG’ (96.41%) or ‘GC-AG’ (3.57%) splicing sites and therefore are likely processed by the major spliceosome. Similar pattern was observed for introns inferred directly from the metatranscriptomic data (for the 4,607 introns supported by at least 10 RNAseq reads, see the table below): ‘GT-AG’ (86.71%) or ‘GC-AG’ (11.33%). In addition, 90 introns (1.95%) inferred from metatranscriptomic data were found to contain ‘AT-AC’ splicing sites typical of the minor spliceosome. Interestingly, 56 of the minor spliceosomal introns were found in genomes of *Demutovirales_05* family. From this analysis, we conclude that introns spliced by the minor spliceosome are largely absent in *Demutovirales* (except *Demutovirales_05*), *Okeanovirales* and *Styxvirales*. More RNAseq data is needed to confirm the presence/absence of these introns in other mirusvirus lineages.

Table 1. Introns inferred directly from metatranscriptomic data.

Family	major spliceosome		minor spliceosome
	GT-AG	GC-AG	AT-AC
Demuto_01	11	2	0
Demuto_02	41	6	3
Demuto_03	4	1	2
Demuto_04	4	3	1
Demuto_05	275	60	56
Demuto_06	0	1	0
Demuto_07	89	4	2
Demuto_09	64	1	0
Demuto_10	100	10	1
Demuto_11	4	0	0
Demuto_12	8	1	0
Okeano_01	2633	324	11
Okeano_03	2	0	0
Okeano_04	16	1	0
Extra_Family_10	48	3	2
Extra_Family_14	5	0	0
Extra_Family_16	0	1	0

Extra_Family_28	1	0	0
Extra_Family_29	22	5	1
Extra_Family_31	2	4	0
Extra_Family_32	0	1	0
Extra_Family_33	4	0	0
Extra_Family_38	1	0	0
Extra_Family_39	1	0	0
Extra_Family_41	20	13	0
Extra_Family_42	39	21	0
Styx_01	255	25	4
Styx_02	0	0	1
Styx_03	14	1	0
Styx_04	177	16	1
not classified MAG	155	18	5
Total	3995	522	90

We have now incorporated those findings in Table S04 and in the main text, as follows:

“Here, we found that numerous mirusvirus genes contain spliceosomal introns (37,703 introns across 17,119 genes covering almost all genomes). An overwhelming majority of identified introns contained well conserved canonical GT-AG splice sites predicted to be processed by the major spliceosome and displayed a length distribution peaking at ~80 bp, typical of spliceosomal introns in protists^{19,44} (Table S4 and Figure S3). Although uncommon, we also identified a small fraction of introns with the AT-AC splice sites, characteristic of the minor spliceosome (noncanonical splicing). These noncanonical introns are enriched in a single putative family within Demutovirales, Demuto_05 (Table S4).”

Introns and splicing: It’s not clear from the main text or materials and methods how the metatranscriptomic data were used. For example, what were the thresholds involved for coverage that could be trusted to infer splicing / presence of an intron? Fig 2 shows histogram data for an obviously compelling case involving trans splicing, but that’s the only bit of information I can see. And how was trans-splicing detected in the first place? Researchers will want to know.

For the prediction of introns, we used BRAKER2 with an option --eptmmode. This option allows the algorithm to use hints from provided metatranscriptomic mapping and hints from the trusted set of proteins. We used the default coverage threshold for including a transcriptomic hint, which is a minimum of 10 reads. We have now amended the Methods section “Gene model for Mirusviricota”, as follow:

“For the metatranscriptomic data, we used a default threshold for the minimal read coverage to support the presence of an intron (minimum of 10 reads).”

The trans-splicing was detected by careful manual analysis of the genomic sequences and metatranscriptomic data. In addition, tblastn search was used to detect the ATPase domain of the terminase. Before the discovery of trans-splicing, we were puzzled by the sequences of terminase large subunit in most *Okeanovirales* and *Styxvirales* because the

essential Walker B motif of the ATPase domain and the nuclease domain could not be detected. Instead, despite being one of the most conserved proteins in mirusviruses, the C-terminal region following the initially predicted ATPase domain of the terminases contained random sequences of different length, encoded by a separate exon. This was clearly an error of the gene prediction method. Fortunately, some of the mirusviruses had a complete sequence of the terminase with a canonical Walker B motif. We used tblastn (protein query against translated nucleotide database) to search for the homologous C-terminal terminase regions encompassing the Walker B motif and the nuclease domain in *Okeanovirales* and *Styxvirales* genomes. We were surprised to find the missing part (often split by introns) far away from the N-terminal region of the terminase (encompassing the Walker A motif), and sometimes encoded on the opposite strand. We hypothesized that to reassemble the coding regions into a functional mRNA, trans-splicing would be necessary. The literature review indicated that a similar scenario was predicted for alloherpesviruses, supporting the trans-splicing hypothesis (which we eventually confirmed using metatranscriptomics). We have now amended the Methods section “Gene model for Mirusviricota”, as follows:

“Finally, the trans-splicing of terminase genes was detected by manual curation of genomic sequences, manual analysis of metatranscriptomic data and tblastn searches for homologs of the C-terminal region of the terminase encompassing the Walker B motif of the ATPase domain and the nuclease domain in Okeanovirales and Styxvirales genomes.”

Figure 3 – what do the different shades of color in the different columns indicate (e.g., dark ‘red’ to lighter gray/red in DNA synthesis module)? This should be mentioned in the figure legend.

We have now corrected this oversight:

“Figure 3: Niche partitioning of Mirusviricota between the cytoplasm and the nucleus. The figure summarizes average genomic trends (functions, spliceosomal introns, terminase trans-splicing, gene fluxes with nucleocytoviruses) across putative families of Demutovirales, Okeanovirales, Styxvirales and Soporavirales. For each family, the fraction of genomes containing each gene of the different functional modules (DNA synthesis, DNA replication and transcription) is presented as a heatmap, with the colour intensity reflecting the corresponding fraction.

Gene flux section – starting line 409. Very interesting observations, but leaves the reader unsatisfied. One line 417, the authors refer to ‘apparent high level of gene flux’. Fig 3 shows between 0 and 10 genes across the mirusvirus families. Is this ‘high’? What does provisional definition of 50% identity mean, really, in terms of LGT for highly divergent viruses with very patchy gene presence. Directionality of transfer? What kinds of functions involved? None of this is mentioned in the text (there is Figure S15 which shows four genes / proteins stated as being evidence of transfers, but the phylogenies are difficult to interpret from such a thousand-foot level).

We regret that sufficient details on this aspect of the work were missing in the original version of the manuscript. We agree that parts of the highlighted paragraph were

confusing or lacked details in the Method section. We substantially amended the relevant parts of the manuscript.

First, we now have two Methods sections dedicated to this analysis, which read as follows:

“Gene flux between mirusviruses and nucleocytoviruses: We performed two complementary protein sequence similarity-based analyses to explore the extent of gene flux between mirusviruses and nucleocytoviruses, under the assumption that high sequence similarity between proteins from these distant virus phyla most likely results from gene flux. First, we computed sequence similarity between mirusvirus proteins (query) and nucleocytovirus proteins (subject sequences of the GOEV resource¹) using diamond blast70 v2.18 (“--ultra-sensitive”; cutoffs: percent identity of at least 50% and a bitscore of at least 100). Second, we computed sequence similarity between all nucleocytovirus proteins (query) and mirusvirus proteins (subject sequences) using the same diamond blast parameters. Each query with at least one hit meeting the above criteria, along with its best hit being from the other phylum, was considered part of the pool of genes horizontally transferred between the two phyla.

Function-aware prediction of directionality for the gene flux: We performed a third protein sequence similarity-based analysis to predict the directionality of gene flux between mirusviruses and nucleocytoviruses in the context of functional annotations. We computed sequence similarity using a merged file containing all the mirusvirus and nucleocytovirus proteins, both as the query and the subject sequences, with diamond blast70 v2.18 (“--ultra-sensitive”; cutoffs: percent identity of at least 50% and a bitscore of at least 100). We analyzed the 20 best hits per query (excluding the hit corresponding to the query) and quantified for each protein the number of hits (up to 19) from each phylum. This last step provided a high-ranking taxonomic ratio between Mirusviricota and Nucleocytoviricota for all the proteins with at least one diamond blast hit meeting the criteria of high similarity. The higher score was used to infer the most likely evolutionary origin of the genes (Mirusviricota versus Nucleocytoviricota) independently of genome-level taxonomy (i.e., a mirusvirus MAG can contain multiple genes linked to Nucleocytoviricota based on the diamond blast) and thus the direction of the gene flux. For example, a mirusvirus protein with only mirusvirus proteins among its top hits would have a score of 1 for Mirusviricota, suggesting it is unlikely to have been recently laterally transferred from a nucleocytovirus. In contrast, another mirusvirus protein with a majority of nucleocytovirus proteins in its top hits (score <0.5 for Mirusviricota) would be labelled as Nucleocytoviricota, suggesting it might have been recently laterally transferred from a nucleocytovirus into a mirusvirus. Finally, for each pool of genes corresponding to the same functional annotation (Pfam) within each of the two phylum-level genomic databases (Mirusviricota genomes versus Nucleocytoviricota genomes), we quantified the number of genes predicted to have been laterally transferred from the other phylum (see above criteria). Taking “dUTPase” as an example of functional annotation, we found that 10% of corresponding genes among the Mirusviricota genomes had a score <0.5 for Mirusviricota, whereas only 0.4% of corresponding genes among the Nucleocytoviricota genomes had score a <0.5 for Nucleocytoviricota. This signal suggests flux of “dUTPase” from nucleocytoviruses towards mirusviruses. Although this approach allows prediction of gene flux

directionality between the two viral phyla, phylogenetic analyses are required to move beyond this preliminary, crude step. For four examples of functions displaying an interesting flux directionality signal, we performed phylogenetic analyses to better assess lateral gene transfers between the two phyla. For that, we performed alignments at the amino acid level using MAFFT⁸² v7.490 with the FFT-NS-i option and default parameters. In each set of protein alignments, sites with more than 70% gaps were trimmed using trimAl⁸³ v1.4.1. Phylogenetic reconstructions were performed using IQ-TREE⁸⁴ v1.6.12 with “-m MFP -safe -alrt 1000 -bb 1000” parameters. ModelFinder⁸⁵ was used to determine and select the best-fitting model. Supports were computed from 1,000 replicates for the Shimodaira–Hasegawa (SH)-like aLRT⁸⁶ and UFBoot⁸⁷. Nodes were considered as strongly supported when SH-like aLRT \geq 80% and UFBoot \geq 95%, moderately supported when only one of the two cut-offs was met, and poorly supported when none of the two cut-offs was met. Anvi'o v.8 was used to visualize the phylogenetic trees.”

Second, we have extensively amended Table S7, which now provides 3 distinct results compared to just one included initially. Importantly, the table now describes the functions (Pfam annotations) related to gene flux between the two phyla in the context of their overall prevalence in each phylum. Our gene flux analysis involves numerous functions, which makes it difficult to summarize. However, we found that some of the most prevalent functions belonged to the DNA precursor synthesis module, and this is now mentioned in the main text (see below).

Third, we have modified the paragraph in the Results section to address the points made by the reviewer:

*“To test this hypothesis, we performed a comprehensive sequence comparison of proteins encoded by Mirusviricota and Nucleocytoviricota. We observed **relatively high levels of gene flux** (~~provisionally~~ defined by presence of highly similar homologs, with >50% amino acid sequence identity) between Nucleocytoviricota and intron-poor mirusvirus families of Demutovirales, and, conversely, either no signal or **lower levels of gene flux** between Nucleocytoviricota and intron-rich mirusvirus groups (Figures 3, 4 and S14, Table S7). **Relatively recent gene transfers (as inferred from the high sequence identity between homologous proteins from the two phyla) between specific lineages of mirusviruses and nucleocytoviruses cover a wide range of functions that include a particularly strong signal for the DNA precursor synthesis module (e.g., dUTPase). For these genes, Nucleocytoviricota to Mirusviricota gene flux directionality was confidently predicted (Table S7) and validated at least partially by phylogenetic analyses (Figure S15).”***

Line 82 – “Thus, with mirusviruses and nucleocytoviruses mainly replicating in the cytoplasm, the nucleus of unicellular eukaryotes currently appears to be a largely vacant ecological niche for large DNA virus infections.” Perhaps a bit more caution here? “appearing to replicate in the cytoplasm”? I don’t know that we know enough about mirusviruses, do we? Almost all of the data are bioinformatics predictions. To be fair, later text is suitably nuanced and compelling, but it struck me as a tad too strong at this point.

The sentence now reads as follow:

*“Thus, with mirusviruses and nucleocytoviruses **predicted to mainly replicate in the cytoplasm, the nucleus of unicellular eukaryotes currently appears to be a largely vacant ecological niche for large DNA virus infections**”*

Line 90, line 137: large and giant viruses. “Analysis of this genomic database shows that Mirusviricota is a highly diversified phylum of large and giant eukaryotic viruses...”. “Notably, 21 genomes were larger than 500 Kb, the standard genome size threshold defining giant viruses, making Mirusviricota the second phylum of large and giant eukaryotic viruses.

What is the value of distinguishing between large and ‘giant’? It is arbitrary, as the authors know, but also has the potential to clash with taxonomic IDs. From work of co-author Koonin and colleagues, it’s clear the viruses (virions and their genomes) have become ‘giant’ on multiple occasions. Moving forward, what if some members of the taxonomic entity Mirusviricota were found to have ‘small’ genomes? What would that mean? So the following: “making Mirusviricota the second phylum of large and giant eukaryotic viruses” (and also in the conclusion) doesn’t really mean much (to me at least). All this is to say that I think the field of ‘giant’ virus research has moved to a point where it doesn’t really matter how big their genomes are in terms of novelty and the more that is discovered the more we are likely to see a range of sizes.

This is a matter of taste and don’t feel strongly about it, but I don’t see the value of adding even more arbitrary designations of size (genomes or virus particles?). This is the source of regular confusion for non-specialist audiences and adding more categories now is just setting the field up for more headaches in the future.

We certainly agree with the reviewer that the genomic size range of major viral lineages is rapidly increasing as thousands of (mainly environmental) genomes are sequenced and analysed. That acknowledged, for this paper, we would like to emphasize that *Mirusviricota* is the second known phylum of eukaryotic viruses, after *Nucleocytoviricota*, that generally includes viruses with relatively large genomes (average of 317 kb for MAGs with quality metric reaching 90%) some of which exceed 500 kb. Throughout the manuscript, we use the phrase “large and giant eukaryotic viruses”, instead of simply “giant viruses” (a commonly used label in other studies), both for *Nucleocytoviricota* and *Mirusviricota*, in order to reflect the broad range of genome sizes in both phyla, with only a minority exceeding 500 kb. The 500 kb cut-off is often used to label viral genomes as “giant” (e.g., PMID: 30635076). We certainly do realize that these definitions are arbitrary. However, historically, ‘giant viruses’ is a highly popular, frequently discussed concept, so using it has the benefit of bringing our work into the context of previous research in the field. Besides, and perhaps more importantly, this concept is helpful to adequately differentiate the two phyla in question, *Mirusvirivota* and *Nucleocytoviricota*, from many groups of viruses (in particular, the overwhelming majority of bacteriophages that have smaller genomes, well below the ‘giant threshold’, usually < 200 kb), as well as herpesviruses that have larger genomes but, so far at least, not crossing the 500 kb mark. Thus, we would prefer retaining the description of the mirusvirus genomes as “large and giant.

Line 120: “However, we also detected mirusvirus MCPs inside shipworm, sponge and coral specimens (e.g., skeleton samples of *Porites lutea*³⁵ and *Isopora palifera*³⁶), ‘...’. Is the implication here that mirusviruses infect animals? Or are the MCPs associated with

mirususes in protist symbionts / parasites within animals? Probably should be careful at this stage not to provide substrate for misinterpretation?

We share the curiosity of the reviewer regarding this signal. Unfortunately, we have no proper way of determining if those mirusviruses infect co-occurring unicellular eukaryotes or directly the animals. The broad MCP survey strongly suggests that the great majority of the hosts of mirusviruses are unicellular, but this is also the case of *Nucleocytoviricota*, among which several important groups, including poxviruses, asfarviruses, iridoviruses and ascoviruses, infect animals. The situation could be similar for mirusviruses, so we must be extra cautious with host assignment. Thus, we thought it was relevant to briefly mention this intriguing signal without making any dubious interpretations. To minimize the risk of misinterpretations, we have amended the text as follows:

*“The vast majority of the mirusvirus MCPs was identified in marine (79.6%) and freshwater (14.4%) ecosystems including surface and deeper layers of all oceans and seas, as well as lakes, thaw ponds and rivers across continents (e.g.,^{30–32}) (Table S1). Most of the remaining MCPs were identified in biofilms and sediments^{33,34}, with signal also present at the ocean bottom (hydrothermal plumes³⁷, oceanic crust³⁸), in continental groundwater³⁹, ice and streams of glaciers⁴⁰, as well as soil⁴¹ and thawing permafrost⁴². **Finally, although we detected some mirusvirus MCPs in shipworm, sponge and coral specimens (e.g., skeleton samples of *Porites lutea*³⁵ and *Isopora palifera*³⁶), this signal might come from co-occurring unicellular eukaryotes. Thus, there is currently no direct evidence that mirusviruses can infect animal cells.** Overall, this global survey of Mirusviricota MCPs dramatically increased the known diversity of mirusviruses, revealing their global prevalence that echoes the well-documented ecological prominence of nucleocytoviruses.”*

Line 139: “The genomic database includes four circular episomes^{7,9}, the only known chromosomal integrant⁹, ...”

Predicting circularity can be difficult, so maybe “obviously circular episomes”? This section of text leaves the reader wondering what the structure of the $1204 - 4 - 1 = 1199$ mirusvirus genomic elements are (inc. the 51 previously ID’d ones). It doesn’t really come up again.

For the vast majority of mirusviruses, genome fragmentation prevented us from making any conclusions regarding circular versus linear genome structure. However, in a few cases, including the episomes which were characterized elsewhere, circularity was confirmed by detecting overlap of terminal fragments with blastn. Nevertheless, we agree that predicting circularity can be difficult, and because the rest of our analysis does not rely on this feature of the genomes, we have simply removed the word. The sentence in question now reads as follows:

“The genomic database includes four episomes^{7,9}, the only known chromosomal integrant⁹, 50 previously characterized MAGs^{1,2}, and 1,202 newly characterized MAGs (96% of the genomes).”

Line 238: “Notably, Soporavirales displayed an intriguing trend, with high intron density in the episome (653 introns/Mb) and chromosomal integrant (624 introns/Mb) of one eukaryotic isolate⁹ and very few introns in the other 14 genomes (25 introns/Mb).” If the authors are referring to only one episome and integrant, and those elements are only ~300 Kb, why refer to them in terms of introns / Mb? I realize this is the useful metric for looking at large MAG datasets, but it seems misleading in this case.

The choice of introns/Mb was made in order to present numbers that could be easier to digest for the readers (e.g., 25 introns/Mb instead of 0.025 introns/kb). However, this is not important, and we agree that, for the cases mentioned by the reviewer, it is best to use introns/kb. Thus, we replaced “/Mb” by “/kb” in the main text, including in the sentence emphasized by the reviewer, which now reads as follow:

*“Soporavirales displayed an intriguing trend, with high intron density in the episome (**0.65 introns/kb**) and chromosomal integrant (**0.62 introns/kb**) of one eukaryotic isolate⁹ but very few introns in the other 14 genomes (**0.025 introns/kb**).”*

Line 252: “contained no introns”. Contained no obvious introns?

We agree that not all introns can be detected, and we modified the sentence as follow:

*“In sharp contrast, in most genomes of *Demutovirales*, **we did not detect any introns in the virion morphogenesis genes** (Figure S6).”*

Line 284: “The mirusvirus introns encoding homing endonucleases lack recognizable signatures of group I or II introns and instead are flanked by typical spliceosomal donor and acceptor sites”. What signatures did the authors look for? I think much of the self-splicing intron signatures studied in the literature are secondary structure based. It would be good for the authors to articulate this briefly (either at this part of the text or elsewhere). Were the predicted introns folded (with the HE-encoding regions removed)? I doubt it, as this would have been a ton of work, but I recommend that the authors clarify the extent to which they dug into the different possibilities here. Are these introns group I/II introns in origin that have somehow adopted GT-AG splice boundaries and are now removed by the spliceosome? Or are they canonical spliceosomal introns that have recently acquired HE genes? This could well be questions for the future, but at least touching on it here would seem to be worth it. I can only assume that they are spliceosomal introns first, since most are too small or (in the case of Fig. 2B) too large to be self-splicing (at least based on what is known about GPI and II introns).

We appreciate this comment that prompted us to address an interesting and important aspect of the intron evolution in mirusviruses. The signatures of self-splicing introns are interactions between the terminal regions resulting in characteristic secondary structures. Aside from the lack of reverse transcriptase (a feature of group II introns) in mirusvirus introns, our attempts to fold the mirusvirus introns after removing the protein-coding regions yielded no predicted stable secondary structures that would be even remotely similar to signatures of self-splicing introns. In particular, we used mmseqs2 to cluster HE protein sequences and then, following removal of the HE-coding region, attempted to fold representative introns from 20 most abundant clusters using Vienna fold. No structures resembling those of group - I/II introns were found. Moreover,

sequences of introns with similar HE were not conserved outside the HE gene boundaries, arguing against the presence of functionally important RNA structures.

Currently, there is no evidence supporting the evolution of the mirusvirus introns from self-splicing ones. Accordingly, the scenario of genes encoding homing endonucleases and other proteins invading spliceosomal introns appears to be favoured.

We expanded the portion of the main text quoted by the reviewer as follows:

*“The mirusvirus introns encoding homing endonucleases lack recognizable signatures of group I or II introns, **namely, complementary interactions between 5'-terminal and 3'-terminal regions that result in distinct, stable secondary structures. Our attempts to computationally fold these introns after removing the protein-coding regions yielded no structures resembling those in self-splicing introns, and actually, no stable secondary structures at all (see Methods). Instead, mirusvirus introns are flanked by typical spliceosomal donor and acceptor sites.**”*

Line 289: “Remarkably, SHI are for the most part inside the virion morphogenesis genes (64.1% of all SHI), and display target gene specificity, ...”. They are predicted to have target gene specificity, but this hasn't been demonstrated experimentally. This is made clear when talking about the HE function in the following sentence, but I'd suggest clarifying the language a bit here.

We have amended the sentence to address this point (note that we have now named those introns “shintrons”). It now reads as follow:

*“Remarkably, shintrons are for the most part inside the virion morphogenesis genes (64.1% of all shintrons), and **appear to have target gene specificity, so that the MCP genes are targeted by shintrons encoding HNH endonucleases (p-value < e-16, chi-square test), whereas the jelly-roll protein gene is invaded by shintrons encoding GIY-YIG endonucleases (p-value < e-16, chi-square test) (Figures 2 and S9, Table S4).**”*

Line 318: similar to above, “the corresponding introns display strong target gene preference...”. I don't understand exactly what “display” means here. Do you mean the intron insertion sites ‘are consistent with’ target site preference on the part of the HEs (or whatever proteins)? The introns themselves don't display target gene preference, that is a function of a trans-acting proteins that mediate homing, correct? For this comment and above, I think the language could be improved. I apologize if I have misunderstood.

We agree that “display” is rather confusing. Similar to the previous comment, we have amended the sentence to address this point. It now reads as follow:

*“Although these proteins lack conserved residues that could constitute an active site of a nuclease, the corresponding introns **appear to have a strong target gene preference, similar to SHIs (p-value < e⁻¹⁶, chi-square test).**”*

Line 693: “Intron-borne” genes”. As written this could mean genes that emerged de novo from introns (which can happen), but this is not what the authors mean. They mean genes inside existing introns (and presumably have landed within the intron)?

This is correct and we agree that the wording was confusing. We now have better clarified this, as follow:

“Although most of the encoded proteins lack functional annotation (Table S4), 334 genes located inside introns (hereafter intron-harboured genes) encode divergent endonucleases of the HNH (n=171), GIY-YIG (n=132), VSR-like (n=16) and PD-(D/E)XK (n=15) superfamilies (Figure S8).”

We thank once again the reviewer for their interest and constructive comments.

Reviewer #2

In this manuscript Medvedeva et al., leverage on Tara Ocean data and taxonomically demonstrate Mirusviricota comprises a highly diversified second phylum of large and giant eukaryotic viruses that rivals the evolutionary scope and functional complexity of viruses in the first characterized phylum, Nucleocytoviricota. Functionally they describe gene context difference including presence of hallmark genes required for replication in the cytoplasm as well some mirusviruses that included spliceosome introns linked to genes encoding essential proteins. They also propose two evolutionary scenarios. Overall, the manuscript is well structured and provided invaluable insights on the ever-growing NCLV.

We thank the reviewer for this appreciation of the value of our study with respect to the rapidly expanding range of large and giant eukaryotic viruses.

Abstract, Line 22: Please remove references from the abstract, as these are typically discouraged in that section.

We removed the abstract references.

Line 134: The statement indicates that non-redundant mirusvirus genomes share an average nucleotide identity (ANI) of <98%. Should this be >98% instead?

For the redundant MAGs (those with a high ANI), we only kept one representative (based on quality metrics when available, length when not) so that no pairs of MAGs in our final database have an ANI >98%. In other words, no MAGs in the final database have less than 2% sequence divergence at the nucleotide level within the scope of their alignment. Therefore, “ANI <98%” is correct.

Lines 708–713: The methodology for constructing the concatenated gene tree is unclear. Please provide additional details regarding how the tree was built, particularly whether each gene partition was assigned a unique evolutionary model.

For each gene (MCP, terminase, portal), we used ModelFinder to determine and select the best-fitting model. However, in all cases, the model turned out to be the same, as stated in

the legend of Figure S1. As a result, these single gene phylogenies as well as the concatenated phylogeny (Figure 1), all use the same model (LG+F+R10). We thank the reviewer for this comment that prompted us to clarify our methodology. The relevant portion of the Methods section now reads as follows:

“Phylogenetic reconstructions (both for individual hallmark genes and concatenations) were performed using IQ-TREE⁸³ v1.6.12 with “-m MFP -safe -alrt 1000 -bb 1000” parameters. ModelFinder⁸⁴ was used to determine and select the best-fitting model, which in all cases was the LG+F+R10 model. As a result, this model was used for all genes included in the concatenated phylogeny. Supports were computed from 1,000 replicates for the Shimodaira–Hasegawa (SH)-like aLRT⁸⁵ and UFBoot⁸⁶.”

Line 719: RED scores are indeed used in GTDB taxonomy assignments. However, the rationale behind selecting a RED score threshold of 11 is not clear. Were any benchmarking efforts performed using ICTV reference sequences to justify this cutoff? Some reasoning or justification for this threshold is necessary.

RED scores are indeed an important metric for our taxonomic framework of *Mirusviricota*. The score thresholds we used follow the exact same strategy as was used for *Nucleocytoviricota* (as stated in the main text with the relevant citation, and with the guidance of one of our authors that led the RED-centric taxonomic framework of *Nucleocytoviricota*): average RED score values below 0.22 for putative orders, and average RED score values below 0.65 for putative families (this is described in the Methods section). This provides for direct comparison of the taxonomic diversity of mirusviruses and nucleocytoviruses, two lineages of large and giant eukaryotic viruses. Note that details of the values across the phylogenomic tree of *Mirusviricota* can be explored in the Table S03 (sheet labelled “Orders and Families”).

Now, what the reviewer refers to (“RED score threshold of 11”) relates to the rooting strategy needed to determine the RED score values. In the case of *Nucleocytoviricota*, with two well defined classes (*Megaviricetes* and *Pokkesviricetes*), the situation was relatively easy and a single rooting, separating the two classes, was applied. However, when the relevant rooting (from an evolutionary standpoint, ideally using a proper outgroup) is unknown, the best practice indicated by GTDB is to use multiple rooting scenarios, all using deep-branching demarcations, and then average the RED score values for the taxonomic framework (“RED values used for rank normalization are averaged over multiple plausible rootings” – source: <https://gtdb.ecogenomic.org/methods>). In the case of *Mirusviricota*, we do not have a proper outgroup to use and as a result, we do not know the root position. Thus, we used all possible deep-branching demarcations (total of 11) in an effort to optimize the relevance of the RED score values. For clarity, here is a figure displaying where those 11 demarcations are:

We have now integrated this figure in a dedicated section of the newly drafted Supplemental information document.

We agree that this particular methodology was not sufficiently explained in the Methods section. We have amended the relevant section, which now reads as follows:

“A taxonomic framework for Mirusviricota: Relative evolutionary distance (RED) values were computed for each node of our concatenated phylogenomic tree (MCP, terminase, portal) by applying the “get_reds” function of the castor R package⁸⁷ on 11 distinct tree rooting positions, corresponding to all the major deep-branching positions of the tree. This multiple-rooting strategy was employed because the root position in the phylogenetic tree of Mirusviricota is currently unknown due to the lack of an appropriate outgroup. The average RED value from the 11 rooting positions was used to define nodes corresponding to putative orders (average RED score below 0.22) and putative families (average RED score below 0.65).”

Line 742: Functional annotation is based on Pfam, which is a reasonable approach. However, Pfam annotations can be ambiguous in certain cases. Have the authors validated the annotations using an additional database, such as eggNOG or KEGG?

While only the Pfam annotations were incorporated in the main text, we have also tested a series of alternatives, including KEGG Kofams (from within anvi'o). We now have also run eggNOG on all mirusvirus genes. Closely similar results were obtained with Pfam, KEGG Kofams and eggNOG. Most importantly, all methods recapitulate the differential occurrence of all the key functions (e.g., DNA and RNA polymerases as well as the DNA precursor synthesis module) described in the manuscript. To address this important point, we have now included the eggNOG and KEGG Kofams results into Table S4, which

contains information on all mirusvirus genes. We have also mentioned this new addition in the main text, as follow:

*“The absence of these functions (including DNA polymerase of any known family) was **validated by alternative functional annotations (see Method and Table S4) and confirmed by 3D structure predictions for all proteins identified in a complete Styxvirales genome and a near-complete Okeanovirales genome, ruling out the possibility that highly derived homologs could be missed by sequence analysis (Figure S13 and Table S6).**”*

Figure S15: How is lateral gene transfer (LGT) defined as recent in this figure? There appears to be no time calibration in the phylogenetic tree to support this statement.

Indeed, there is no time calibration per se. Supplemental figure S15, which shows four distinct examples of apparent lateral gene transfers between the two viral phyla based on single gene phylogenies, is mainly relevant for the identification of the direction of transfer. However, our bioinformatic workflow (involving, in particular DIAMOND BLAST) for gene flux detection which precedes phylogenetic analysis confines the signal to “relatively recent transfers”. Indeed, we only focused on potential gene flux corresponding to protein matches between the two phyla with a sequence identity >50%, a stringent cutoff that excludes matches for distantly related genes sharing the same function (e.g., the DNA polymerase between *Demutovirales* and *Nucleocyotoviricota*). In the revision, we improved Table S07 and added an new Method section (this was an oversight) to better summarize and explain our methodology on the gene flux. Finally, we also improved the paragraph in the Results section to address comments made by this reviewer and others, as follow:

*“To test this hypothesis, we performed a comprehensive sequence comparison of proteins encoded by *Mirusviricota* and *Nucleocyotoviricota*. We observed **relatively high levels of gene flux (provisionally defined by presence of highly similar homologs, with >50% amino acid sequence identity) between *Nucleocyotoviricota* and intron-poor mirusvirus families of *Demutovirales*, and, conversely, either no signal or **lower levels of gene flux between *Nucleocyotoviricota* and intron-rich mirusvirus groups (Figures 3, 4 and S14, Table S7). Relatively recent gene transfers (as inferred from the high sequence identity between homologous proteins from the two phyla) between specific lineages of mirusviruses and nucleocyotoviruses cover a wide range of functions that include a particularly strong signal for the DNA precursor synthesis module (e.g., dUTPase). For these genes, *Nucleocyotoviricota* to *Mirusviricota* gene flux directionality was confidently predicted (Table S7) and validated at least partially by phylogenetic analyses (Figure S15).**”***

Data Availability: A link to the updated HMM is not provided. The reference provided is to the initial HMM only. Please include access to the updated version.

The updated HMM for the MCP of mirusviruses is part of the HMMs provided in the Data availability section. In the revision, we clarified the statement:

“Data availability: All databases our study generated have been made publicly available at <https://figshare.com/s/33ebaa0ef9edf2df1f83>. Those include (1) MCP proteins identified in the global metagenomic survey, (2) the non-redundant database of 1,257 *Mirusviricota* genomes (both contigs and predicted genes in the form of nucleotide and protein coding sequences), (3) the intron-informed gene model for *Mirusviricota*, (4) HMMs for the *Mirusviricota* virion morphogenesis module (**including the updated HMM for the MCP of mirusviruses**), (5) the database of single-copy genomic MCPs used for the taxonomic annotations of metagenomic MCPs, (6) protein alignments and phylogenetic trees corresponding to the MCP, portal and terminase hallmark genes, (7) intron sequences found in the genomic database, (8) the global protein database (Bacteria, Archaea, Eukarya, plastids, *Nucleocytoviricota*, *Mirusviricota*) used for high-ranking taxonomic annotation of contigs, (9) and our protein database for *Nucleocytoviricota* and *Herpesvirales*.”

Additional Question: The authors present 3D-predicted structures of the terminase protein. How do these predicted structures from the MAGs compare with known reference structures? Does a structure-based evolutionary analysis (structure based tree) supports the inferred functions.

Terminase structures are highly conserved, which together with the conservation of the catalytic motifs of the ATPase and nuclease domains leaves no doubt regarding the function. More generally, homology between the mirusvirus terminase and terminases of other duplodnaviruses could be unequivocally established using sequence-based analyses. Thus, using structure-based comparisons to further validate this relationship appears unnecessary. The comparison of the structure-based trees and sequence-based phylogenies is an interesting avenue for future research on mirusviruses. However, for this comparison to be meaningful, many more (predicted) structures of mirusvirus proteins would be needed, whereas extensive structural modelling of mirusvirus protein families was outside of the scope of the present work.

While two evolutionary scenario's are proposed in a future direction paragraph under conclusion would be great to discuss how to mechanistically test.

Although evolutionary scenarios such as those for transition of mirusviruses from cytoplasmic to nuclear reproduction (or vice versa) cannot be tested directly, the strong prediction of this work, namely, that the intron-rich mirusviruses lacking the transcription machinery reproduce in the nucleus, certainly can and we hope and believe will be confirmed experimentally once appropriate mirusvirus-host system becomes culturable under laboratory conditions. We emphasize this in the revised Discussion:

“In the future, the establishment of relevant laboratory cultures will be key for testing predictions made in our study and for performing mechanistic studies of the homing endonucleases and enigmatic MING-1 proteins encoded by spliceosomal introns of mirusviruses.”

Reviewer #3

Summary

This manuscript presents a tour de force in environmental virology, dramatically expanding our knowledge of mirusvirus diversity through the characterisation of over 1,000 high-quality genomes. The authors make the compelling case that Mirusviricota represents a second phylum of large and giant eukaryotic viruses, rivaling Nucleocytoviricota in evolutionary scope and functional complexity. The discovery of widespread nuclear-replicating viruses that lack DNA polymerases and are enriched in spliceosomal introns challenges conventional understanding of large DNA virus biology. While the work is technically impressive and the major conclusions are well-supported, several aspects of the interpretation and presentation warrant further consideration.

We thank the reviewer for appreciating the impact of our work and for the constructive comments that we believe allowed us to substantially improve the manuscript as detailed below.

Strengths

The scope and scale of this study are remarkable. By surveying over 100,000 metagenomic assemblies and developing iterative binning approaches, the authors have increased the known diversity of mirusviruses by an order of magnitude. The methodological framework, particularly the iterative refinement of genome bins and the development of clade-specific quality metrics, represents best practices in environmental genomics that will be of interest to the field.

We are indeed satisfied with the performance of our iterative genome-resolved metagenomic survey, and we appreciate that the reviewer identifies this methodology as a strength of the study, even though this is less important compared to the insights highlighted below.

The manuscript's most exciting contributions are its novel biological discoveries. The identification of spliceosomal homing introns represents the first documented case of homing endonucleases within spliceosomal introns, potentially revealing a new mechanism for intron mobility. Similarly, the evidence for trans-splicing in terminase genes, supported by metatranscriptomic data, provides rare documentation of this phenomenon in DNA viruses. Perhaps most striking is the discovery of giant viruses that apparently lack DNA polymerases yet maintain genomes exceeding 500 kb, fundamentally challenging our assumptions about the minimal requirements for large DNA virus replication.

The evolutionary framework presented is thoughtful and well-reasoned. The authors' proposal of "evolutionary trap" and "steal and escape" models for transitions between cytoplasmic and nuclear replication provides testable hypotheses for understanding mirusvirus evolution. The phylogenomic analyses are robust, and the use of relative evolutionary divergence (RED) to establish taxonomic ranks follows current best practices in viral taxonomy.

We thank the reviewer for summarizing our key insights in such a positive way.

Major Concerns

My primary concern relates to how definitively the nuclear replication hypothesis is presented throughout the manuscript, beginning with the title itself. The phrase "nuclear predators" in the title presents the nuclear replication hypothesis as established fact, when it is actually an inference based on indirect evidence. While the genomic evidence is compelling—the absence of DNA replication machinery combined with extensive spliceosomal introns strongly suggests nuclear dependence—these remain inferences rather than direct observations. The abstract continues this pattern, stating that these viruses "reveal that the nucleus of unicellular eukaryotes is a major niche for giant virus reproduction," again presenting hypothesis as fact.

This is indeed a critical point, and we largely agree with the reviewer: we did not directly demonstrate that okeanoviruses and styxviruses replicate in the nucleus. We have now toned down our main conclusions regarding the cytoplasmic versus nuclear replication sites of *Mirusviricota* putative orders, to better reflect the lack of available cultured representatives and imaging data related to the two putative orders of *Okeanovirales* and *Styxvirales* that are central to our study. Changes to this effect were made in the title, Abstract and Discussion sections. Nevertheless, we have maintained a confident tone on this issue given that it is difficult to imagine (a view also shared with the reviewer) how viruses full of spliceosomal introns (especially in the virion morphogenesis gene module) and lacking any polymerase (among other key functions for cytoplasmic replication) can create a viral factory and replicate those large genomes in the cytoplasm.

The title now reads as follow:

Intron-rich mirusviruses widespread in aquatic habitats are predicted to reproduce in nuclei of unicellular eukaryotes (Eugene)

The abstract was shortened to follow the journal guidelines and now reads as follow (only the relevant change is in **bold**):

*"Mirusviruses infect unicellular eukaryotes and are related to tailed bacteriophages and herpesviruses. Here, we dramatically expand the known diversity of mirusviruses and characterize more than a thousand high-quality genomes. Mirusviricota comprises a highly diversified **second** phylum of large and giant eukaryotic viruses that rivals the evolutionary scope and functional complexity of nucleocytoviruses. Critically, major Mirusviricota lineages lack essential genes encoding components of the replication and transcription machineries, and concomitantly, encompass numerous spliceosomal introns that are particularly enriched in virion morphogenesis genes. These features point to multiple transitions from cytoplasmic to nuclear reproduction during the evolution of mirusviruses. Many mirusvirus introns encode diverse homing endonucleases, suggestive of a previously undescribed mechanism promoting the horizontal mobility of spliceosomal introns. We also provide transcriptomic evidence for long-range trans-splicing in a virion morphogenesis gene. Collectively, our data **strongly suggests** that nuclei of unicellular eukaryotes in freshwater and marine ecosystems worldwide are a major niche for replication of intron-rich mirusviruses."*

Here is a summary of relevant changes in the Introduction, Results and Discussion:

*“The reliance of intron-rich mirusviruses on host replication and transcription machineries **strongly suggests** that the nuclei of unicellular eukaryotes are a major niche for giant virus reproduction worldwide.”*

*“The lack of the DNA replication and transcription machineries **strongly suggests** that okeanoviruses and styxviruses, similarly to herpesviruses, complete all stages of their reproduction cycle in the nucleus.”*

*“Their hallmark features, such as **apparent** nucleus-centric replication and terminase gene trans-splicing reinforce the direct evolutionary connection between mirusviruses and animal-infecting herpesviruses.”*

*“Overall, our cross-biome diversity survey and subsequent characterization of the complex Mirusviricota genomic landscape **strongly suggest** that mirusviruses fill a major subcellular virus reproduction niche among protists, the host cell nucleus, previously thought to be only sparsely occupied by giant viruses.”*

This issue permeates the manuscript. For instance, statements like "the nucleus of unicellular eukaryotes is a major niche for giant virus reproduction" make strong claims based on indirect evidence. The manuscript would benefit from more nuanced discussion acknowledging the assumptions underlying these inferences. Could some viruses utilise host DNA polymerases in the cytoplasm? Might intron-rich viruses still have early cytoplasmic stages before nuclear entry? The authors briefly mention that some intron-containing demutoviruses might replicate partly in the cytoplasm, but this complexity deserves fuller treatment throughout the manuscript. A more accurate title might be "Intron-rich mirusviruses lacking DNA polymerases suggest widespread nuclear replication in unicellular eukaryotes" or something similarly qualified.

As mentioned above, we have toned down the statements on nuclear reproduction of intron-rich mirusviruses, to acknowledge that this is inference, albeit based on, in our opinion, compelling evidence. We also changed the title (see previous comments). In the original version of the manuscript, we suggested that demutoviruses likely have a cytoplasmic phase, but might also have a nuclear phase. Given the uncertainty regarding demutoviruses (which will only be resolved once cultivation of these viruses becomes possible, as is now stated in the Discussion), we elected not to expand too much on these speculations and focus instead on the two other major putative orders, okeanoviruses and styxviruses, that are central to the insights of our investigation.

To our knowledge, there is no evidence of relocation of host DNA polymerases to the cytoplasm which is why the great majority of nucleocytoviruses encode their own DNA polymerase that enables the cytoplasmic replication. In addition, other genes are required for DNA replication and transcription in the cytoplasm, including enzymes for DNA synthesis and RNA polymerase subunits, which most intron-rich mirusviruses lack (this point is thoroughly described in the manuscript). Relocation of all these enzymes from the nucleus to cytoplasmic viral factories is unprecedented among other known viruses. Although the lack of precedence does not formally exclude the theoretical possibility of such translocation, we consider this scenario to be highly unlikely. Thus, although uncoating of intron-rich mirusviruses might happen in the cytoplasm, transcription most likely occurs in the nucleus, given that they do not encode their own transcription

machinery. We edited the Discussion accordingly to emphasize this argument but, regrettably, we do not have room to elaborate:

“By contrast, okeanoviruses (solely marine viruses) and styxviruses (found in marine and freshwater ecosystems) contain many spliceosomal introns, lack the genes required for deoxyribonucleotide biosynthesis, DNA replication and transcription, and in all likelihood, cannot reproduce in the cytoplasm. **Thus, although at least partial virion uncoating in the cytoplasm is plausible, our results suggest that early gene transcription takes place in the nucleus.** The most notable function lacking in most okeanoviruses and styxviruses is the viral DNA polymerase, which until now had been considered indispensable for the replication of large viral genomes, in particular, nucleocytooviruses and herpesviruses⁶¹.”

A critical technical concern involves the potential impacts of metagenomic binning, which is not yet widely established practice in viral ecology. The authors report 21 genomes exceeding 500 kb, but it's unclear whether these represent single contigs or binned assemblies. If these are bins rather than contiguous sequences, there's significant risk of genome size inflation through incorrect binning. This concern is particularly acute for the "cryptic putative orders" that are substantially enriched in singleton genes (43% on average). Could some of these unusual genomes represent chimeric bins combining multiple viral genomes or contamination from cellular sources? The authors should clarify: (1) how many of their large genomes are single contigs versus bins, (2) what additional validation was performed for binned genomes, especially those with unusual characteristics, and (3) whether bin contamination could explain some of the extreme genomic features observed. Given that viral binning relies primarily on sequence composition and coverage patterns—methods developed for cellular organisms with different genomic constraints—the possibility of binning artifacts deserves explicit discussion.

The reviewer points to important considerations, questions and concerns regarding the environmental genomes, which are central to many studies and often go far beyond what cultivation (a central research front) make accessible in the laboratory. In the context of our study that includes well tested (manual binning, inspection and curation within *anvi'o*) as well as novel methodologies for MAGs (our iterative automatic binning, which was critical to expand our survey and dive into the cryptic putative orders), we present in the following sections three contextual points: (A) our extensive expertise on MAGs, (B) technical binning aspects of the study, (C) and the complex evolutionary history of the phylum *Mirusviricota* (especially in the case of the cryptic putative orders) as inferred from our marker gene phylogenies focused on the virion morphogenesis module (MCP, terminase, portal). Finally, the last section (labelled D) directly addresses all the points made by the reviewer.

A. Lessons learned from 10 years of genome-resolved metagenomics with Tara

We have a long publication record history on genome-resolved metagenomics especially within the scope of the highly complex *Tara* Oceans metagenomic co-assemblies (Bacteria, Archaea, eukaryotes, chloroplasts, Nucleocytooviricota, and more), which started in 2015, and takes advantage of (1) the bioinformatic platform *anvi'o* for manual

binning, inspection and curation of individual MAGs with the help of a highly elaborate interactive interface, (2) and reference or newly developed single copy core gene collections to assess key quality metrics of MAGs (gold standard in the field). Here are some of our key publications that successfully applied this methodology:

<https://www.nature.com/articles/s41564-018-0176-9>

<https://www.sciencedirect.com/science/article/pii/S2666979X22000477>

<https://www.nature.com/articles/s41586-023-05962-4>

<https://www.nature.com/articles/s41586-023-05962-4>

To this date, manual binning within *anvi'o* is widely considered as the best practice in the field. Its only weakness is that it is labour intensive, hence the use of automatic binning by many researchers, especially when dealing with a large amount of metagenomic assemblies.

The discovery of mirusviruses in 2023 was made possible thanks to *anvi'o* (programs and interface) and the original use of “phylogeny-guided binning”. Most critically, it is because we visualized and manually curated the first mirusvirus MAGs that we could gain confidence in their quality and biological relevance. A brief description of our binning method and how it was used for viral metagenomic binning is available as a short blogpost (<https://anvio.org/blog/mirus-discovery/>). This blogpost and another one showcasing the binning of a large Nucleocytoviricota MAG (<https://anvio.org/blog/giant-viruses/>) directly address some of the reviewer concerns on the front of the quality of manually curated viral MAGs. Thanks to years of explorations of the *Tara* Oceans metagenomic assemblies with *anvi'o*, it is now well established (blogposts and publications) that differential coverage and sequence composition are equally relevant to Bacteria, Archaea, eukaryotes and their large viral genomes (both *Nucleocytoviricota* and *Mirusviricota*).

B. Technical points regarding our binning framework for mirusviruses

In the present study, our methodology for binning can be delineated into two complementary approaches: manual binning and curation (inside *anvi'o*) versus the newly developed iterative automatic binning (in the context of a global genomic resource for effective decontamination). Here we briefly summarize the key aspects of the two approaches (which are also adequately described in the Methods section):

(1) **Manual binning and curation:** Following the 2023 publication on mirusviruses that focused on the RNAPolB gene for binning (this effort was constrained to what we now call the putative order *Demutovirales*), we improved the HMM for the MCP, and used that more relevant marker to perform a second round of manual binning and curation from the large metagenomic co-assemblies of *Tara* Oceans. This work allowed the recovery of hundreds of MAGs beyond the scope of *Demutovirales*. We have high confidence in the quality of the MAGs, not just because of the use of the single copy core genes, but also because of the high coherence of contigs within a MAG with respect to both distribution patterns across *Tara* Oceans metagenomes (the so-called differential coverage information), and the sequence composition (the broadly used tetranucleotide frequency). Critically, this first extended database of *Mirusviricota* MAGs adequately covered the main families of *Demutovirales*, *Okeanovirales* and *Styxvirales*. At that stage, we knew that *Mirusviricota* was substantially more diverse compared to our initial survey focused on the RNAPolB gene (*Okeanovirales* and

Styxvirales lack this marker), and we wanted to expand our genomic survey far beyond our few *Tara* Oceans co-assemblies. This is why we transitioned from the manual binning to the iterative automatic binning using a very large legacy of metagenomic assemblies from mOTUs.

(2) **Iterative automatic binning of the mOTUs database:** On the one side, we acquired all the automatically generated bins from the mOTUs database that contained at least one MCP from *Mirusviricota* (based on our improved HMM). These bins were defined with metabin2 as part of the mOTUs workflow (led by some of the authors of the present study), using differential coverage and sequence composition. On the other side, we created a comprehensive database of genomes (mainly MAGs from our 10 years of binning with *Tara* Oceans) covering Bacteria, Archaea, eukaryotes, chloroplasts, *Nucleocytoviricota* and the extended database of manually curated mirusvirus MAGs. We created a DIAMOND database of this resource, and developed a simple method based on hits to determine the high-ranking taxonomy of contigs from the mOTUs bins as well as family-level assignment of mirusvirus contigs. This allowed a first iteration in which we retained mirusvirus contigs from the mOTUs bins, extracted MAGs (rarely more than one per mOTUs bin), and estimated their quality, when possible, using a first round of single copy core gene collections. We integrated the additional mirusvirus MAGs >50kb into the DIAMOND database for a second iteration of the process, allowing detection of more mirusvirus contigs. Finally, in the last iteration, we focused on mOTUs contigs with very low taxonomic signal from the DIAMOND blast and successfully extracted a series of MAGs that were later found to represent a highly diversified set of cryptic mirusvirus lineages. Key aspects of the iterative binning endeavour are described in the Table S02.

In summary, the manual binning effort was constrained to a few large metagenomic co-assemblies but provided a much-needed foundation for our global genomic survey of mirusviruses. The iterative automatic binning allowed the recovery of more than a thousand non-redundant MAGs, including the cryptic putative orders mainly thanks to the last iterative step. Because of the relevant data in the DIAMOND database, and because of the single copy core gene collections, we remain highly confident in the quality of most mirusvirus MAGs despite the use of an automatic binning strategy without visual inspection in *anvi'o* (this was not possible because the mOTUs bins currently are not integrated into *anvi'o*, which requires a series of specific files that could not be generated in the context of our study).

C. The complex evolutionary history of *Mirusviricota* challenges the recovery and quality assessment of MAGs

After generating our final database of non-redundant mirusvirus MAGs, we selected 3 key marker genes from the virion morphogenesis module that were well represented across clades: the MCP, terminase and portal genes. What we learned is that (1) those markers are highly congruent (Figure S1), (2) and that MAGs from the third automatic binning iteration correspond to highly diversified and undersampled mirusvirus clades that we labelled as cryptic. But despite the lack of single copy core gene collections for most of those cryptic clades, which prevents optimal quality assessment, it is important to keep in mind that those MAGs, which were targeted from the standpoint of the detection of an MCP, contain the other two marker genes which provide a very similar phylogenetic signal. This is why we are confident in the biological relevance of those cryptic MAGs.

Again, because of our inability to create relevant single copy core gene collections (not enough closely related genomes are available for now), our view is that those MAGs should be used with more caution compared to the MAGs affiliated with *Demutovirales*, *Okeanovirales* and *Styxvirales*. Since our investigation focuses mainly on fundamental genomic property differences between these three major putative orders, this limitation does not affect the interpretations of the results.

D. Addressing the reviewer comments

The reviewer shared legitimate concerns and asked specific questions on the relevance of the MAGs. With the context on binning now extensively described, we can more adequately address those concerns and questions, which we organized in the following subsections:

(1) Mirusvirus MAGs >500kb:

None of the MAGs >500 kb corresponds to a single contig, as described in the Table S03 that summarizes many aspects of the genomic database. This is inherent to metagenomic assembly fragmentations from short Illumina reads leading to the rarity of contigs that are hundreds kilobases in length especially from complex biomes such as the oceans (the leading source of MAGs in our investigation).

Here we provide the relevant metric for MAGs >500 kb (extracted from Table S03):

MAG_ID	RED_Order	RED_Family	nb_contigs	MAG_size	GC_percent	N50	completion estimate	redundancy estimate	quality_score metric
MIRUS_G_0396	Extra_Order_7	Singleton	29	792024	42,02	32802			
MIRUS_G_0095	Demuto	Demuto_05	13	729701	34,25	121468	98,97	18,56	80,41
MIRUS_G_0157	Extra_Order_6	Singleton	10	620979	27,81	91228			
MIRUS_G_0560	Demuto	Demuto_05	40	598616	36,36	22619	98,97	4,12	94,85
MIRUS_G_0265	Demuto	Demuto_05	20	595914	36,31	47500	98,97	3,09	95,88
MIRUS_G_0225	Styx	Styx_01	16	593698	49,02	53829	97,87	6,38	91,49
MIRUS_G_0483	Demuto	Demuto_05	30	587369	36,23	26221	97,94	2,06	95,88
MIRUS_G_0370	Demuto	Demuto_05	21	575922	37,9	35089	96,91	6,19	90,72
MIRUS_G_0073	Styx	Styx_01	16	572385	48,59	137107	97,87	2,13	95,74
MIRUS_G_0352	Demuto	Demuto_05	20	570335	44,46	40727	98,97	4,12	94,85
MIRUS_G_0013	Demuto	Demuto_05	6	556078	37,9	304467	97,94	4,12	93,81
MIRUS_G_0777	Demuto	Demuto_05	48	542679	38,27	14342	97,94	6,19	91,75
MIRUS_G_0619	Demuto	Demuto_05	36	540914	41,58	19849	95,88	10,31	85,57
MIRUS_G_1380	Demuto	Demuto_10	136	540101	48,97	3906			
MIRUS_G_0771	Demuto	Demuto_05	50	535085	36,67	14426	96,91	2,06	94,85
MIRUS_G_0253	Demuto	Demuto_05	18	533204	42,11	49453	98,97	7,22	91,75
MIRUS_G_0514	Extra_Order_6	Singleton	28	530909	60,36	24411			

Critically, most of the MAGs >500 kb correspond to well described putative orders (especially *Demutovirales*, but also *Styxvirales*) for which we had enough genomic material to successfully build family-level single copy core gene collections. Only 4 out of the 21 MAGs lack a quality metric based on the highly relevant single copy core genes, which is best practice also for bacteria, archaea and eukaryotes (e.g., see <https://www.nature.com/articles/s41564-018-0176-9>). In the case of Demuto_05 (the Family ID 05 of the putative order *Demutovirales*), the single copy core gene collection covers 97 distinct genes we can rapidly screen using 97 dedicated HMMs. As seen in the table provided here, the quality scores are quite outstanding for Demuto_05 and for the other families, which strongly indicates (in the context of our DIAMOND database allowing (1) exclusion of contigs not related to *Mirusviricota*, (2) and family-level taxonomic assignment of mirusvirus contigs) that the MAGs display no (or a very low level of contamination). With those results in the context of our global framework, we demonstrate that there is no significant risk of genome size inflation through incorrect

binning for most MAGs >500 kb in length, as well as for the vast majority of MAGs overall (Table S03). Again, the use of single copy core genes is the best practice in the field of genome-resolved metagenomics, and its use for large viral genomes is legitimate, and in the case of our investigation also highly effective.

In summary, we are highly confident that most, if not all of those 21 MAGs are biologically relevant and correspond to mirusvirus genomes that are larger than 500 kb, a feature that is seemingly relatively common especially in the Family ID 05 of *Demutovirales*. In contrast, we have limited confidence in the longest mirusvirus MAG, because it covers a cryptic clade for which we could not build any relevant collection of single copy core genes. This is why we did not advertise such a long MAG in the main text. In the future, more genomes for *Mirusviricota* will allow researchers to better assess the quality of a broader range of mirusvirus clades. For now, we can gain high confidence mostly for the putative orders *Demutovirales*, *Okeanovirales*, *Styxvirales* and *Soporavirales*, which are the focal clades of this investigation.

(2) MAGs from the cryptic putative orders:

As described in the sections B and C, one must consider the MAGs corresponding to cryptic lineages with a lower confidence compared to the major putative orders, due to the lack of single copy core gene collections. On the other hand, we are highly confident in the overall biological relevance of the cryptic clades for two complementary reasons: (1) the use of a wide range of genomes in the DIAMOND database (see the section B and Methods of the manuscript for more details), which includes various eukaryotic lineages also occurring abundantly in the sunlit ocean (e.g., diatoms, green alga, Haptista and Cryptista), was done by design to minimize the risk of contaminations originating from outside of *Mirusviricota*, (2) and the mirusvirus cryptic MAGs contain most if not all hallmark genes of *Mirusviricota*, with those marker genes having a good congruence in terms of phylogenetic signal based on the single gene phylogenies described in Figure S1. As a result, we are confident in all the claims we made regarding the cryptic putative orders, including the high density of singleton genes. The most logical explanation is that those singletons are due to the genomic under sampling of these lineages. The only aspect we maintain a low confidence on is the length of the very long cryptic mirusvirus MAGs, as those could represent two or more closely related mirusvirus genomes co-occurring in the environment, leading to their contigs being put together in the same mOTUs bin. This is the main reason why the lack of single copy core genes is a limitation, and in that case, we fully agree with the reviewer that there is a more acute risk of genome size inflation. Again, this has no impact whatsoever on all the insights of our investigation, since we do not use the length of cryptic MAGs to make any claim (i.e., most MAGs >500 kb are not cryptic).

(3) First direct question by the reviewer: how many of their large genomes are single contigs versus bins?

This question is addressed in the section A. Given that all the MAG details are available from Table S3 and given that we did not claim that the MAGs >500 kb are contiguous or complete, we did not modify the manuscript. We are confident that our extensive clarifications to the reviewer are sufficient.

(4) Second direct question by the reviewer: what additional validation was performed for binned genomes, especially those with unusual characteristics?

Our detailed binning protocol is described in the section B. We applied two independent binning strategies, both providing high quality MAGs as assessed by the single copy core gene collections. For the cryptic MAGs, which for the most part lack single copy core gene collections, both the DIAMOND database and the multiple hallmark genes used for phylogenies helped us gain confidence in their biological relevance. Note that some of those cryptic MAGs have very low genomic fragmentation, further increasing the confidence in their quality. In the most extreme case, one of the cryptic MAGs corresponds to a single contig of 427 kb with key hallmark genes and 149 detected introns. This example showcases the value of those cryptic MAGs which can correspond to rather large viral genomes containing the key hallmark virion morphogenesis genes and at times are highly enriched in introns. Finally, some cryptic MAGs were also characterized from the manual binning endeavour, for which manual inspection and curation in the context of especially the differential coverage information (the most central component of binning with *Tara*) was done.

(5) Third direct question by the reviewer: Could bin contamination explain some of the extreme genomic features observed?

Aside from the extreme length, all the notable genomic features observed (high degree of singletons for some, enrichment of introns or else lack of DNA polymerase for others) are recapitulated in single contig MAGs of our database (Table S3), which indicates that none of the “extreme genomic features observed” could be due to contaminations. Regarding the extreme length, it is the single copy core gene collections that indicate that contamination cannot explain this feature (this is addressed in detail in a previous section dedicated to MAGs >500kb).

We thank the reviewer for pushing us to better describe those important methodological aspects, which have now been integrated into a supplemental information document. We updated the manuscript as follow:

“We performed two genome-resolved metagenomic surveys (a foundational manual binning and curation of the Tara Oceans metagenomic co-assemblies followed by the iterative automatic binning of MCP-containing bins from the mOTUs database), resulting in characterization of genomes from a broad range of distantly related lineages (see Table S2, Methods, and Supplemental Information).”

The uncertainty surrounding host range represents another significant limitation that deserves more explicit acknowledgment. While the authors broadly claim that mirusviruses infect “unicellular eukaryotes,” direct evidence for host identity exists for only a tiny fraction of the characterised viruses. The vast majority of the 1,257 genomes derive from environmental metagenomes where virus-host linkages cannot be established. The remarkable genomic diversity observed—with GC content ranging from 25% to 72%—certainly suggests a broad host range, but this remains inference. This uncertainty has important implications for the evolutionary scenarios proposed, as different protist lineages possess vastly different nuclear architectures, splicing

machineries, and life histories. The authors might consider attempting computational host prediction using established methods, such as co-occurrence patterns, or shared gene content with potential hosts. Alternatively, they should more clearly articulate why host identity uncertainty doesn't fundamentally affect their main conclusions about nuclear replication.

We agree that the current information on mirusvirus hosts was not sufficiently described in the submitted manuscript. We are actively working on a parallel study dedicated to gene flux (the “shared gene content” mentioned by the reviewer) between mirusviruses and eukaryotes. For the sake of transparency, we share here in confidence that okeanoviruses most likely infect the *Haptista* while styxviruses most likely infect the *Cryptista*. Results for *Demutovirales* is less clear but the largest eukaryotic signal is for the clade of MAST. Thus, this parallel study clearly links the 3 major putative orders of *Mirusviricota* with 3 high ranking lineages of unicellular eukaryotes: Haptista, Cryptista and Stramenopile. However, the results of this detailed analysis cannot be integrated into the present manuscript, and will be published as a separate paper. Thus, we have only amended the main text to more completely cite the published literature on mirusvirus predicted hosts, including a recent preprint by Hiroyuki Ogata's lab (Prof. Ogata being also an author of the present study). It should be noted, however, that as long as there is confidence that the hosts of mirusviruses are eukaryotes, the uncertainty about the host range has no bearing on our conclusions regarding the nuclear site of replication. Changes were made in the Introduction, as follows:

*“Mirusviruses are abundant in marine and freshwater ecosystems where **they are predicted to infect a broad range of unicellular eukaryotes (protists)**¹⁻⁷.”*

A change was also made in the Discussion about host uncertainties:

“Although much uncertainty remains regarding the host range of mirusviruses, our survey and others studies¹⁻⁷ all indicate that they predominantly infect unicellular eukaryotes, which are abundant in aquatic ecosystems.”

Importantly, knowledge about the hosts gained in our parallel study have no impact on the evolutionary models described here. There is no obvious difference, in our knowledge, between MAST on one side (predicted to be primarily infected by intron-poor demutoviruses containing the polymerases), and Haptista and Cryptista on the other side (predicted to be infected by intron-rich okeanoviruses and styxviruses that lack any polymerase). More broadly, all the eukaryotes have in common a nucleus with its spliceosomal machinery, and ribosomes in the cytoplasm. Those are the fundamental aspects of the eukaryotic cell structure that our evolutionary models are based upon. As a result, it is our view that lack of host knowledge in the present manuscript does not impact our data interpretation and conclusions. The only notable insight from the host, at least to us, is that *Cryptista* and *Haptista* have a special shared evolutionary history with the chloroplast having been transferred from the former to the later by means of tertiary endosymbiosis. Thus, it is possible that okeanoviruses and styxviruses also have a shared evolutionary history, albeit relatively ancient given the phylogenetic signal. Again, this has no obvious link to the evolutionary models of transition between cytoplasm and nucleus described in the present study.

The functional interpretation of the novel genetic elements discovered, while fascinating, remains highly speculative and would benefit from more mechanistic consideration. For the spliceosomal homing introns, the proposed mechanism by which homing endonucleases would promote intron spread between viruses is unclear. Classical homing mechanisms operate within a genome through DNA repair processes, not between different viral genomes during co-infection. What selective advantage would intron acquisition provide to viruses, given that introns generally appear deleterious for viral fitness? Why are these elements specifically enriched in virion morphogenesis genes? Similarly, the MING-1 proteins are proposed to function as RNA chaperones based primarily on structural predictions, but no functional evidence supports this hypothesis. The extreme conservation of MING-1 insertion position across divergent viral lineages implies strong selective pressure, but for what function? The authors could either provide more detailed mechanistic models, present additional bioinformatic support, or more explicitly frame these as mysteries requiring future investigation.

We certainly share the reviewer frustration, since we failed to elucidate some of the most fundamental questions surrounding introns and genes inside them (especially the homing endonucleases and MING-1) within the scope of *Mirusviricota*. It remains unclear if introns and/or spliceosomal homing introns are beneficial or detrimental to the virus, and if their spread/dynamics is (1) mainly neutral, (2) promoted by viruses within the scope of their individual genomes for gain in fitness, (3) promoted by viruses within the scope of other genomes to neutralize closely related co-infecting viruses, (4) or promoted by the host as a defence mechanism against viruses. We have now clarified this uncertainly in the Discussion, as follow:

*“In the case of endogenous nucleocytoviruses, introns are considered as a sign of viral genome decay, endogenization and assimilation into the eukaryotic genome, consistent with the lack of transcription of the viral genes⁴⁸. This is in stark contrast with mirusviruses, where genomes are transcriptionally active, with high levels of expression and processing of the intron-rich genes. **The roles, if any, of introns and spliceosomal homing introns among mirusviruses remain to be elucidated. Notably, it has been recently suggested that viruses can serve as vectors for the spread of introns in microbial populations^{67,68}, and intron-rich mirusviruses might play a major role in this process.**”*

“In the future, establishing relevant laboratory cultures will be key for testing predictions made in our study and for performing mechanistic studies of the homing endonucleases and enigmatic MING-1 proteins encoded by spliceosomal introns of mirusviruses.”

Additional Considerations

The ecological implications of the nuclear versus cytoplasmic replication strategies deserve deeper exploration. How might these different strategies affect viral population dynamics, host range evolution, or ecosystem functioning? The authors touch on niche partitioning between viral phyla but could develop these ideas further.

These are truly interesting issues but the connection with virus replication strategy and ecology is not straightforward, and any discussion of this aspect would be speculative for which we simply do not have space in this article.

The statistical analyses appear sound, though the authors should clarify whether multiple testing corrections were applied for the numerous chi-square tests performed on intron distributions.

Chi-square tests were performed to show enrichment of specific homing endonucleases in virion module genes. In total, seven chi-square tests were made (details with contingency tables presented in Table S4), and no multiple test corrections were applied.

Recommendations for Improvement

First, the authors should revise the title and abstract to more accurately reflect the inferential nature of the nuclear replication hypothesis. Throughout the manuscript, adopt more careful language distinguishing direct observations from inferences. Phrases like "strongly suggests," "is consistent with," or "likely indicates" would be more appropriate than definitive statements about nuclear replication in many instances.

We have modified the manuscript accordingly (see responses to previous comments)

Second, a dedicated paragraph in the discussion acknowledging the limitations of metagenomic inference would strengthen the manuscript. This should explicitly address what can and cannot be concluded from the available data, the potential impacts of binning artifacts, and outline key experiments or observations that would test the nuclear replication hypothesis.

Given that the manuscript must adhere to the length limits of *Nature Microbiology*, unfortunately, there is not much room to add an entire paragraph on this point. However, we do recognize the importance of stating the limitations. Thus, we have now detailed the binning process in a Supplemental Information document which addresses the points made (see our extensive comments on binning above), and slightly edited the Discussion as follows:

*"Apart from a wide range of deep-branching cryptic putative orders (currently highly undersampled, **which in most cases, made it impossible to derive the respective single copy core gene sets; see Supplementary Information**), Mirusviricota includes three major putative orders, which include most of the currently available genomes: Demutovirales, Okeanovirales and Styxvirales."*

Third, the novel genetic elements discovered deserve a more thorough mechanistic treatment. Even if definitive functions cannot be assigned, the authors should present specific, testable hypotheses about how spliceosomal homing introns might spread and what advantages MING-1 proteins might provide.

Under the article length limit of *Nature Microbiology*, we do not really have space for the discussion of these aspects. In the revised Discussion, we now conclude with the following sentence:

“In the future, establishing relevant laboratory cultures will be key for testing predictions made in our study and for performing mechanistic studies of the homing endonucleases and enigmatic MING-1 proteins encoded by spliceosomal introns of mirusviruses.”

Finally, consider adding a brief section on host prediction attempts or explain why such analyses weren't pursued. This would address the host range uncertainty more directly.

As already described in a previous response to the reviewer, we are actively working on a parallel study dedicated to the host prediction, which strongly points towards different high-ranking clades of unicellular eukaryotes that co-exist abundantly with mirusviruses in aquatic ecosystems. This is a separate study that we cannot integrate in the present study, which is already quite packed with important aspects of the evolution and genomic structure of mirusviruses. Unfortunately, we have no preprint for this parallel study to cite in the present manuscript, and thus we cannot take advantage of the main findings for the evolutionary models described. We elected to better clarify that we have many uncertainties regarding the host range, especially due to the lack of cultured representatives for most of the major mirusvirus families.

Minor Points

- The choice of 98% ANI for non-redundancy (line 134) should be justified

We chose the 98% ANI cut-off to align with our environmental genomic surveys of the last ten years with Tara Oceans, including the 2023 paper on mirusviruses. So, multiple publications are available to justify this point. We have now modified the text to address this point:

*“We created a database of 1,257 high-quality, non-redundant mirusvirus genomes (average nucleotide identity <98%, **in line with previous reports**^{1,20,44}), with a mean size of 265 kb and GC-content ranging from 25% to 72% (Table S3).”*

- Clarify whether genome completeness (line 136) refers to mean or median

We modified the text to clarify this point:

*“Analysis of clade-specific single copy core genes indicate that **these genomes have a mean of 88% completion.**”*

- Figure 4B text is difficult to read and should be enlarged

Done

- Minor typographical error at line 452 (extra bracket)

We thank the reviewer and the typo is now fixed.

Conclusion

This manuscript makes fundamental contributions to our understanding of giant virus diversity and evolution. The discovery of a major group of potentially nuclear-replicating DNA viruses with unique genomic features will undoubtedly stimulate new research

directions in environmental virology and viral evolution. The quality of the genomic data and analyses is exceptional, and the biological insights are profound. While questions remain about host range, replication mechanisms, and the function of novel genetic elements, these uncertainties do not diminish the importance of the findings. With revisions addressing the presentation and interpretation concerns outlined above, this work would make an excellent addition to Nature Microbiology and significantly advance the field.

We thank the reviewer for the positive and highly constructive review that helped us to strengthen and clarify the manuscript.